# Structural details of helix-mediated multimerization of the conserved region of TDP-43 C-terminal domain

Azamat Rizuan [1,7], Jayakrishna Shenoy[2,7], Priyesh Mohanty[1], Patricia M. dos Passos[3], José F. Mercado Ortiz[2], Leanna Bai [2], Renjith Viswanathan[2], Julia Zaborowksy[2], Szu-Huan Wang [2], Victoria Johnson[2], Lohany D. Mamede[3], Amanda R. Titus[3], Yuna M. Ayala [3], Rodolfo Ghirlando[4], Jeetain Mittal [1,5,6] ✉ & Nicolas L. Fawzi[2] ✉

Pathological inclusions of the C-terminal domain (CTD) of TAR DNA binding protein-43 (TDP-43) are neurodegenerative hallmarks in amyotrophic lateral sclerosis (ALS) and frontotemporal dementia, yet CTD's aggregation propensity complicates structural characterization of native TDP-43. Here we propose structural models for the physiological multimerization of TDP-43 CTD's conserved region (CR) essential for TDP-43 RNA processing. Using NMR spectroscopy, we establish that the native state of TDP-43 CR at physiological conditions is α-helical. Hydrophobic residues drive CR helix-helix assembly, phase separation, and TDP-43 nuclear retention, while polar residues downregulate these processes. An integrative approach combining analytical ultracentrifugation, NMR-derived contacts, AlphaFold2-Multimer modeling, and all-atom molecular dynamics simulations together suggest that TDP-43 CR forms dynamic, multimeric helical assemblies stabilized by a methionine-rich core with specific contributions from a tryptophan/leucine pair. These structures show how ALS-associated mutations disrupt TDP-43 function and provide pharmacologically targetable structures to prevent its conversion into pathogenic β-sheet aggregates.

TAR DNA-binding protein of 43 kDa (TDP-43) is an RNA-binding protein engaged in diverse cellular functions across both the nucleus and cytoplasm, including RNA transcription, splicing, and transport[1,2]. Many of TDP-43's functions are linked to dynamic, intracellular membraneless organelles formed via phase separation[3-6]. Furthermore, TDP-43 is identified as a major component of ubiquitinated neuronal inclusions in the cytoplasm, a pathological hallmark of amyotrophic lateral sclerosis (ALS), frontotemporal dementia (FTD), and limbic-predominant age-related TDP-43 encephalopathy (LATE), a condition with clinical presentation overlapping with Alzheimer's disease[7-11]. TDP-43 consists of a folded N-terminal domain (NTD) responsible for dynamic homo-oligomerization[12,13], two RNA recognition motifs (RRMs) that together bind UG-rich RNA sequences[14,15], and an intrinsically disordered C-terminal domain (CTD) characterized by a

[1]Artie McFerrin Department of Chemical Engineering, Texas A&M University, College Station, TX, USA. [2]Department of Molecular Biology, Cell Biology & Biochemistry, Brown University, Providence, RI, USA. [3]Edward Doisy Department of Biochemistry and Molecular Biology, Saint Louis University School of Medicine, St. Louis, MO, USA. [4]Laboratory of Molecular Biology, National Institute of Diabetes, Digestive and Kidney Diseases, National Institutes of Health, Bethesda, MD, USA. [5]Department of Chemistry, Texas A&M University, College Station, TX, USA. [6]Interdisciplinary Graduate Program in Genetics and Genomics, Texas A&M University, College Station, TX, USA. [7]These authors contributed equally: Azamat Rizuan, Jayakrishna Shenoy. ✉e-mail: jeetain@tamu.edu; nicolas_fawzi@brown.edu

low-complexity (LC) sequence[16,17]. The aggregation-prone CTD is a hotspot for ~90% of TDP-43's ALS-associated mutations[18,19], and proteolytically cleaved C-terminal fragments are enriched in pathological inclusions[7], underscoring its importance in TDP-43 dysfunction. In biochemical assays, the CTD is sufficient for phase separation in the presence of physiological salt concentrations or RNA[16]. Importantly, TDP-43 CTD condensation is promoted by the hydrophobic conserved region (CR, spanning residues 319–341), working in concert with the adjacent IDRs[5,20].

The CR in CTD is uniquely well-conserved (100% identity) throughout vertebrates compared to the flanking IDRs[16,21] (Fig. 1A). Previous work from our laboratories[16] and others[22] show that the CR adopts transient α-helical structure under physiological conditions. Furthermore, the CR mediates intermolecular helix-helix contacts that are essential for liquid-like phase separation of CTD in vitro[16], phase separation of TDP-43 reporter constructs in cells[21], splicing[5], 3' polyadenylation function[3], and TDP-43 RNA-level autoregulation[3,6], as well as the cytosolic TDP-43 aggregation under oxidative stress[23]. Correspondence between the impact of mutations on TDP-43 function in cells and its phase separation in cellular overexpression and in biochemical assays suggests that even if TDP-43 does not form micronsized membranelles organelles in cells, the same interactions that mediate phase separation drive cellular function[3]. The disruption of functional interactions by ALS-associated mutations or by engineered helix-disrupting variants within the CR can perturb TDP-43 phase separation[5,16] and lead to aggregation via structural conversion of the CR into β-sheet aggregates in disease[9,10,23,24]. Conversely, helical CR self-assembly can be enhanced by designed CR variants, enabling control over the function and material properties of phase-separated droplets[5]. Using NMR experiments combined with computer simulations, we previously observed that α-helicity within the CR is enhanced during helix-helix contact, leading to significant structural changes[5].

Most importantly, the structural details of TDP-43 CR interactions and hence the reasons for the 100% conservation have proved elusive. It remains unclear if the CR forms dimers, higher-order structures, or an ensemble of heterogenous multimeric states. Although definitive evidence has yet to emerge, the presence of helix-mediated higherorder multimers larger than dimers has been suggested based on several lines of evidence including NMR diffusion data as a function of CTD concentration[5] and elevated NMR relaxation values extracted from chemical exchange experiments[16]. However, the molecular structure of these self-assembled species remains unresolved due to the challenges of rapid aggregation of CTD at higher concentrations needed for structure determination and dynamic interconversion of weak affinity states. Determining structural models of TDP-43 CR assembly therefore necessitates an integrative approach that combines artificial intelligence (AI)-based modeling, computer simulations, and constraints from multiple complementary experimental methods.

The emergence of AlphaFold2 (AF2) has revolutionized the field of structural biology, offering accurate predictions of protein 3D structures[25]. Though best known for highly accurate determination of protein structures solely from their primary sequences, a recent extension of AF2, AlphaFold2-Multimer (AF2-Multimer), has marked a significant step forward in the structure prediction for both homomeric and heteromeric protein complexes[26]. However, the structural training set and co-evolutionary sequence data for predicting multimerization data is much more limited, especially for transient, weaklybinding, dynamic assemblies that are not extensively found in the Protein Data Bank (PDB). Hence, it is not clear if AF2-Multimer can generate reliable predictions of dynamic multimeric assemblies or discriminate between potential structures. Instead, we propose that the structural outputs from AF2-Multimer can serve as starting configurations for molecular dynamics (MD) simulations, providing insights into the conformation and dynamics of multimeric states. Recent advancements in atomistic force fields[27,28], combined with GPU-

accelerated MD engines[29,30], have significantly improved the accuracy and efficiency of exploring the conformational landscapes and interactions of intrinsically disordered proteins (IDPs) and their assemblies[23,31–35]. MD simulations can offer feedback on the stability of multimeric assemblies generated by AF2-Multimer, serving as a valuable complement to biophysical experiments and offering atomic-level insights that are not currently achievable through experimental or AIbased tools alone. These approaches provide workable structural hypotheses that can guide experimental validation[36]. Together, AF2 and MD simulations have become indispensable tools in structural characterization, as demonstrated by their increasing application in recent studies[37–42], significantly accelerating progress in structural biology[36].

Here, we employed an integrative modeling strategy based on extensive experimental characterization using NMR spectroscopy and sedimentation velocity analytical ultracentrifugation (SV-AUC), coupled with state-of-the-art computational methods such as AF2-Multimer[25,26] and all-atom molecular dynamics (AAMD) simulations, to probe the atomic structural details of TDP-43 CR helix-mediated assembly. Importantly, we establish atomic-level structural models of TDP-43 CR multimeric assemblies that provide critical information required to understand the early steps in functional TDP-43 CTD assembly. These structural insights constitute a significant advancement in our comprehension of the molecular pathways underlying TDP-43 phase separation and function, thereby opening avenues for studying and preventing the emergence of its pathogenicity.

## Results

### The native state of TDP-43 CR at physiological conditions is α-helical

Previously, we provided direct structural evidence for the presence of α-helical structure within the CR of TDP-43 and its stabilization via helix-helix assembly using solution NMR spectroscopy[5,16]. The experiments were performed at a slightly acidic pH (6.1) and without salt to attain optimal NMR spectra and prevent phase separation. However, recent work using hydrogen-bond disrupting variants suggested (without direct structural evidence) that the CR of TDP-43 adopts a β-sheet conformation at neutral pH and physiological salt and that the α-helical structure previously observed by us and others is an artifact of non-physiological conditions[43]. To resolve this important issue, we conducted solution NMR spectroscopic measurements at both our previous conditions (slightly acidic pH 6.1 with no salt) and at physiological conditions (neutral pH 7.0 with 150 mM NaCl) and low protein concentration to prevent phase separation.

The largest differences in the spectra come from the loss of several resonances for serine and glycine residues in the IDR regions, (Fig. 1B), due to rapid exchange of the amide hydrogen with water hydrogen, which is elevated for these residue types and is faster at higher pH[44]. Focusing on the overlay of $^1H$-$^{15}N$ HSQC spectra for residues within the CR (Fig. 1C), we observed only minimal changes in chemical shifts, suggesting no significant change in the overall structure of this region. To further probe structural changes, we compared the differences between experimental $C_\alpha$ chemical shifts and the random coil shifts, $\Delta\delta C_\alpha$ (Fig. 1D)[45,46], and found highly similar $\Delta\delta C_\alpha$ values in both cases. Positive $\Delta\delta C_\alpha$ values within the CR reveal its helical character at neutral pH and physiological salt conditions; in fact, helicity is slightly enhanced at these conditions. Enhancement in helicity could also lead to enhanced helical interaction and cooperate with reduced electrostatic repulsion at higher pH and higher ionic concentration that drive TDP-43 phase separation as well as aggregation as previously observed[47,48]. Furthermore, our previous computer simulations, performed under physiological salt and pH conditions using a modern state-of-the-art force field for IDPs[27], also confirmed the presence of a transient helix spanning residues 321–330 in the monomeric CTD ensemble and its stabilization by intermolecular

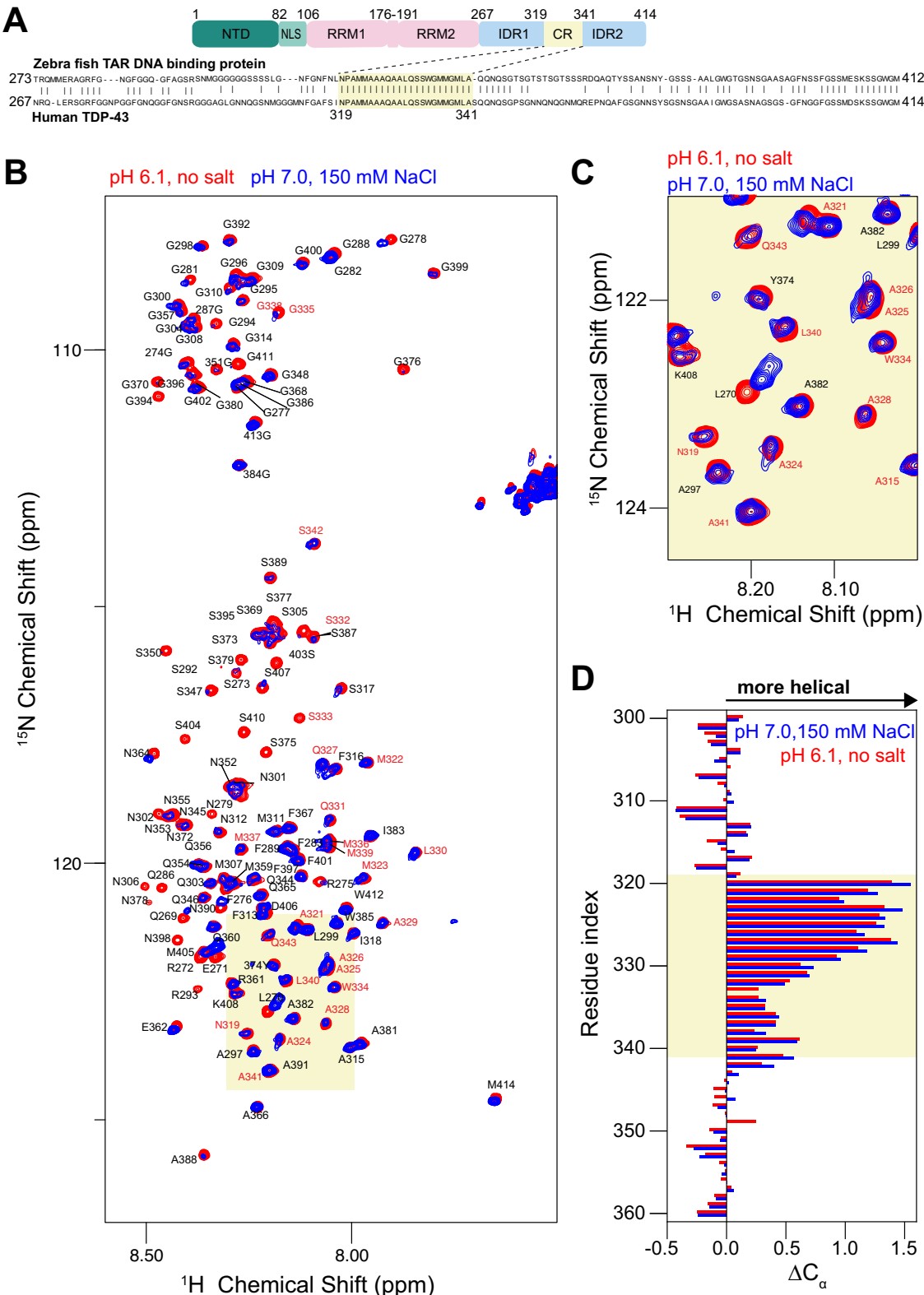

**Fig. 1 | The conserved region (CR) of TDP-43 CTD adopts α-helical structure at native buffer conditions. A** Domain composition of TDP-43, with CR residues highlighted. Pairwise alignment of the CTD of human TDP-43 (UniProt id: Q13148) and zebrafish TAR DNA binding protein (UniProt id: Q802C7) shows CR residues are evolutionarily conserved in vertebrates. Identical residues are indicated with vertical bars. **B** A comparison of the ¹H-¹⁵N HSQC of WT CTD at previous conditions optimal for NMR (20 mM MES, pH 6.1, red) and at neutral pH with physiologically relevant monovalent salt concentration (20 mM HEPES, 150 mM NaCl buffer pH 7, blue) shows very similar spectral fingerprint, especially in the CR region (red text). **C** An excerpt of the ¹H-¹⁵N HSQC shows resonances from the CR are nearly identical at both conditions. **D** Secondary $C_\alpha$ chemical shifts comparing the observed chemical shift and the value for a random coil reference, $\Delta\delta C_\alpha$, at pH 6.1 without salt and pH 7.0 with 150 mM NaCl, showing that the helicity is observed in both conditions. Positive $\Delta C\alpha$ values are indicative of α-helical secondary structure. Source data are provided as a Source Data file. Abbreviations: NTD N-terminal domain, NLS Nuclear localization signal, RRM RNA Recognition Motif, IDR Intrinsically disordered region.

interactions[5,16,20]. These observations are also consistent with solid-state NMR experiments on in vitro aged liquid droplets[49] and fibrillar aggregates[50,51], which showed the presence of helical structures in TDP-43 CR even after transition to fibrils[51]. Taken together, our direct structural measurements using NMR unequivocally show the presence of native α-helical structure within TDP-43 CR at physiological conditions.

## Alanine-scanning mutagenesis within TDP-43 CR reveal key residues in phase separation

TDP-43 CR promotes the formation of phase-separated CTD droplets[16]. Disease-associated and designed mutations that reduce helical stability or disrupt intermolecular helix-helix interactions decrease CTD phase separation, while variants enhancing helicity promote it[5]. By simultaneous substitution of all five CR methionines to alanine, we recently found that methionine residues within CR are crucial for intermolecular helix-helix contacts mediating CTD phase separation[20]. To comprehensively investigate the role of each CR position, we created TDP-43 CTD variants substituting each non-alanine position with alanine to preserve helicity while potentially disrupting CR:CR interactions[20]. We measured the saturation concentration ($c_{sat}$), the protein concentration threshold for phase separation, using droplet sedimentation assays for all variants. Microscopy was used to qualitatively evaluate phase separation and whether droplets remained dynamic and liquid-like or transitioned to static aggregates.

WT CTD does not undergo phase separation in the absence of salt but forms liquid-like droplets with increasing salt concentrations, having a $c_{sat}$ ~ 15 μM at 150 mM NaCl (Fig. 2A, B, Supplementary Fig. 1). Highlighting the essential role of a single tryptophan in promoting CTD condensation, W334A shows the most profound effect on phase separation, resulting in no droplets at the studied conditions (Fig. 2B, Supplementary Fig. 1). Similarly, $c_{sat}$ was increased (decreased phase separation) three-fold by F316A which lies directly adjacent to the helical CR, whereas the effect of F313A in the region preceding the helical CR was less, showing $c_{sat}$ twice as high as that of WT (Fig. 2A).

Expanding from our previous results for the 5 M→A variant within the CR that collectively impaired CTD phase separation, each single methionine substitution within CR (M322A, M323A, M336A, M337A, or M339A) also reduce phase separation with approximately two-fold higher $c_{sat}$ than WT (Fig. 2A). Similarly, I318A and L340A result in $c_{sat}$ values similar to single methionine to alanine substitutions, suggesting that bulky aliphatic residues contribute similarly to phase separation. Somewhat surprisingly, the substitution of L330A emerges as a major outlier compared to other aliphatic residues, resulting in significant destabilization of the condensate ($c_{sat}$ ~ 60 μM). This effect is much higher than L340A near the C-terminal end of the CR, suggesting a special role of L330 in self-assembly.

Next, we consider the polar residues which are just as conserved as the hydrophobic residues in the CR (Fig. 2A). Unlike substitutions to alanine at hydrophobic residue positions which decrease phase separation, alanine mutations at serine positions enhance phase separation. S333A shows the most enhancement in phase separation, while the adjacent S332A and the more distant S342A show modest enhancement, and S317A shows little change in $c_{sat}$. Glutamine to alanine mutations (Q327A, Q331A, Q343A) display the greatest enhancement of phase separation with $c_{sat}$ approximately three-fold lower than WT $c_{sat}$. Microscopy of these variants after phase separation confirms the formation of droplets. For Q327A and S333A, the droplets appear slightly less round, suggesting these variants may form more viscous or viscoelastic assemblies in addition to enhancing phase separation, while N319A robustly forms morphologically irregularly shaped assemblies (Fig. 2B). Similar to how glycine-to-alanine substitutions (G335A, G338A) enhance CTD self-association and phase separation by stabilizing the helicity of the monomeric CR[5], the enhancement in phase separation for polar-to-alanine substitutions reinforce a critical

role of the CR helix, especially since many of these substitutions are known to discourage phase separation in non-helical contexts[52,53]. Taken together, in vitro phase separation experiments highlight the importance of the entire CR for phase separation, with phase separation promotion in particular by L330 and W334.

## TDP-43 CR helicity and helix-mediated interactions are linked to CTD phase separation

Above we probed the contribution of each CR position to phase separation. Next, we sought to test if and how these effects of alanine substitutions on phase separation are related to helix-mediated interactions probed by NMR. We evaluated the impact of alanine mutations on the helical assembly by monitoring NMR chemical shift perturbations (CSPs) as a function of protein concentration at pH 6.1 and without salt where TDP-43 CTD does not readily phase separate. The relative magnitude of helix-mediated self-assembly for each variant was quantified by the $^{15}N$ CSP ($\Delta\delta^{15}N$) for the A328 resonance at 90 μM (where intermolecular interactions are formed) and 20 μM (monomeric reference state), normalized to WT CTD values ($CSP_{norm}$). Q327A, S332A, and S333A were excluded due to broadening of CR signals beyond detection in HSQC experiments at the higher concentration (90 μM), unlike Q331A, which remained detectable despite also leading to enhanced self-assembly (Fig. 2A). This difference may result from variations in monomer-multimer exchange kinetics, binding and dissociation rates, and conformational changes within the assemble state. W334A and N319A were also excluded as W334A does not phase separate at these conditions, and N319A results in irregularly shaped aggregates, making the apparent $c_{sat}$ unreliable. Remarkably, we observed a strong correlation between $CSP_{norm}$ and $c_{sat}$ for alanine mutants, with a Pearson correlation coefficient ($r$) of −0.94 (Fig. 2C), i.e., higher $CSP_{norm}$ values were observed for alanine variants enhancing phase separation and lower values for those reducing phase separation. We note that these CR variants may contribute to disruption in helix-helix interactions either by destabilizing the helicity of the monomeric form (which is significantly populated from 320 to 330) as previously seen for some ALS-associated, engineered, and phospho-mimetic variants in the 321–333 region[16,54] or by disrupting the helix-helix interface without changing the helicity of the monomer as suggested for other ALS-associated and engineered variants at M337, or a combination of these factors. Previously, we found that simultaneous substitution of all five CR methionines to alanine does not suppress helicity but does disrupt phase separation[20]. However, the opposing effects on CTD phase separation of alanine variants at two neighboring residues S333 and W334 present an interesting case. NMR spectra of monomeric W334A reveal features suggesting the variant reduces helicity, highlighting W334's unique role (Supplementary Fig. 2) and suggesting that this variant may suppress phase separation by decreasing helicity and/or weakening inter-helix interactions. Although alanine can moderately enhance local helicity compared to serine[5], S333A does not markedly alter the helical signatures across the monomeric CR, implying that its enhancement of phase separation primarily driven by strengthened inter-helical interactions or by local helical stabilization (Supplementary Fig. 2). Although the detailed structural mechanisms for each of the ~20 variants studied here likely have multiple structural contributions, the focus here is that the correlation observed between biochemical and biophysical behaviors suggest that helical assemblies that drive phase separation in physiological salt conditions are structurally related to the helix-mediated transient interactions that stabilize CR contacts in salt-free conditions visible by NMR.

## Structural ensemble for TDP-43 CTD dimer

We next sought to probe the contacts formed by CR dimers as the first step in higher-order multimerization. We used AF2-Multimer (see Methods) to predict structures of a TDP-43 subregion containing the

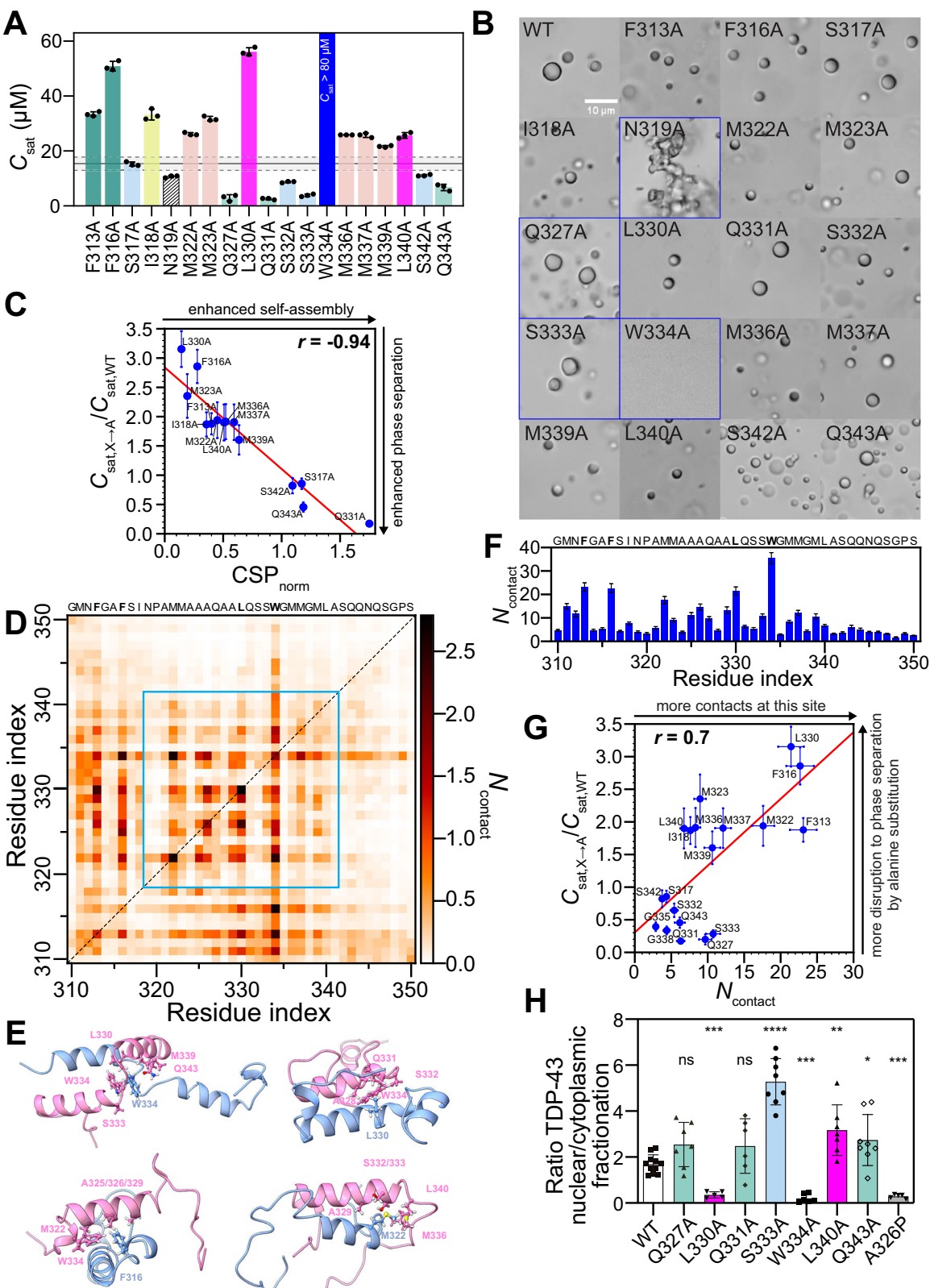

CR (residues 310–350) (Supplementary Fig. 3A). This length of protein fragment, from our previous work on TDP-43 CTD[5,16], was selected as it is more computationally efficient for MD simulations and better suited for determination of the correct bound conformation using AlphaFold-Multimer[55,56]. Importantly, AF2-Multimer predicts dimer states with high structural heterogeneity and low model confidence (Supplementary Fig. 3B). We reasoned that molecular simulation

starting from predicted structures could help define the structural ensemble and provide insight as to which structures may in fact be stable or populated in solution. To this end, we conducted extensive, unbiased AAMD simulations (18 independent trajectories, aggregate time ~100 µs) starting from the TDP-43$_{310-350}$ dimer models obtained from AF2-Multimer (Supplementary Fig. 3A, B). Importantly, all dimer starting structures of TDP-43$_{310-350}$ showed lack of stability and

**Fig. 2 | Site-specific alanine substitutions within the TDP-43 CR reveal key residues in TDP-43 phase separation, helix assembly, and function.**
**A** Saturation concentration ($c_{sat}$) at 150 mM NaCl for WT CTD (gray box with dashed lines covering the mean ± standard deviation from independent experiments shown in Supplementary Fig. 1) and single alanine substitution variants within CR and its adjacent residues. Data obtained from $n$ = 3 technical replicates, mean ± SD. **B** DIC micrographs of CTD variants at 80 μM in 150 mM NaCl. Scale bar, 10 μm. All variants form round liquid-like droplets except N319A (morphologically irregular), Q327A/S333A (slightly less round), and W334A (no droplets), highlighted with a blue box. **C** $^{15}$N chemical shift perturbations (CSP) (Δδ$^{15}$N) and $c_{sat}$ (mean ± SD) for the single alanine substitution variants exhibit a strong correlation (Pearson correlation coefficient, $r$ = −0.94). Both axes are normalized by WT CTD values. **D** Pairwise intermolecular contact map from two-chain AAMD simulations of TDP-43$_{310-350}$. Contacts averaged over 18 trajectories; CR contacts highlighted (cyan). **E** Snapshots from two-chain AAMD simulations show heterogeneous helix-helix

contacts, involving a multitude of residues. **F** Total number of contacts ($N_{contact}$) for each residue position (mean ± SEM from 18 replicas) computed by one-dimensional summation of pairwise contacts from (**D**). **G** $N_{contact}$ (mean ± SEM) in **F** show a good correlation (Pearson correlation coefficient, $r$ = 0.7) with the $c_{sat}$ (mean ± SD) of single alanine substitution variants in **A** (including G335A and G338A $c_{sat}$ values from our previous work[5]), normalized to WT. **H** TDP-43 nuclear retention was quantified by measuring the ratio of HA-TDP-43 in nuclear/cytoplasmic fractionation experiments. Nuclear/cytoplasmic fractionation ratio of WT TDP-43 and its single alanine substitution variants within CR in HEK293$^{HA-TDP-43}$ were measured by immunoblotting. Nuclear/cytoplasmic fractionation ratio of helix-disrupting mutant, A326P is included as a control of reduced nuclear retention. Statistical analysis was assessed using a two-sided Mann–Whitney U test. ****$p$ < 0.0001, ***$p$ < 0.0005, **$p$ = 0.001, *$p$ = 0.01, ns non-significant. Mean ± SD is shown for ≥5 independent replicates. Source data are provided as a Source Data file.

dynamic behavior, with dimers able to dissociate and re-associate within a microsecond simulation period (Supplementary Fig. 3C) with highly fluctuating interhelical contacts (Supplementary Fig. 3D).

We then analyzed the structural diversity within the dimer ensemble, finding >80% helicity for residues 320–331 and ~30% for residues 339–343, with a ~ 20% increase in helicity for residues 320–331 compared to the monomeric ensemble (Supplementary Fig. 3E), aligning with previous simulation results demonstrating helix stabilization upon dimerization[5]. Secondary structure maps highlighted diverse helical conformations with different helix lengths and positions (Fig. S3F), with both parallel and antiparallel binding modes (Supplementary Fig. 3F). Using the simulated dimer ensemble, we characterized the intermolecular contacts stabilizing the helix-mediated dimerization (Fig. 2D, E), which showed significant differences compared to the initial contacts (Supplementary Fig. 3G), reflecting dynamic structural interconversions.

The total number of per-residue contacts revealed that W334, F313, F316, and L330 form the most contacts (Fig. 2F), correlating with their role in stabilizing CTD condensates, whereas polar residues form fewer contacts, consistent with enhanced phase separation upon polar-to-alanine substitutions. We found a strong correlation (Pearson $r$ = 0.7) between the per-residue intermolecular contacts from dimer simulations and experimental phase separation ($c_{sat}$) from alanine mutagenesis across CR (Fig. 2G). These data support the view that residues stabilizing CR helix-helix contacts can contribute to the phase separation of CTD but that the dimer is not likely to populate a single stable structure.

## TDP-43 CR helix-mediated assembly affects TDP-43 nuclear retention and growth restriction in cells

We then probed the relevance of the contacts probed by biophysical/biochemical experiments and simulations by testing the impact of a set of TDP-43 CR alanine variants in cells. TDP-43 is a predominantly localized in the nucleus, and loss of its nuclear function is a major contributor to RNA processing defects associated with ALS[57]. While TDP-43 nuclear import is mostly an active mechanism mediated by a nuclear localization signal and importins α/β (karyopherins α/β)[58,59], TDP-43 cytoplasmic exit is a passive and size-dependent process[60–62]. We previously showed that RNA binding, condensate formation and multivalent assembly through NTD and CTD self-interactions are essential for TDP-43 function and for its nuclear localization, as demonstrated by nuclear-cytoplasmic fractionation and quantitative immunofluorescence[6,62]. These complexes promote nuclear retention by increasing the size of macromolecular TDP-43 assemblies[62]. We examined the impact of key alanine variants that alter helical interactions and phase separation on TDP-43 nuclear retention by comparing the nuclear-cytoplasmic distribution of hemagglutinin (HA)-tagged full-length TDP-43 variants stably expressed in isogenic cell lines (HEK293$^{HA-TDP-43}$)[6,62]. Importantly, to minimize undesired and

heterogenous overexpression, these cell lines express a single copy of the HA-TDP-43 transgene upon tetracycline induction. We investigated variants that either enhance (Q327A, Q331A, S333A, Q343A), moderately reduce (L340A), or impair (W334A and L330A) CTD condensation. TDP-43 levels in the nuclear and cytoplasmic fractions were quantified via immunoblotting (Fig. 2H, Supplementary Fig. 4A, B). W334A and L330A dramatically reduce TDP-43 nuclear retention by approximately 90% and 80%, respectively, indicating strong nuclear loss upon decreasing phase separation in the context of full-length TDP-43. This aligns with the effects of the deletion of CR[62] or the helix-disrupting mutant, A326P, which also significantly diminishes TDP-43 nuclear retention (Fig. 2H). Conversely, variants that enhance helix-mediated self-assembly and phase separation (Q327A, Q331A, S333A, Q343A) promote nuclear retention, with S333A showing a 3-fold increase in nuclear retention compared to WT. Single glutamine-to-alanine substitutions generally increase the mean nuclear retention ratio by about 50%, compared to WT. The results for Q327A and Q331A show a trend toward enhanced nuclear retention, however, they were not statistically significant. Unexpectedly, L340A slightly increased TDP-43 nuclear retention, contrary to our expectations based on its modestly decreased phase separation (small increase of $c_{sat}$). We also performed immunofluorescence microscopy to compare the subcellular localization of HA-tagged TDP-43 variants in intact cells (Supplementary Fig. 4C, D). Consistent with our fractionation assays and previous report[62], A326P significantly decreases the nuclear localization. W334A and L330A show a trend toward lower nuclear localization, while S333A shows a moderate increase in TDP-43 nuclear localization compared to WT; however, these differences do not reach statistical significance (Supplementary Fig. 4D). Differences in the extent of nuclear localization loss caused by mutations disrupting phase separation, more pronounced in nuclear-cytoplasmic fractionation than in immunofluorescence, may reflect leakage of HA-TDP-43 during biochemical nuclear isolation. Our previous studies of macromolecular complexes including exogenous HA-TDP-43 in HEK293$^{HA-TDP-43}$ cells suggest that endogenous TDP-43 does not significantly contribute to increased nuclear retention. WT HA-TDP-43 expression in these cells decreases endogenous TDP-43 levels by approximately 60% due to autoregulation, which serves as a negative feedback loop controlling protein synthesis[6]. WT HA-TDP-43 exhibits similar nuclear retention and macromolecular size distribution as endogenous TDP-43[62]. In contrast, expression of A326P does not significantly reduce endogenous TDP-43 levels due to defects in autoregulation[3]. Despite the presence of endogenous TDP-43, A326P shows lower nuclear retention as well as decreased formation of large macromolecular complexes, compared to WT HA-TDP-43[62]. Overall, these findings indicate that the structures populated upon phase separation in vitro as detected by NMR are important for cellular TDP-43 assembly.

We next explored the impact of helix-mediated assembly of full-length human TDP-43 expressed in yeast on growth restriction using

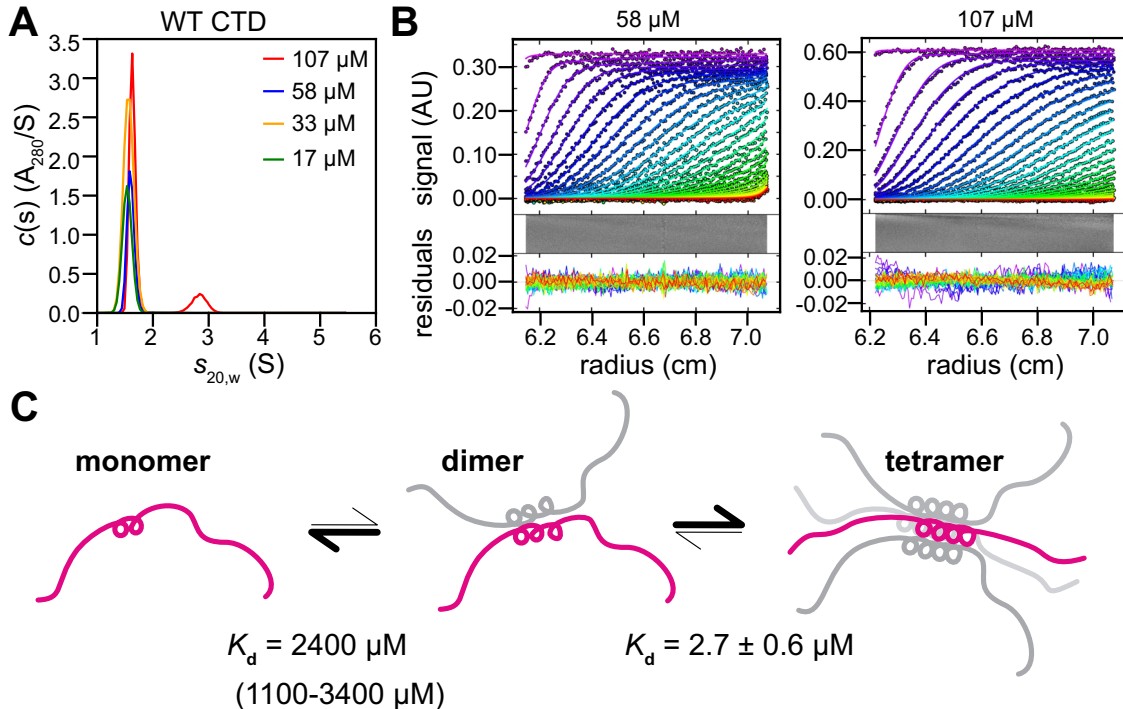

**Fig. 3 | TDP-43 CTD forms multimers. A** Sedimentation velocity analytical ultra-centrifugation (SV-AUC) of wild-type C-terminal domain (WT CTD) as a function of protein concentration. SV-AUC data were analyzed as a continuous $c(s)$ distribution of sedimenting species suggesting the presence of species larger than dimer. Data were collected in 12 mm cells, except for WT CTD at 58 and 107 μM, where 3 mm pathlength cells were used. **B** Absorbance sedimentation data (in absorbance units, AU) collected for WT CTD at (left) 58 μM (3 mm pathlength cell) and (right) 107 μM (3 mm pathlength cell) were analyzed globally in terms of a monomer-dimer-tetramer self-association using Lamm equation modeling, along with trace amounts of an aggregate. The analysis, carried out in SEDPHAT[87], returns a monomer-dimer dissociation constant, $K_{d,mon-dim}$ of 2.4 mM (confidence interval

representing one standard deviation of 1.1–3.4 mM) and a dimer–tetramer $K_{d,dim-tet}$ of 2.7 ± 0.6 μM. Data were plotted in GUSSI[88] and for clarity only every third scan and every third experimental data point are shown. Best-fits are represented by a solid line through the experimental points. A bitmap representation of the residuals, together with the combined residuals, are shown below each plot. We note that the aggregation-prone nature of TDP-43 CTD prevented saturation of the multimeric form and hence the data suggest but cannot conclusively demonstrate that the most stable multimeric form is the tetramer. **C** Schematic of the monomer-dimer-tetramer equilibrium and dissociation constants. Source data are provided as a Source Data file.

deep mutagenesis data from the Lehner lab, which includes over 1200 single mutations in TDP-43 CTD[63]. Mutations within the CR showed a stronger influence on yeast growth restriction compared to mutations in the flanking IDRs, underscoring the critical role of the CR in TDP-43 function and toxicity[63]. Notably, mutations at W334 increased toxicity, except for L and M substitutions, correlating with W334's essential role in CTD self-assembly and nuclear retention function. A strong correlation (Pearson $r = 0.79$; Spearman $r = 0.86$) was observed between in vitro phase separation propensity and yeast toxicity (Supplementary Fig. 5). Alanine mutations that enhance phase separation reduce toxicity in yeast cells, whereas ones that decrease phase separation do the opposite. This further supports the cellular significance of helical self-assembly, as mutants enhancing cellular toxicity also enhanced accumulation in the cytoplasm at the nuclear periphery[63]. We propose that helix-mediated TDP-43 self-assembly aids nuclear retention, mitigating improper localization and toxicity.

## TDP-43 CTD forms multimers

The importance of the CR in TDP-43 CTD helical assembly has been established, yet the stoichiometry of assembly has not been probed. Here, we used sedimentation velocity analytical ultracentrifugation (SV-AUC), an experimental tool that provides information on macromolecular size in solution, to probe the size of TDP-43 CTD multimers formed in solution without salt, where TDP-43 CTD does not readily phase separate. At the lowest concentrations (17 μM) where the protein remains monomeric, the best-fit model shows a single species with sedimentation coefficient (corrected to standard conditions, water at

20 °C), $s_{20,w}$, of 1.53 S with an estimated mass of 14.0 kDa, consistent with a monomer (Fig. 3A). As the concentration is raised (33 μM and 58 μM), the apparent sedimentation coefficient for the species increases slightly, consistent with weak self-association (Fig. 3A). At the highest concentration studied, 107 μM (a value obtained directly from the AUC data by integration of the absorbance $c(s)$ profile), the best fit model is notably different despite less than a two-fold increase in concentration, showing an additional faster-sedimenting species (Fig. 3A). Although a dimer would be expected to have an s-value approximately 1.58x that of the monomer, the s-value for this species is approximately 2x that of the monomer, suggestive of a weak CTD self-association into multiple higher-order species, consistent with previous NMR data[5]. To properly account for the contribution of interchanging species to the sedimentation profiles, direct Lamm equation modeling was conducted. "Two state" monomer-dimer or monomer-trimer models and "three state" monomer-dimer-trimer models resulted in poorer fits and/or physically unreasonable sedimentation coefficients (Supplementary Table 1). In contrast, excellent fits were found for a monomer-dimer-tetramer model with sedimentation coefficients of 1.54 S, 2.43 S, and 3.94 S that scale as expected for a CTD monomer, dimer, and tetramer (Fig. 3B). Using the data from the two highest concentrations simultaneously, best fits yielded a monomer-dimer dissociation constant $K_{d,mon-dim}$ of 2.4 mM (confidence interval corresponding to one standard deviation: 1.1–3.4 mM) and a dimer–tetramer $K_{d,dim-tet}$ of 2.7 ± 0.6 μM, suggesting that the dimer is only a weakly-binding intermediate, consistent with our simulated dimer ensemble, but that it cooperatively forms a tetramer, with 1000-

fold tighter affinity for the dimer-tetramer transition than for monomer-dimer (Fig. 3C). At the 107 μM condition, 6.2 μM is predicted to be in the dimer form and 14.2 μM in the tetramer form. Additionally, a two-state monomer-tetramer model results in similar fit quality and shows comparable effective binding equilibria, as seen in the monomer-dimer-tetramer model (i.e., $c_{eq}$, the concentration for equal population of monomer and tetramer is in the range of 250–300 μM for both models). Both models are also consistent with previous analyses at slightly different conditions using NMR relaxation dispersion of TDP-43 CTD exchange between monomeric and multimeric structures[16] and it remains difficult to conclusively distinguish between models where the dimer is a very weakly populated intermediate (monomer-dimer-tetramer model) or a transiently populated transition state (monomer-tetramer model). We caution that because samples at higher concentrations lead to aggregation, we cannot approach saturation of the multimeric forms. Hence, while the SV-AUC data for WT CTD support the presence of multimers that are likely tetrameric, we cannot conclusively establish the stoichiometry by AUC of WT CTD alone. Therefore, below we employ additional AUC analyses on samples designed to boost self-assembly as well as decrease aggregation, and complementary experimental and computational tools to test this picture further.

## NMR shows key contacts for TDP-43 higher-order self-assembly

Given the transient, dynamic nature of CR interactions and their small population, structural characterization of the CR helix-helix interaction is not amenable to x-ray crystallography or electron cryo-microscopy (cryoEM). Hence, we next sought structural information for the assembled CR region of TDP-43 CTD using nuclear Overhauser enhancement (NOE) NMR. However, several features make CR multimerization an extremely challenging target. First, NOE experiments are often conducted at 1 mM but TDP-43 CTD samples at higher concentrations (≥100 μM) are not stable and convert to aggregates, preventing long NMR experiments. Second, the multimer is not fully populated at achievable conditions, reducing NOEs from intermolecular contacts. Third, exchange between the monomeric and multimeric conformations broadens the resonances from the CR[16]. Fourth, the CR itself is aggregation prone, as short peptides (residues 311–360) containing the CR readily form fibrils[64], suggesting the flanking IDR regions contribute to solubility. Hence, we explored mutants of TDP-43 CTD IDRs to further enhance solubility while keeping the CR and adjacent regions (310–350) unaltered to preserve CR multimerization. We recently demonstrated that an aggregation-reducing variant of TDP-43 CTD that replaces phenylalanines in the IDRs with alanine[65] (TDP-43 CTD 6 F → A) retains phase separation and CR helix-helix interaction at higher protein concentrations[20]. We conducted filtered, edited NOE-HSQC experiments on samples of 200 μM (1:1 mix of unlabeled and $^{13}$C/$^{15}$N proteins) TDP-43 CTD 6 F → A to selectively probe intermolecular contacts[66]. Experiments were performed at 12 °C to enhance multimer population and NOEs by slowing molecular motion as well as to extend sample stability.

Two-dimensional NOE strips for NOEs to $^{13}$C-attached hydrogens suggest prevalent interactions involving aliphatic M/L/I/A methyl within the helical region (Fig. 4A), much more than control experiments using 100% labeled protein to account for any intramolecular artifacts from incomplete labeling. NOEs are large for I318 but smaller for A315, consistent with structure forming primarily in the conserved region (Fig. 4B). NOEs to $^{15}$N attached hydrogens provide additional intermolecular interactions with residue-by-residue resolution, showing backbone and side chain amide hydrogens (NH and NH$_2$) interact with methyl groups, aromatic and polar groups, and H$_\alpha$ positions (Fig. 4C). Importantly, NOEs to backbone NH hydrogens are observed for several positions from A325 to L340, suggesting interactions across the entire CR (Fig. 4C). Additionally, W334 sidechain positions interact with aliphatic methyl, aromatic, backbone amide, and H$_\alpha$ positions

(Fig. 4C, D). Contacts involving the phenyl group of phenylalanine (F313/316) with aliphatic methyl groups are also present (Fig. 4D), suggesting these residues contribute to multimerization.

Collectively, NOE experiments reveal the presence of contacts between many residue types in the TDP-43 CR assembly, prominently involving aliphatic and aromatic residues. However, the repetitive CR sequence (five M, six A, two S, three Q residues), relatively large separation between resolved backbone $^{1}$H/$^{15}$N positions in adjacent helices, and difficulties in obtaining higher resolution/signal-to-noise data due to sample instability make it challenging to derive molecular models directly from these experimental constraints. To complement the structural data from TDP-43 CTD 6 F → A, we also extensively characterized the assembly via AUC and contacts via NMR of a series of cross-linked variants of TDP-43 (See Supplementary Notes 1,2, Supplementary Figs. 6–9). Phosphorylation at the very C-terminal part of TDP-43 CTD (S403, S404, S409, S410) is a major feature of TDP-43 inclusions and placing aspartic acid residue substitutions mimicking phosphorylation at these sites can drive aggregation[67]. In contrast, placing aspartic acid residues at 12 sites across the entire CTD appears to counteract pathological aggregation[68], enhance the liquidity and dynamic nature of TDP-43 condensates, while preserving nuclear import or RNA regulatory functions, similar to what we have observed for poly-phosphomimetic FUS[69]. Similarly, here we made serine to glutamate (S→E) substitutions across the disordered flanking regions of CTD (See Supplementary Table 2), which significantly enhanced TDP-43 solubility while preserving the helical structure in the CR structure as indicated by NMR (Supplementary Fig. 6). We previously showed that adding disulfide cross-links to WT TDP-43 CTD enhances CR interaction[5] and we find here that it enhances multimerization as seen by AUC (Supplementary Fig. 7A), but it also enhances aggregation. However, cross-linked S → E variants remain sufficiently soluble for AUC analysis across a range of concentrations to approach saturation of the assembly (Supplementary Figs. 7, 9). These cross-linked variants populate multimeric assemblies requiring the helical region, as seen by AUC, and show enhanced helicity compared to the uncross-linked forms (Supplementary Figs. 7, 9). Using cross-linked forms created from 1:1 mixture of unlabeled and $^{13}$C/$^{15}$N proteins, we again see a similar pattern of methyl NOEs as observed for TDP-43 CTD 6 F → A (Supplementary Fig. 10), suggesting that CR assembly is similar in both contexts where the TDP-43 CTD solubility is enhanced without altering the CR.

## Structural model of tetrameric helical assemblies of TDP-43 CR

Given the challenges in direct determination of an atomic resolution structure of multimeric TDP-43 CTD using NMR alone, we turned to computer modeling using AF2-Multimer to predict potential multimeric structures (trimer, tetramer, hexamer, octamer). Like for predicted dimers (see above), the model confidence score for all predicted multimeric structures is notably low compared to a control homo-tetrameric coiled-coil sequence (CC-tet)[70] (Fig. 5A). This suggests that either AF2-Multimer cannot identify TDP-43 CR multimers or that the dynamic TDP-43 CR multimers[5] are unlike stable coiled-coil sequences that are more easily predicted. Hence, we employed AAMD simulations (~4–5 μs in duration, or until a significant structural deviation was observed, defined as distance root-mean-square deviation (dRMS) > 1 nm) to directly assess the structural stability of these multimers and to explore whether AAMD could result in refined interhelical contacts that stabilize the assembled states (Fig. 5B).

Interestingly, all trimer (eight), hexamer (five), and octamer (four) models were found to be less stable, exhibiting large structural changes from their initial conformations (Fig. 5B, Supplementary Fig. 11A). Notably, among the tetramer structures, one model (which we named Tet-1) with four-fold symmetry where adjacent helices run antiparallel, exhibited greater stability during the simulation (Fig. 5B). To further test Tet-1's stability, we performed 50 independent AAMD simulations

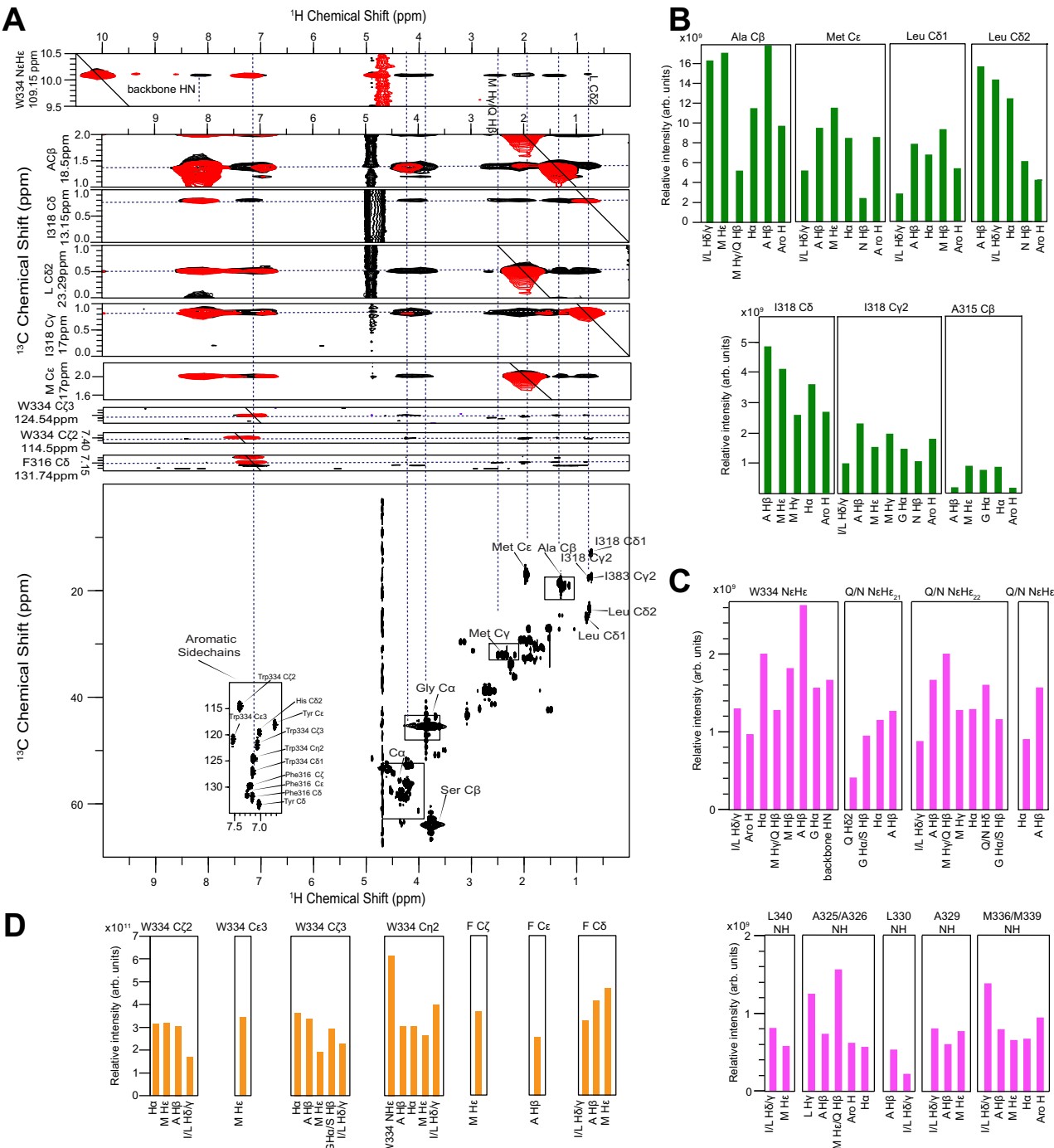

**Fig. 4 | Intermolecular NOEs show key contacts for TDP-43 CR assembly. A** $^{13}$C filtered-edited NOE-HSQC strips of 200 µM 6 F→A (1:1 mix of $^{13}$C/$^{15}$N labeled and unlabeled, black) measured at 12 °C, 600 MHz, 20 mM MES pH 6.1, Transient NOEs (arising from equilibrium exchange between monomeric and multimeric states) are not artifacts as demonstrated by data from control sample (100% $^{13}$C/$^{15}$N sample), red, that do not have extensive methyl NOEs. **B** Quantification of NOEs from $^{13}$C-attached positions of different residues. Intensities were corrected for intramolecular artifacts arising from incomplete labeling by subtraction of 0.5*NOEs measured in a control sample 100% $^{13}$C/$^{15}$N sample). **C** Quantification of NOEs from $^{15}$N-attached positions of different residues. Intensities were corrected for artifacts as above. **D** Quantification of NOEs from $^{13}$C-attached positions of aromatic residues. Intensities were corrected for artifacts as above. Source data are provided as a Source Data file.

(each 100 ns) of Tet-1 along with Tet-2, the second most stable tetramer that has a distinct helical arrangement and, unlike Tet-1, quickly loses its symmetry (Fig. 5B). The dRMS distributions from these simulations further confirmed the higher stability for Tet-1 (Fig. 5C). Collectively, AAMD simulations of AF-predicted multimers reveal a stable tetramer structure, suggesting that dynamic nature of CR self-interactions may adopt stable tetramer configurations, providing additional support for the idea that TDP-43 CTD forms tetramers via the CR suggested as the most likely assembly by AUC.

Intermolecular contact analysis reveals that the core of the Tet-1 is stabilized M322, A326, A329, S333, M336 and L340 residues that form the highest total number of contacts (Fig. 5D, E). The stability is further enhanced by interhelical contacts between adjacent helices involving residues M323, A325, L330, S332, W334, M337 and M339 (Fig. 5D, E).

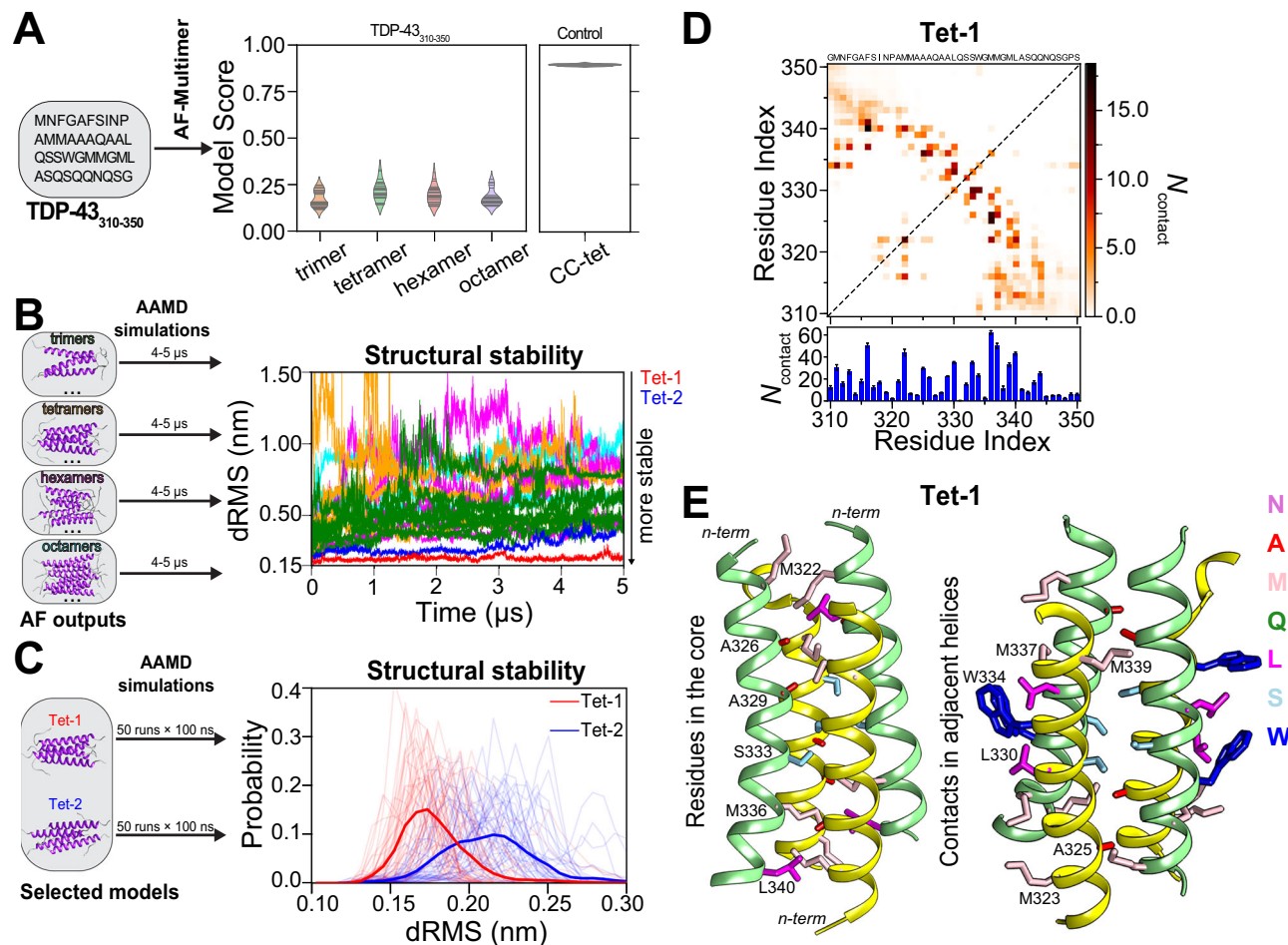

**Fig. 5 | Structural model of the tetrameric helical assembly of TDP-43 CR.**
**A** Potential multimeric structures (trimer, tetramer, hexamer, octamer) of TDP-43$_{310-350}$ fragment are predicted using AF2-Multimer. The model confidence score ($0.8 \times$ ipTM + $0.2 \times$ pTM) of predicted structures for different multimeric states is compared with that of AF2-Multimer predicted tetrameric structures of homo-tetrameric coiled-coil sequence (CC-tet, 4-KE-4[70]), which served as a control. **B** AF2-Multimer predicted multimeric structures are then fed to run all-atom MD (AAMD) simulations (~4−5 µs each) and structural stability was assessed by computing the distance root-mean-square (dRMS) of heavy atoms (residues 320−341, excluding hydrogens) relative to the initial conformations. The time evolution of dRMS values is shown. The dRMS values of the trimers are represented in dark green, tetramers in orange (with Tet-1 and Tet-2 specifically highlighted in red and blue, respectively), hexamers in magenta, and octamers in cyan. (See Supplementary Fig. 11A for more details). **C** dRMS distributions from AAMD simulations (50 independent replicates with 100 ns each) of selected tetramer models (Tet-1 and Tet-2) are

shown. The lighter colors represent the dRMS distributions for independent runs, while the darker colors represent the average dRMS distributions from 50 replicates. Tet-1 exhibits greater stability compared to Tet-2. **D** (top) Pairwise inter-molecular contact maps of Tet-1 from AAMD simulations (single run, 5 µs). (bottom) Total number of contacts per residue position ($N_{contact}$) derived through summation of all pairwise contacts along y-axis based on two dimensional pairwise intermolecular contact map. Reported as mean ± SEM using block averaging over 5 time blocks. **E** Representative structure of the CR (aa: 319−341) of Tet-1 from the last microsecond of AAMD simulations (single run, 5 µs) are shown. Parallel helices are shown as yellow (chain 1 & 3) and light green (chain 2 & 4) colored ribbons. Side chains (excluding hydrogen atoms) of CR residues that are in the core (left) and in contact between adjacent helices (right) are shown as sticks, with their colors indicating different residue types as illustrated in the right corner. Source data are provided as a Source Data file.

Importantly, these residues correspond closely to the positions that give rise to intermolecular backbone HN NOEs. Notably, W334 and L330 form an extended hydrophobic patch, enhancing the overall stability of the complex (Fig. 5E). Taking together, the intermolecular contacts in this model are qualitatively consistent with experimental NOEs, showing interactions primarily from methyl groups and backbone NH atoms of aliphatic residues (M/A/L/I), W334 with methyl groups (M/A/L/I), and polar residues (Q/N/S).

The most stable Tet-1 model also remained stable within the full-length CTD (see Supplementary Note 3, Supplementary Fig. 12). W334, surface-exposed in Tet-1, showed enhanced interactions in the full-length CTD, interacting with hydrophobic segments of the flanking regions ($^{276}$FGGNPGGF$^{283}$, $^{307}$MGGGMNF$^{313}$, $^{383}$IGWGS$^{387}$), consistent with its essential role in CTD self-assembly (Fig. 2A, B). Additionally,

aromatic (F283, F289, F313, F316, F367, Y374, W385, F397, F401, W412) and aliphatic (M307, M311, I383) residues in the flanking IDRs formed significant intermolecular contacts in agreement with previous findings[20]. These results reinforce the essential contributions of flanking regions and W334 in CTD self-assembly and further support the structural relevance of the Tet-1 model. Although we were not able to create samples where the assembled state is 100% populated, we did assess whether this structure (Tet-1) appears consistent with NMR-derived measurements we made of the cross-linked forms where the assembly is present. The simulated transverse relaxation rate, $^{15}$N $R_2$, of the Tet-1 model demonstrates higher values compared to monomeric and dimeric ensembles (Supplementary Fig. 11B), consistent with the slowed motions observed in experiments where the multimers are partially populated (Supplementary Fig. 6H). The helix fraction

computed from atomistic simulation of Tet-1 using DSSP[71] also aligns qualitatively with enhanced helicity seen in experiment (Supplementary Fig. 6G), showing stabilization of helical structure within the region spanning G335-Q343 (Supplementary Fig. 11C).

To further assess the structural model, we performed solvent paramagnetic relaxation enhancement (sPRE) measurements (Supplementary Fig. 13). This NMR-based method qualitatively characterizes the relative solvent accessibility of residues within biomolecular structures[72]. The sPRE pattern for cross-linked dimeric variants of S→E S305C showed some residues are more protected. Cross-linked forms revealed complete signal loss for the sidechain NH groups of W334, Q331, Q327, and N319 (Supplementary Fig. 13B), suggesting these groups are highly solvent exposed even upon multimerization. Indeed, our structural model (Tet-1) shows that the side chains of N319, Q327, Q331, and W334 are more solvent-exposed than other residues (Fig. 5E). Given the challenge of incomplete population of the multimeric forms and the inability to directly compute sPRE and NOE data due to transient interactions, the qualitative agreement between the model for the TDP-43 CR tetramer and the experimental constraints (surface accessible positions from sPRE experiments and NOEs on intermolecular contacts) provide the strongest evidence currently possible for the essential structural features of TDP-43 CR in its functionally assembled state.

## Discussion

Uncovering the atomic structural details of TDP-43 CR in its self-assembly pathway is crucial for a mechanistic understanding of CTD functional states and their disruption in disease. In this study, we provide evidence from complementary computational and experimental techniques suggesting that α-helical tetramers are the most likely native form of the CR that is important for its splicing and nuclear retention functions[5,6,62]. Although our previous work showed that the first half of the CR (aa:320–330) was partially structured in the monomeric form[16], these results here explain why the entire 21-residue CR is conserved – to mediate contacts important for assembly. Importantly, our structural model exhibits numerous contacts and stable secondary structure of the region spanning G335-Q343, suggesting a reason for the perfect conservation in vertebrates of this region. Additionally, our structure models rationalize the impact of mutations, such as ALS-associated M337V and engineered M337P[16] and M337A, that disrupt the CR assembly without altering the monomeric helicity[16].

Unlike many known coiled-coils and leucine zippers, the functional CR-mediated assembly is characterized by dynamic and low-affinity α-helical interactions and likely adopts a stable tetrameric configuration with methionine side chains at its core. These dynamic CR self-interactions are commensurate in affinity with the transient, multivalent contacts formed by the disordered regions[20], N-terminal domain[13], and RRMs[14,73] of TDP-43[74], supporting a view where these domains cooperate to form multivalent, functional complexes. Importantly, our integrative approach in determining atomic structural models of TDP-43 multimeric state also shows that although AI-based predictors cannot currently discriminate between the possible conformers of dynamic assemblies, pairing these predictors with molecular simulation and experiment allows for refinement of the structural models that may serve to improve future predictions as more data on dynamic assemblies become available[42].

The sensitivity of TDP-43 nuclear retention to single CR mutations that alter CTD self-assembly highlight the precise tuning of TDP-43 assembly and function. Hence, a series of TDP-43 CR variants may serve as a way to precisely control its nuclear concentration and function[5]. Even without explicit NLS-disrupting variants, as seen in cases of familial ALS of the related protein FUS[75], TDP-43 CTD mutants that disrupt helical assembly may drive TDP-43 aggregation by increasing TDP-43 cytoplasmic accumulation. These data also suggest

that caution should be exercised in interpretation of localization and function of TDP-43 bearing protein domain tags which may influence its size and hence the nuclear/cytoplasmic equilibrium.

Our data support a model in which significant conformational rearrangement must occur for the formation of the β-sheet assemblies of TDP-43 CTD, including the CR, in ALS and neurodegeneration[9,10]. Our recent work suggests that the CR self-interactions are essential not only for TDP-43 function but also for its aggregation[23]. Given that the CR must convert to β-sheet aggregates to form pathogenic fibrils[9,10,76,77], an intriguing therapeutic avenue may be to pharmacologically stabilize the helical assembly observed here to prevent conversion of TDP-43 CTD into β-sheet aggregates. However, as TDP-43 participates in neuronal transport granules[78], it is unclear if excessive nuclear retention of TDP-43 will result in toxic loss of its cytoplasmic function, another important avenue to explore in future cellular studies. Prior work has also suggested that both region preceding the CR and Q/N-rich region directly following it may contribute to CTD aggregation[79,80]. Our simulations also suggest the propensity of these flanking IDRs to nucleate transient intermolecular β-sheets in the tetrameric helical assemblies (Supplementary Fig. 14), which could then spread and lead to conversion of the CR helix into β-sheet structures. Consistent with our findings, previous studies on TDP-43 CTD dimerization also reveal transient intermolecular β-sheet formation in flanking disordered regions[81]. Hence, the assembly of TDP-43 CR appears delicately balanced to drive precisely tuned functional assembly[5] but discourage aggregation. Given the importance of TDP-43 CR in nuclear retention and function, design of small molecules to stabilize CR helical assemblies, as suggested for other transient interactions like the androgen receptor[82] make TDP-43 CR tetramers an exciting drug target for treatment of ALS and other TDP-43 aggregation diseases.

## Methods

### Expression and purification of recombinant proteins

All TDP-43 CTD variants were produced using codon-optimized sequences from a pJ411 bacterial expression vector[16]. The expression was carried out in BL21 Star (DE3) *Escherichia coli* cells obtained from Life Technologies. The proteins were expressed in either LB or M9 minimal media supplemented with $^{15}NH_4Cl$, following a slightly modified version of previously described, with details as follows[5,16]. Bacterial cultures were induced at an optical density (OD) of 0.8 with 1 mM IPTG and incubated for 4 h at 37 °C and shaking at 220 rpm. The cells were then collected by centrifugation (7800 g, 15 min, 4 °C). The resulting cell pellets (2 liters of culture) were resuspended in a 20 mL buffer (20 mM Tris, 500 mM NaCl, 10 mM imidazole, pH 8.0), lysed using an ultrasonic cell disrupter, and the lysate was cleared by centrifugation at a speed such that inclusion bodies are pelleted but membranes are not completely pelleted (15,000 g, 1 h, 4 °C). The insoluble pelleted material, containing the inclusion bodies, was resuspended in a 40 mL solubilizing buffer (8 M urea, 20 mM Tris, 500 mM NaCl, 10 mM imidazole, pH 8.0) and the cell debris was cleared by centrifugation at high speed (74,766 g, 1 h, 19 °C).

The supernatant was then filtered using a 0.45 μm syringe filter, and the protein was purified using a 5 mL Histrap HP column with a gradient of 10 to 500 mM imidazole added to the solubilizing buffer. The purified protein fractions were desalted using a HiPrep 26/10 Desalting Column into TEV cleavage buffer (20 mM Tris, 500 mM GdnHCl, pH 8.0) and subjected to TEV cleavage overnight at room temperature. After cleavage, solid urea was added to the solution to reach a concentration of approximately 8 M urea. The resolubilized protein was then applied to the Histrap HP column to remove the histidine tag and histidine-tagged TEV protease. The cleaved protein fractions were concentrated, buffer-exchanged into a storage buffer (20 mM MES, 8 M urea, pH 6.1) at approximately 1.5 mM, aliquoted, flash-frozen, and stored at −80 °C for further use. Samples with

disulfide cross-links were created using copper phenanthroline catalysis and purified to ensure removal of residual (non-crosslinked) monomers by size exclusion chromatography, as previously described[5].

## Microscopy

Following the manufacturer's instructions, the protein stocks in 8 M urea were diluted 8x (to 1 M urea) with experimental buffer (20 mM MES, pH 6.1) to a final concentration of 150 to 200 μM TDP-43 CTD and then buffer exchanged into experimental buffer with equilibrated 0.5 mL Zeba spin desalting columns from Thermo Scientific following manufacturer instructions. An equal volume of NaCl stock solution prepared in MES buffer was added to obtain final salt concentration of 150 mM to induce phase separation. The samples were gently mixed, and the phase separation was monitored using DIC micrographs obtained with a Nikon Ti2-E Fluorescence Microscope equipped with a 40x objective. To capture the images, 10 μL of each sample was spotted onto a coverslip. The resulting images were subsequently processed using Fiji software.

## In vitro phase separation assay for determining saturation concentration

To quantitatively analyze the phase separation of CTD mutants, we performed assays to determine the saturation concentration by measuring the protein concentration in the supernatant after centrifuging samples with increasing salt concentrations, which induced phase separation. During centrifugation, the protein in micrometer-sized droplets sediments represents the phase-separated state. The protein remaining in the supernatant corresponds to the dispersed phase. By measuring the amount of protein remaining in the supernatant, we determined the saturation concentration, $c_{sat}$, which is the concentration above which the protein undergoes phase separation at the given condition. This value decreases with higher salt concentrations. To initiate phase separation, desalted protein samples were diluted to 80 μM using MES buffer. An equal volume of NaCl stock solution prepared in MES buffer was added to achieve final salt concentrations of 0, 37.5, 75, 150, and 300 NaCl. The samples were gently mixed and then centrifuged for 10 min at 18,000 $g$ at room temperature. After centrifugation, the protein concentration in the supernatant was measured using a Nanodrop 2000c spectrophotometer. All measurements were performed in triplicate to ensure the accuracy and consistency of the data points.

## NMR spectroscopy

NMR spectroscopy was conducted using Bruker Avance III HD 850 MHz or Avance Neo 600 MHz $^{1}$H Larmor frequency spectrometers running Topspin 3.4 or 4.2, respectively, equipped with HCN TCI z-gradient cryoprobes at a temperature of 320 K, 298 K, or 285 K, as specified. The NMR samples were prepared in 20 mM MES buffer at pH 6.1, with the addition of 5% $^{2}$H$_2$O to serve as a lock solvent. The NMR sample in Fig. 1 for the native condition was prepared in 20 mM HEPES, 150 mM NaCl buffer pH 7 with the addition of 5% $^{2}$H$_2$O to serve as a lock solvent. Backbone assignments for the dimers were obtained by transfer of assignment from the monomer (BMRB: 26823, [https://doi.org/10.13018/BMR26823]) and confirmed for the CR through standard triple resonance experiments (HNCACB, HNCA, and CBCACONH) using Bruker pulse sequences (hncacbgp3d, hncagp3d, cbcaconhgp3d, respectively). The acquired NMR data were processed using Bruker Topspin 4.2 software and analyzed using CCPN (2.5.2)[83]. Detailed information on the parameter choices regarding experimental NMR pulse sequences and parameters can be found in our previous publications[5,16]. The $^{15}$N spin relaxation experiments at 850 MHz were recorded with 128 and 4096 total points, with acquisition times of 39 ms and 200 ms, in the indirect $^{15}$N and direct $^{1}$H dimensions, respectively using Bruker sequences hsqct1etf3gpsitc3d,

hsqct2etf3gpsitc3d, hsqcnoef3gpsi, respectively. The spectral width was set to 19 ppm in the indirect $^{15}$N dimension and 12 ppm in the direct $^{1}$H dimension, centered at 116.6 ppm and 4.7 ppm. The $^{15}$N $R_2$ experiments consisted of six interleaved relaxation delays, with an interscan delay of 2.5 s. The CPMG field was set to 556 Hz, and the total $R_2$ relaxation CPMG loop lengths were 16.5 ms, 264.4 ms, 33.1 ms, 132.2 ms, 66.1 ms, and 198.3 ms. Each $^{15}$N $R_1$ experiment consisted of six interleaved $^{15}$N $R_1$ relaxation delays: 5 ms, 1000 ms, 100 ms, 800 ms, 500 ms, and 300 ms. The ($^{1}$H) $^{15}$N heteronuclear NOE experiments were conducted using interleaved sequences with and without proton saturation and a recycle delay of 5 s. The ($^{1}$H) $^{15}$N hetNOE experiments were recorded with 256 and 4096 total points in the indirect $^{15}$N and direct $^{1}$H dimensions, respectively. Additional information on the experimental relaxation NMR parameters can be found in our previous publication[35].

For intermolecular NOE experiments, 3D $^{13}$C,$^{15}$N-filtered/edited NOE−$^{1}$H−$^{13}$C-HSQC (noesyhsqcgpwgx13d) experiments were also recorded with a mixing time of 100 ms and with 128, 60 and 3072 total points with spectral widths of 9, 56 and 12 ppm centered at 4.7, 42 and 4.7 ppm for aliphatic regions in the F2 dimension, or spectral widths of 9, 56 and 12 ppm centered at 4.7, 110 and 4.7 ppm for aromatic regions in the F2 dimension. For the PRE experiments, the TEMPOL reagent was purchased from Sigma-Aldrich. The protein stocks were diluted from storage buffer containing 8 M urea and buffer exchange to 20 mM MES at pH 6.1, followed by the addition of 1.0 M TEMPOL stock solutions to achieve final concentrations of 0, 5, and 20 mM TEMPOL in a 30 μM (in dimer units) protein solution. Subsequently, $^{1}$H-$^{15}$N HSQC experiments were conducted on an 850 MHz spectrometer with acquisition parameters set at 4096 direct points and 256 indirect points. The acquired NMR data were processed using NMRPIPE with consistent parameters, and peak intensities were analyzed using CCPN software (2.5.2). Peak intensities were normalized to intensities of samples without TEMPOL to determine the attenuation of resonances induced by the presence of TEMPOL.

## Analytical ultracentrifugation

Sedimentation velocity experiments were conducted at 25 °C and 50,000 rpm (corresponding to 201,600 x g at 7.20 cm) on a Beckman Coulter ProteomeLab XL-I or Beckman Optima XL-A analytical ultracentrifuge following standard protocols[84]. Samples of TDP-43 CTD (wild-type) were prepared by Zeba desalting columns, as described above, flash frozen for shipment, and thawed for measurement. Samples of cross-linked forms of TDP-43 CTD (TDP-43 CTD S273C and TDP-43 CTD S → E S273C, S → E S305C, S → E S317C and S → E A236P S305C were prepared from frozen protein stocks in 20 mM MES pH 6.1 and 8 M urea that were thawed and diluted to 15 mL in a Falcon tube. Buffer was exchanged using a 10 kDa Amicon filter at 3000 × g and 4 °C. The material was washed three times until the urea concentration was below 20 mM and the protein concentration was approximately 100 μM, except for TDP-43 CTD S273C cross-linked (without solubility enhancing S → E substitutions) that was prepared such that the final urea concentration was -100 mM to avoid issues with sample aggregation. Before the sedimentation experiment, samples were centrifuged at 18,000 × g and 4 °C to clear aggregated material. Proteins were diluted with the final Amicon filtrate and studied at various loading concentrations in 3 mm or 12 mm pathlength cells. Sedimentation velocity scans were collected using absorbance optical systems at 280 nm, and data were analyzed in SEDFIT[85] (version 16.1c) in terms of a continuous c(s) distribution of Lamm equations. When necessary, early scans were not used for model fitting to remove contributions from large aggregates. Solution densities ρ and solution viscosities η were measured at 20 °C on an Anton Paar DMA 5000 density meter and Anton Paar AMVn rolling ball viscometer, respectively, and corrected to the experimental temperature. The partial specific volumes of the peptides were calculated in SEDNTERP[86]

(version 3.0.4) and corrected for isotopic substitution. Experimental sedimentation coefficients, s were corrected to standard conditions, $s_{20,w}$. To determine the affinities for the self-association of the S → E cross-linked variants S305C and S317C, the c(s) distributions were integrated to get the weighted-average sedimentation coefficients and sample concentration. The resulting isotherms were modeled in SEDPHAT[87] (version 15.2b). The S → E cross-linked variant S273C and helix disrupting cross-linked S → E A236P S305C variant presented as dimers at all concentrations. For WT CTD, sedimentation velocity data at two concentrations were modeled directly using Lamm equations describing a monomer-dimer-tetramer self-association and other alternative models. Similarly, in the case of S → E cross-linked variant S317C, sedimentation velocity data at three concentrations were modeled directly using Lamm equations describing the dimer-tetramer self-association. Here, data were plotted in GUSSI[88] (version 2.10). All concentrations are in dimer subunits for cross-linked dimers. For the non-self-associating species, plots describing the dependence of the sedimentation coefficient with concentration were generated in SigmaPlot (https://grafiti.com) (Supplementary Fig. 9).

## Cloning, cell culture, and stable cell production
Generation of stable cells lines HEK293[HA-TDP-43] expressing HA-tagged WT, and A326P was previously completed[62]. Constructs to express TDP-43 mutants Q327A, L330A, Q331A, S333A, W334A, L340A, and Q343A were generated by site-directed mutagenesis[3–6] using the primers listed in Table S3. Generation of HEK293[HA-TDP-43] stable cell lines, expressing WT and mutant HA-tagged TDP-43 was carried out according to previously described protocols[14,15] as described below.

## TDP-43 cellular nuclear retention experiments
In brief, HEK293-Flp-In T-Rex 293 cells (Thermo Fisher Scientific) were stably transfected to express HA-TDP-43 upon induction with tetracycline (1 μg/ml). Cells were grown and maintained in DMEM (Dulbecco's Modified Eagle's Medium–High Glucose, Corning) supplemented with 10% FBS (fetal bovine serum) and incubated in a humid atmosphere at 37 °C and 5% $CO_2$. Expression of HA-tagged TDP-43 construct was induced at 30% confluence for 48 h. Cell fractionation, immunoblotting, and quantification of nuclear to cytoplasmic ratio of HA-TDP-43 variants[62], probing with TDP-43 antibody (10782-2-AP, Proteintech).

## Immunofluorescence analysis
Glass coverslips were pre-coated with poly-D-lysine (0.1 mg/mL) and incubated for 10 min at room temperature. The coverslips were rinsed with Phosphate-buffered saline (PBS) 1x three times and left to dry under the tissue culture hood for at least 45 min. HEK293[HA-TDP-43] wild type and mutant cells were plated on the pre-coated coverslips and incubated at 37 °C in a humid atmosphere incubator with 5% $CO_2$ for 24 h. HA-TDP43 expression was induced by 0.1 μg/mL tetracycline for 48 h. Cells were fixed in 4% paraformaldehyde in PBS 1X for 20 min, and washed with PBS three times, 5 min each. Cells were then treated with 5 μg /mL wheat germ agglutinin Alexa fluor Oregon green 488 conjugate (Invitrogen # W6748) in PBS to stain the cytoplasm region. The coverslips were then washed three times with PBS and permeabilized with 0.2% Triton X-100 for 5 min on ice. Cells were washed with PBS and incubated with the primary antibody anti-HA (dilution 1:200, Cell signaling #2367S) for 1 h to label HA-TDP-43. Cells were washed three times with PBS and incubated with the secondary antibody Alexa fluor 555 donkey anti-mouse (Invitrogen #A31570) for 1 h protected from light. The coverslips were then incubated for 5 min in DAPI 1 μg/mL in PBS for nuclear staining and washed twice with PBS. The slides were mounted with ProLog Glass Antifade Mountant, according to manufacture instructions

(Thermo Fisher #P36980). Imaging was performed with a Keyence BZ-X800 immunofluorescence microscope using a 40X and 60X objective. The intensity of nuclear and cytoplasmic HA-TDP-43 was measured using standard ImageJ protocols.

## AlphaFold-multimer predictions
TDP-43$_{310-350}$ multimer structures were generated using AF2-Multimer, providing a multi-sequence FASTA file of TDP-43$_{310-350}$. The predictions for multimers were executed based on multiple sequence alignment alone, with the exclusion of PDB templates through the use of the parameter --max_template_date=1950-1-1. The full database (--db_preset=full_dbs) was used to create the alignments. Dimer and tetramer models were generated utilizing both AF2-Multimer v2.2.0 and v2.3.0, while trimer, hexamer, and octamer structures were generated using AF2-Multimer v2.3.0. In AF2-Multimer v2.2.0, one seed per model was employed, resulting in a total of five predictions. In AF2-Multimer v2.3.0, five seeds were used per model, yielding a total of 25 predictions. The relaxed structures (--use_gpu_relax=True) without major clashes were employed as the initial configurations for AAMD simulations. It is important to note that the predicted structures contain major clashes, especially in the case of higher-order multimers; thus, the number of structures simulated for each multimeric state differs from one another.

## All-atom MD simulations
**Initial structure, force-field choice and system setup.** The protein was modeled using the Amber03ws force field[89] (https://doi.org/10.5281/zenodo.17059939[90]) and solvated with TIP4P/2005 water model[91]. We selected the AMBER03ws protein force field[89] for its proven ability to accurately capture both the global dimensions and local structural features of various intrinsically disordered proteins, including the TDP-43 CTD[27]. Specifically, AMBER03ws has been shown to reliably reproduce NMR-derived helical propensities in our previous simulations of a subsegment of the TDP-43 CTD (amino acids 310–350) used in this study[5,16]. Hence, we conducted 45 μs-long unbiased, monomeric simulations of TDP-43$_{310-350}$ (five independent replicates, with different starting configurations) using this force field and found good agreement between the simulated and experimental NMR observables, including $^3J_{HNH\alpha}$ scalar coupling constants and chemical shifts, as well as the simulated and NMR-derived helix fractions[16] (Supplementary Fig. 15). Based on these results, we selected AMBER03ws to simulate the structural ensembles of the higher-order assemblies of TDP-43$_{310-350}$ presented in this study.

The dimer (5 models from AF2-Multimer version 2.2.0 (v2.2.0) and 13 models from AF2-Multimer version 2.3.0 (v2.3.0), 18 models in total; Supplementary Figs. 16, 17), trimer (8 model from AF2-Multimer v2.3.0; Supplementary Figs. 18, 19), tetramer (2 models from AF2-Multimer v2.2.0 and 5 models from AF2-Multimer v2.3.0, 7 models in total; Supplementary Figs. 20, 21), hexamer (5 models from AF2-Multimer v2.3.0; Supplementary Figs. 22,23), and octamer (4 models from AF2-Multimer v2.3.0; Supplementary Figs. 24, 25) structures of TDP-43$_{310-350}$ from AF2-Multimer were employed as the starting configurations for AAMD simulations with explicit solvent and ions. For the cross-linked dimer, TDP-43$_{310-350}$ S317C_S−S, serine at position 317 was mutated to cysteine in each chain of the parallel dimer models generated using AF2-Multimer v2.2.0 in Chimera version 1.18. A disulfide bond (S−S) was introduced between the cysteine residues using the pdb2gmx -ss command using GROMACS version 2021.6. Simulation details for all systems are summarized in Table S4.

To mimic the physiological salt concentration (150 mM), Na$^+$ and Cl$^-$ ions were added to the protein-water system. The improved salt parameters from Lou and Roux were used for all simulations[92]. The protein was solvated in a truncated octahedron box, ensuring 1.2 nm separation between protein atoms and box edges.

**Simulation protocol.** Following solvation, the system was minimized via the steepest descent algorithm using GROMACS 2021.6[93]. After minimization, the Nose-Hoover thermostat[94] was used for temperature equilibration at 300 K with a coupling constant of 1 ps for protein, water and ions. After the NVT equilibration, a 100 ns NPT equilibration was run using the Berendsen barostat[95] with isotropic coupling and a coupling constant of 5 ps to achieve pressure of 1 bar. Production simulations were conducted in the NPT ensemble (1 bar, 300 K) using the Langevin Middle Integrator[96] (friction coefficient = 1 ps$^{-1}$) and the Monte Carlo Barostat with isotropic coupling in AMBER 22[29]. Hydrogen mass was increased to 1.5 amu to enable a timestep of 4 fs. Short-range nonbonded interactions were calculated based on a cutoff radius of 0.9 nm, while long-range electrostatics were handled with the Particle Mesh Ewald (PME) method[97]. Hydrogen-related bonds were constrained using the SHAKE algorithm[98]. Following the production runs, Amber NetCDF trajectory files were converted into GROMACS.xtc compressed trajectories using the CPPTRAJ module[99] of AmberTools 22[29] for subsequent analysis.

**Trajectory analysis.** The distance root mean square deviation of atom distances (dRMS) for all heavy atoms across the CR (amino acids 320–341) from reference structures was calculated using the gmx rmsdist command. Minimum distance between the monomeric units in the two-chain simulation trajectories were computed using gmx mindist. For the other analyses described in the text, the first 250 ns of each replica were excluded as equilibration time. In contact analysis, two residues $i$ and $j$ with sequence separation greater than 3 ($|i-j| > 3$) were counted in contact if any two heavy atoms of the residues were within 0.6 nm of each other. This cutoff has been extensively used in our previous studies and has been thoroughly explained in the work of Zheng et al.[100]. Helix fraction calculations were performed using the gmx do_dssp, relying on the DSSP library[71]. The NMR relaxation parameters ($^{15}N\ R_2$) were computed using a previously published method[16,34]. The most representative structures from AAMD simulations of multimer models were determined using the gmx cluster tool with a GROMOS clustering algorithm. Backbone atoms of CR residues (aa: 320–341) were considered, with an RMSD cutoff of 0.3 nm, based on the last microsecond of the trajectory. Analyses performed with GROMACS were conducted using version 2021.6. All snapshots from atomistic simulations were generated using UCSF Chimera version 1.18[101]/ ChimeraX version 1.8[102].

### Reporting summary

Further information on research design is available in the Nature Portfolio Reporting Summary linked to this article.

## Data availability

Sequences for proteins used in this work are included in the Supplementary Information file. Plasmids for these sequences are available from Addgene https://www.addgene.org/Nicolas_Fawzi/. NMR chemical shift assignments for the TDP-43 C-terminal domain residues 300-360 at pH 7 are deposited with the Biological Magnetic Resonance Data Bank (BMRB) entry 53381. Time domain and processed NMR data are deposited at BMRbig with entry identifier bmrbig135 [https://bmrbig.org/released/bmrbig135]. BMRB entry 26823 from previously published studies[16] was used in this work. AlphaFold2-Multimer–predicted structures used in the AAMD simulations (in PDB format), together with their corresponding .pkl files containing pLDDT and PAE information, have been deposited on Zenodo [https://doi.org/10.5281/zenodo.17059939] (ref. 90). Simulation input files, along with the starting and final configurations for all atomistic simulations, have been deposited on Zenodo [https://doi.org/10.5281/zenodo.17059939] (ref. 90). Unless otherwise stated, all data supporting the results of this study can be found in the article, supplementary, and source data files. Other materials are available upon reasonable request to the corresponding authors. Source data are provided with this paper.

## Code availability

Codes to run and analyze atomistic simulations are available publicly and can be found at https://ambermd.org/, https://gromacs.org/, https://www.mdtraj.org/ and https://www.mdanalysis.org/. Codes to reproduce residue pairwise contacts are publicly available on Zenodo [https://doi.org/10.5281/zenodo.13963614] (ref. 103). For preparation and analysis of the analytical ultracentrifugation (AUC) data, publicly available software was used and can be found at https://www.utsouthwestern.edu/research/core-facilities/mbr/software/, https://sedfitsedphat.nibib.nih.gov/software, https://sedfitsedphat.nibib.nih.gov/software, https://www.utsouthwestern.edu/research/core-facilities/mbr/software/, http://www.jphilo.mailway.com/sednterp.htm.

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

## Acknowledgements

This research was supported by NINDS and NIA R01NS116176 (to N.L.F. and J.M.). J.S. was supported by a Milton Safenowitz Postdoctoral Fellowship from the ALS Association (23-PDF-629 to J.S.) and Judith and Jean Pape Adams Postdoctoral Fellowship at Brown University (to J.S.). R.G. was supported by the Intramural Research Program of the National Institute of Diabetes and Digestive and Kidney Diseases (NIDDK) within the National Institutes of Health (NIH). The contributions of the NIH author are considered Works of the United States Government. The findings and conclusions presented in this paper are those of the author and do not necessarily reflect the views of the NIH or the U.S. Department of Health and Human Services. Y.M.A. was supported in part by NINDS and NIA R01NS114289 (to Y.M.A.). NMR experiments were conducted with the support of the Structural Biology Core Facility in the Division of Biology and Medicine at Brown University and with assistance from Mandar Naik. Use of the Texas A&M High Performance Research Computing is greatly acknowledged for the computational resources utilized in this work. We thank Dr. Tien Phan for help with the initial AF2-Multimer setup. The funders had no role in study design, data collection and analysis, decision to publish or preparation of the manuscript.

## Author contributions

N.L.F. and J.M. conceived of and supervised the research. A.R. and J.M. designed and performed the AlphaFold2-Multimer predictions and MD simulations, and carried out the analyses. P.M. contributed to the interpretation of the simulation results. J.S. and N.L.F. designed and analyzed the biochemical and NMR experiments, which were performed by J.S., J.F.M.-O., L.B., R.V., J.Z., S.-H.W., and V.J. R.G. designed, performed, and analyzed the AUC experiments. P.M.P., L.D.M., A.R.T., and Y.M.A. designed, conducted, and analyzed the cellular and biochemical fractionation assays. A.R., J.M., and N.L.F. wrote the manuscript with input from all authors.

## Competing interests

J.S. is currently an employee of Nuage Therapeutics. The remaining authors declare no competing interests.
