## [Transparent Peer Review file · Nature Communications]

Structural details of helix-mediated multimerization of the conserved region of TDP-43 C-terminal domain

Corresponding Author: Professor Nicolas Fawzi

Version 0:

Reviewer comments:

Reviewer #1

(Remarks to the Author)

The manuscript by Fawzi et al. investigated the atomic structural details of TDP-43 CR helix-mediated assembly utilizing an integrative modeling strategy including NMR spectroscopy, SV-AUC, AF2-Multimer and all-atom molecular dynamics simulations. They found that the native state of TDP-43 CR is α -helical under physiological conditions, and helix-mediated interactions are linked to phase separation of C-terminal domain. The simulation results showed that TDP-43 CR assembly are stabilized by a methionine-rich core with specific contributions from a tryptophan/leucine pair. Overall, this work is carefully designed and executed. The manuscript is well written and the conclusions are supported by data. Some additional comments are listed below.

1. In the method section, the simulation setup details including systems, number of simulations, number of atoms are suggested to be summarized. It is also necessary to give the reason for choosing Amber03ws force field for protein structure.
2. In Figure 5B, it would be easier for readers to understand the plot of structural stability if different size of oligomers are labeled.
3. The authors performed 5- μ s AAMD simulation for each system. However, the duration of some trajectories are less than 5 μ s in Figure S8A and S10C.
4. Previous computational studies (10.1021/acs.jcim.3c00513; 10.1021/acs.jcim.4c00943) have investigated the dimerization of TDP-43 low-complexity domain and reported the conformational transition from helices to β -sheets, which may be helpful for the discussion.

Reviewer #2

(Remarks to the Author)

Major comments

The authors developed a model for the oligomerization and initial stages of pathogenic aggregation of TAR DNA binding protein-43 (TDP-43) by structure/function analysis of the C-terminal domain (CTD) of TDP-43.

The study largely consists of the analysis of wild-type and mutant CTD proteins. The mutant proteins contained serine to glutamic acid mutations (S->E) to preserve solubility. CTD proteins were His-tagged for purification, followed by removal the tags and his-cleaving TEV protease. They were subjected to denaturing conditions in 500 mM GdnHCl during purification and exchanged into 8M urea for storage. It is not clear if, when or how urea was removed before subsequent analysis. What was the solvent used in the experiments? The native oligomeric state of these proteins isolated and stored under these harshly denaturing is open to question and seems to be a major flaw in the study.

Fig. 3A shows SV AUC c(s) distributions as a function of loading concentration of WT CTD. There is a major 1.5 S species and at the highest loading concentration, 107 μ M, a minor 2.9 S species. From these results, the authors propose that there is a monomer-dimer equilibrium that possibly proceeds to higher order oligomerization. However, the small 2.9 c(s) peak shown in Fig. 3A does not strongly support a monomer-dimer equilibrium of wild-type CTD, or at least one with a dissociation constant of less than \sim 100 μ M.

The authors proceed to state “to better characterize the assembled state, we experimentally trapped a dimeric state by forming a disulfide cross-link (S-S) between engineered cysteine residues at S273C”. This maneuver seems highly suspect – it may represent an artifice to create dimers and subsequent higher order oligomers with no physiologic or pathologic relevance. The rationale for making the mutations S273C, S305C and S317C is not presented. These mutations evidently are not associated disease states.

SV AUC of the S305C and S317C CTD mutants revealed the formation of higher order oligomers beyond the dimer. Modeling with to dimer-tetramer and dimer-tetramer-octamer equilibria was done by sw isotherm analysis and Lamm equation modeling. Although good fits to the models was obtained, it is not clear if equally good fits to alternative models would be obtained. This does not detract substantially from the results, which clearly reveal some sort of oligomerization or aggregation beyond the dimer. However, as noted above, for proteins that are engineered to be soluble with S → E mutations, stored in chaotropic buffers and artificially disulfide-linked to form dimers, the relevance of these higher order structures is questionable.

The peak areas of c(s) distributions seem inconsistent with the loading concentrations. In Fig. 3A there is a ~6-fold increase in loading concentrations. Are the associated peak areas consistent with this? In Fig. 3B, are the c(s) peak areas consistent with loading concentrations of 11 μM and 21 μM?

For nearly monodisperse systems, continuous c(s) distribution analysis often produces an accurate estimate of the molar mass from combined estimates of the diffusion coefficient (expressed in terms of the frictional ratio, f/f_0) and sedimentation coefficient. Were the molar mass estimates of the 1.5 S species and the 2.9 S species consistent of monomeric and dimeric CTD? Fig. 3A legend states a molar mass of 31.5 kDa. Where did this number come from?

Minor comments

Rotor speeds evidently were 50000 rpm, not 5000 as stated in the manuscript.

The authors state the SV-AUC interference optical system was used, but only absorbance data are shown.

References are not given for SEDPHAT, Lamm equations modeling or GUSSI.

ALS is not defined in the abstract.

Regarding the statement “... the dynamic association of TDP-43 CTD ...”, it is not clear what TDP-43 CTD is associating with.

The authors state in that SV AUC experiments “were conducted at the specified temperature”. Fig. 3 and Fig. S6 state SV AUC was done at 25 C. The temperature is not specified in Fig S7. If all experiments were at 25 C, please revise Methods to state that SV AUC experiments were conducted at 25 C.

Reviewer #3

(Remarks to the Author)

The manuscript by Rizuan et al. focuses on the role of the conserved region (aa 310-350) in TDP-43's low complexity region (LCD), in particular its structural details and role for TDP-43 assembly. For this, they integrate several biophysical and biochemical approaches as well as molecular dynamic simulations to investigate the structural rearrangement and interactions of the conserved region during higher order assembly. The study not only confirms the presence of an alpha-helix under various buffer conditions, but also pinpoints the importance of several residues for promoting self-assembly, in particular L330, W334 and methionines, while most polar residues appear to dampen phase separation. The significance of this study lies in the characterization of the conformational states of the alpha helix of TDP-43, highlighting its role in promoting oligomer formation, which might affect biophysical and functional activity of TDP-43.

Overall, the authors provide extensive biochemical and biophysical data supporting their findings. The characterization of the role of different residues in TDP-43 LCD can potentially provide important insight into the mechanism of TDP-43 self-assembly. Structural details of TDP-43's CR can so far only be obtained using the LCD, and biochemical experiments have been performed introducing disulfide bonds to promote oligomer formation, which is also a major limitation of the study and should be discussed accordingly by the authors (and to some extent addressed as mentioned below).

In summary, we recommend publication of this work in Nature Communications, provided that the following points are being addressed by the authors:

Major points:

I) the condensates presented in Figure 2B are very small and it is nearly impossible to see the described morphological differences described by the authors, such as reduced roundness. Please additionally provide zoomed-in images in this panel. Also, please provide scale bars to allow judgement of the condensate size.

II) To due the strong focus on structural analysis of the LCD, the study is limited with respect to the full length protein. To support their model for a role of the CR in TDP-43 self-assembly, the authors should support their findings by more cellular data in addition to the ones already provided in the SI. Here they should use key mutants identified in their structural analysis that either disrupt or enhance the alpha-helix (w.g. by comparing WT and A326P with at least L330A and W334A). Are those mutants also in context of the full-length protein in cells less/more prone to undergo higher order assembly/phase separation? This could for example be done by studying their subcellular localization in response to stress or by applying size exclusion chromatography as previously shown by Dos Passos et al (PMID: 38422113) using their stable cell lines. Ideally, this should be performed in absence of endogenous TDP-43 (if possible) to avoid complex formation of mutant with

wt TDP-43. Regarding Figure S3, please indicate the difference between upper and lower blot panel in A and B, respectively, as the signal distribution in the different fractions for TDP-43-HA appears to be quite variable. Also, why is endogenous TDP-43 not visible in the upper blot panel? Which blots were used for the quantification? Also, are those mutants in intact cells localized to the nucleus or already cytoplasmically mislocalised (i.e. more than WT as this is an overexpression system)? The provided assay mainly addresses the extractability/ nuclear retention of TDP-43 upon cell lysis – this should also be clear from the experimental description. The authors should therefore only use “nuclear retention” and not refer to “nuclear localization” or “nuclear depletion” (unless specific mutants show a significant cytoplasmic mislocalization in intact cells).

Minor points:

1. The authors start their work by showing the conservation of the CR in vertebrates – yet they only show 2 species (zebrafish and homo sapiens). A comparison with additional species would further strengthen this point.
2. The authors start their result section with referring to “recent work using hydrogen-bond disrupting variants” that suggested a β -sheet confirmation for the CR of TDP-43 – yet they fail to provide any references.
3. There appears to be a typo in the sentence “Expanding from our previous results for the 5M->A variant within CR, which collectively impaired the CTD phase separation, each single methionine substitution within CR (M322A, M333A, M336A, M337A, M339A) also reduce phase separation with approximately two-fold higher csat than WT (Fig. 2A)” -probably the author refers to M323A.
4. Sedimentation assay shown in Figure S1: Even though standard deviations of three plicates are mentioned in the figure legend, they appear to be lacking from the figure itself. Did the author compare these results with other methods employed to characterize protein multimers/oligomers (e.g. DLS or are strongly assembly-promoting mutants measurable by SV-AUC)?
5. TDP-43 cell fractionation. How many replicates were performed for this experiment? Please indicate this also in the respective figure legend. Also, to allow the reader a better understanding of the reproducibility/variability of this assay, it would be better to present also the individual values for each experiment in Figure 2H. Additionally, the ANOVA statistical test assumes that the data are normal distributed, did the author check this assumption? Alternatively, I would recommend the use of non-parametric test like Kruskal-Wallis.
6. It is very interesting that a mutation of two closed amino acid very like S333A and W334A leads to two opposite effects on the stability of the alpha helix (Figure 2A, 2B). The author may use this example to emphasize the specific contribution of W on alpha helix stabilization.
7. In case there is a public-repository for NMR data, the author should upload the raw data to make them accessible for the scientific community (FAIR principles to optimize the reuse of data).
8. The authors indicate that Q327, S332A and S333A were not measurable by NMR due to broadening of the CR signal and refer to the strong self-assembly they have observed before. Yet, Q331 showed a similar strong self-assembly in Figure 2A, but could be measured by NMR? Do the authors have an explanation for this?
9. Please include page numbers for revisions.

Reviewer #4

(Remarks to the Author)

TDP-43 has been identified as a major component of inclusions found in patients with Amyotrophic Lateral Sclerosis (ALS) and certain forms of dementia. This RNA-binding protein contains an intrinsically disordered region (IDR) approximately 160 amino acids long within its C-terminal domain (residues 267–414). Numerous studies, including analyses of postmortem patient biopsies, have demonstrated that these aggregates primarily consist of fragments from the IDR. Additionally, most disease-associated familial mutations are located within this region. Therefore, understanding the aggregation mechanisms of the IDR is crucial for elucidating the underlying causes of these diseases.

Current knowledge from various biophysical characterizations indicates that the IDR of TDP-43 forms an amyloid-dominant structure, as evidenced by cryo-electron microscopy (cryo-EM) studies on inclusions from postmortem patients. Although not entirely disordered, the IDR contains a transient alpha-helical region (residues 319–341; the CR in this manuscript) that has a propensity to self-assemble. This domain facilitates the protein's ability to undergo phase separation, which is related to its role in forming stress granules or RNA granules.

In the manuscript by Rizuan et al., the authors aim to understand the detailed structure of the assembly of this alpha-helical region in its dynamic state. While their experiments and simulations are extensive and methodologically sound, the article falls short of achieving its main goal—to provide a detailed understanding of the structural aspects of the C-terminal region of TDP-43. The reliance on molecular dynamics (MD) simulation interpretations is heavy, yet the experimental support is weak. The rationale behind the design of specific mutants and their subsequent interpretations is unclear. Furthermore, some interrelated factors, such as the interplay between alpha-helical propensity and self-association tendency—which could significantly affect the results—have been overlooked.

The major concerns of this work are listed below.

(1) It has been reported that the IDR of TDP-43 exhibits a stronger tendency to aggregate at higher pH levels and increased NaCl concentrations. This is because the net positive charge of the disordered region leads to enhanced self-association as the pH rises, reducing protonation and thereby diminishing electrostatic repulsion (PMID: 30814253). Similarly, higher NaCl concentrations screen electrostatic repulsive forces, further promoting aggregation (PMID: 30489059). Although the alpha-helical propensity remains and the chemical shift perturbations between pH 7 with 150 mM NaCl and pH 6.1 are minimal, the effects on self-association and alpha-helical tendency—which are key for the protein's phase separation or aggregation—are also small, as demonstrated in the authors' seminal paper (Figures 5 and 6; PMID: 27545621). Given that even small differences in alpha-helical structural propensity under these two conditions can significantly impact the helical component population and self-assembly—as shown in their previous work—such variations should not be dismissed as negligible or

overlooked in subsequent studies.

(2) In the alanine-scanning studies on the CR region, the alpha-helical propensity would differ among these mutations, and this structural tendency modulates self-association, as also noted in the authors' previous work (PMID: 32132204). However, the dynamic nature of the helical structure in self-association seems to have been overlooked. Although the manuscript discusses helical propensity changes in one paragraph, these are attributed solely to results from "dimerization"—bearing in mind that the dimer was the authors' input for AlphaFold-multimer, not an outcome of the simulation. In the chemical shift analysis ($\delta^{15}\text{N}$ CSP), this parameter is affected by both the equilibrium between different structural populations (random coil versus alpha-helix) and interactions from self-association. Thus, the concluding sentence in this section, "This support(s) the idea that residues stabilizing CR helix-helix contacts in the dimer state contribute to the phase separation of CTD," may not be fully justified.

(3) In their attempt to verify the helix-mediated assembly and function, the authors used a cell model to measure the nuclear-to-cytoplasmic ratio. They found that mutants enhancing helical assembly also increased nuclear localization. However, the underlying mechanism by which helix-promoted self-assembly enhances nuclear import is not apparent; one might expect that larger assemblies would impede nuclear entry. Although the authors note this discrepancy by stating that it "highlights the need for further investigation into this mechanism," this point should be elucidated. Furthermore, since the most extensively studied characteristics of TDP-43 assembly involve its cellular toxicity or amyloid-fibril formation, the authors should also investigate how these helix-mediated assembly mutants affect these well-characterized phenomena.

(4) The AUC data indicated the presence of dimers and possibly larger oligomeric species in equilibrium. However, additional complementary methods could be employed to more conclusively determine the size and distribution of these oligomers. Techniques such as dynamic light scattering, nanoparticle tracking analysis, or even cryo-electron microscopy imaging could provide more insights into the oligomerization states. Applying these methods would strengthen the rationale for the subsequent molecular dynamics simulations that use different oligomeric starting models.

(5) The rationale for mutating 13 serines to glutamates (S to E) in the non-CR regions to enhance solubility is unclear. Many serine phosphorylation sites are associated with this protein's pathological aggregation (S379, S403, S404, Ser409, Ser410), and substituting serine with glutamate is often used to mimic phosphorylation effects, which leads to amyloid-fibril. Introducing an additional cysteine to force dimerization is also questionable. While their previous work investigating alpha-helical propensity using disulfide linkages was reasonable, applying this method here may not be suitable because the aim of this manuscript is to characterize the structure of the dynamic alpha-helical assembly. The antiparallel conformation in the AlphaFold-multimer-generated structures for MD simulations is not considered in these cases. Furthermore, the NMR experiments in Figures 3D–F are expected results because cross-linking at the N-terminus connects two monomers, inducing helix-helix interactions (evident from chemical shifts) and increasing R_2 relaxation rates due to the larger molecule reducing overall tumbling. As mentioned above (point 2), chemical shifts are influenced by both structural contacts and the equilibrium between coil and helix conformations. Here, the measured R_2 is also related to the chemical exchange between coil and helical conformations, adding to the line broadening caused by interactions between monomeric and dimeric states. These factors are ignored and cannot be simply attributed to the formation of higher-order (beyond dimer) structures.

(6) The NOE data provided information only on the types of amino acid contacts but did not reveal detailed structural insights. While it is understandable that these may be the best results obtainable using NOE to characterize such a heterogeneous interacting system, the NOE experiments themselves are also limited because they were conducted on a construct where six phenylalanine residues in the non-CR region were replaced with alanine to increase solubility. This modification could overlook potential assembly contacts, such as the interaction involving tryptophan 334 in the CR region through pi-pi stacking (PMID: 33909608, PMID: 29511089). While the authors aim to focus on alpha-helix interactions, it has been reported that other parts of the IDR contribute to phase separation—for example, aromatic residues acting as "stickers" in the organization and formation of higher-order oligomers. These interactions may be critical in assembly studies and experimental characterizations but are overlooked in the MD modeling work. Furthermore, the "structure" of the helical assembly used for MD simulations was derived from AlphaFold-Multimer models. Lacking direct evidence of tetramer formation—and considering that many experimental investigations have used induced dimerization through disulfide linkages—it is difficult to be convinced by the simulated results. Collectively, these data are not convincing that this manuscript significantly advances our current knowledge or offers novel insights into the structural mechanisms of TDP-43 aggregation.

Reviewer #5

(Remarks to the Author)

Version 1:

Reviewer comments:

Reviewer #1

(Remarks to the Author)

The revisions are well executed, and I have no further suggestions at this stage.

Reviewer #2

(Remarks to the Author)

In the revised manuscript, the authors provide more detail of SV AUC of TDP-43 CTD in an attempt to establish the stoichiometry of a putative self-association. They report that Lamm equation modeling SV scans of TDP-43 loaded at 58 μM and 107 μM using SEDPHAT is consistent with a monomer – dimer – tetramer model with weak self-association to the dimer and stronger association of the dimer to the tetramer. The fitted dissociation constants for the monomer – dimer and dimer – tetramer equilibria were 2000 μM and 2.9 μM , respectively. They found that alternative monomer-dimer, monomer-trimer, monomer-tetramer and monomer-dimer-trimer models “resulted in poor fits and/or physically unreasonable sedimentation coefficients”, but that “excellent fits” were found for the monomer-dimer-tetramer model. The analysis represents state of the art use of SV AUC to model self-association equilibria, although they could also have presented F-statistics, which are used in SEDPHAT to show statistical significance of differences in fitting alternative models. The acceptance of the model must be viewed with skepticism. The highest loading concentration of 107 μM is ~20-fold lower than the K_d for the weaker, obligate dimerization step. As a result, as they note, there is very little population of the tetramer. To accept a binding model with confidence, a system should approach saturation. It is surprising that reported errors are relatively small. Overall, the available concentrations limit the investigation of an apparent ultraweak self-association of the protein.

Reviewer #3

(Remarks to the Author)

Reviewer #4

(Remarks to the Author)

I thank the authors for addressing my previous concerns with their revised manuscript. Especially, the updated analysis of the AUC data demonstrates the monomer-dimer-tetramer model as the best fit, which effectively resolves many of my concerns regarding the computational parts. However, I suggest that the alternative models evaluated (i.e., monomer-dimer, monomer-trimer, monomer-tetramer, and monomer-dimer-trimer) and their respective fitting parameters (such as R^2 or reduced chi-square values) be included in the Supporting Information for completeness and clarity. Additionally, based on the relaxation dispersion NMR data previously presented in Figure 7A of the 2016 Structure paper, I did a quick estimation of the apparent dissociation constant (K_d). Using the reported values of $k_{\text{off}} = 1100 \text{ s}^{-1}$ and $k_{\text{on}} = 100 \text{ s}^{-1}$, and considering the protein concentration ($[\text{protein}] = 107 \mu\text{M}$), the estimated K_d is approximately $(1100/100) \times 107 = 1177 \mu\text{M}$. This estimate aligns well with the magnitude derived from the current AUC fitting and may provide valuable complementary support. If I am correct on this estimation, I recommend that the authors consider including this comparative estimation to strengthen the coherence of their analyses.

Reviewer #5

(Remarks to the Author)

By incorporating the revisions and including microscopy analyses highlighting the distinct localization of the mutants (new Fig. S4C), the authors have adequately addressed my comments. Therefore, I recommend the manuscript for publication.

Version 2:

Reviewer comments:

Reviewer #2

(Remarks to the Author)

The authors now present global reduced chi-square values for the fits with various Lamm equation models (Supplementary Table S1). They report a significantly better fit with the monomer–dimer–tetramer model, which has improved the manuscript.

The authors state solubility issues limit the highest concentration allowed in SV AUC experiments. While this is unfortunate, it is unsound to assert conclusions that extend beyond the resolution or reliability of the available data.

The authors cite Brautigam (<https://pubmed.ncbi.nlm.nih.gov/21187153/>) and state that the paper “demonstrates that one does not require saturating concentrations to reliably measure an affinity due to the large number of data points fitted in the analysis (in this case, more than 71,000) and the extensive exploration of the parameter space.” They are referring to Lamm equation modeling by Brautigam in which an estimate of 100 μM was obtained for the K_d a monomer – dimer equilibrium in which the highest SV AUC loading concentration 87 μM . In a static system, this predicts monomer and dimer concentrations at equilibrium of 47 and 20 μM . In contrast, for the monomer - dimer - tetramer equilibria proposed by the authors with K_d 's for the monomer -dimer and dimer - tetramer equilibria of 2.4 mM and 2.6 μM , respectively, the equilibrium concentrations of monomer, dimer and tetramer at a loading concentration of 107 μM would 91, 3 and 5 μM respectively. Thus, the comparison to Brautigam is not particularly convincing.

The large data set argument is somewhat specious. For example, there are a lot more SV data points with interference

optics compared to absorbance optics because of the data density and smaller minimum scanning interval. However, interference optics generally do not provide higher sensitivity or better estimates of fitting parameters than absorbance optics.

The integration of the manuscript with the Supplement lacks coherence. In the Supplement, the authors state “to overcome low solubility, we engineered a new CTD variant by introducing charged, glutamic acid residues at serine positions (S->E) in the disordered flanking regions of CTD ...” and that they “designed two obligate dimers with cross-linking site closer to CR at position S305 and S317 (S->E S305C and S->E S317C)”, a “the helix-breaking mutant, S->E A326P S305C dimer” and they “formed disulfide cross-links (S-S) between engineered cysteine residues at S273C”. The S273C mutant evidently is S->E S273C described in Supporting Fig. S9. SV-AUC experiments are described for these variants in the Supplement, but not in the manuscript.

Minor comments

The authors state SV experiments were conducted at 7.20 cm, which is not a meaningful statement in AUC. The figures show that SV data are analyzed at radial positions ranging from ~6.2 to 7.1 cm.

The authors state in the manuscript “for the non-self-associating species, linear plots describing the dependence of the sedimentation coefficient with concentration were generated in SigmaPlot (grafiti.com)”. They are referring to a figure in the Supplement, which is not stated in the manuscript. Additionally, the plots are not linear, possibly because they are semi-log plots or because fitting polynomials of degree greater than 1 is linear regression.

RESPONSE TO REVIEWER COMMENTS:

We thank the reviewers for their helpful comments, which we have addressed extensively through additional experiments, simulations, and analyses, as well as substantial revisions to the manuscript.

First, we provide direct evidence for the presence of tetrameric multimers of WT CTD through extensive new AUC analysis, which was a key concern raised by the reviewers.

Second, we emphasize the need for an integrative approach to overcome the significant challenges in structurally characterizing WT CTD multimers observed by AUC. We also clarify the rationale and significance of each designed CTD construct and experiment used to support our structural model of CTD helical multimers.

Third, we revised our MD simulation stability metric, replacing inter-dRMS with dRMS, which offers a more global assessment of structural stability beyond intermolecular contact loss. This analysis revealed that only one tetrameric model (Tet-1) was highly stable across all multimeric structures studied, and remained stable in the new full-length CTD simulations.

Finally, we incorporated new in-cell experiments and conducted additional analyses of the effect of a library of TDP-43 variants on cellular growth restriction data already available in the literature. These results demonstrate a direct correspondence between full-length TDP-43 cellular behavior, specifically its nuclear retention and toxicity, and the CTD variants we studied *in vitro*.

We are confident that these new data strengthen our conclusions and provide robust support for our NMR and computational findings.

Reviewer #1 (Remarks to the Author):

The manuscript by Fawzi et al. investigated the atomic structural details of TDP-43 CR helix-mediated assembly utilizing an integrative modeling strategy including NMR spectroscopy, SV-AUC, AF2-Multimer and all-atom molecular dynamics simulations. They found that the native state of TDP-43 CR is α -helical under physiological conditions, and helix-mediated interactions are linked to phase separation of C-terminal domain. The simulation results showed that TDP-43 CR assembly are stabilized by a methionine-rich core with specific contributions from a tryptophan/leucine pair. Overall, this work is carefully designed and executed. The manuscript is well written and the conclusions are supported by data.

We sincerely thank the reviewer for their careful evaluation of our work.

Some additional comments are listed below.

- I. In the method section, the simulation setup details including systems, number of simulations, number of atoms are suggested to be summarized. It is also necessary to give the reason for choosing Amber03ws force field for protein structure.

We have used AMBER03ws protein force field in combination with TIP4P2005 water model, both of which have been shown to accurately capture global dimensions and local properties such as transient secondary structure in intrinsically disordered proteins (Robustelli et al., *PNAS*, 2018; Zerze et al., *J Phys Chem Lett.*, 2019), including in TDP-43 CTD (Conicella et al., *Structure*, 2016, Conicella and Dignon et al., *PNAS*, 2020). Specifically, for the TDP-43 CTD, we have established the suitability of this force field in our previous studies by directly comparing simulation results with NMR observables. Single chain simulations of the TDP-43 CTD fragment (aa: 310-350), performed using a well-tempered metadynamics approach (Deighan et al., *JCTC*, 2012) with the AMBER03ws force-field, accurately reproduced the NMR-derived secondary structure chemical shifts and experimental $^3J_{\text{HNNH}\alpha}$ scalar coupling constants (Conicella et al., *Structure*, 2016). In addition, this force field was successfully applied in dimer simulations of TDP-43₃₁₀₋₃₅₀ using well-tempered metadynamics, capturing helical stabilization of TDP-43 CR through self-interaction, as observed by NMR ^{15}N $\Delta\delta$ quantification (Conicella and Dignon et al., *PNAS*, 2020).

In this study, we also performed 45 μs long unbiased simulations of TDP-43₃₁₀₋₃₅₀ and compared the resulting ensemble to experimental NMR $^3J_{\text{HNNH}\alpha}$ scalar coupling constants and chemical shifts reported in BMRB ID: 26823. The $^3J_{\text{HNNH}\alpha}$ scalar coupling constants were computed from the simulation trajectory using MDTraj, following the Bax2007 model (Vogeli et al., *JACS*, 2007). The root mean square deviation (RMSD) relative to experimental data was below 0.6 Hz, which falls within the expected margin of prediction error and shows strong agreement (**Fig. S15A**) (Vogeli et al., *JACS*, 2007). For residues 321–330, the simulated coupling constants averaged 4.87 ± 0.38 Hz, supporting the presence of transient α -helical structure (Smith et al., *JMB*, 1996).

Next, we computed ^{13}C chemical shifts ($\delta C_{\alpha/\beta}$) using the SPARTA+ algorithm (Shen et al., *Biomol. NMR*, 2010). The RMSD between experimental and simulated C_{α} shifts was 0.5 ppm, and for C_{β} shifts, 0.38 ppm. Both values are within the expected prediction error and demonstrate good

consistency with experimental data. To directly assess helicity, we analyzed per-residue secondary chemical shift differences ($\Delta\delta C_\alpha - \Delta\delta C_\beta$ shifts), as detailed previously (Mohanty et al., *PNAS*, 2023). This analysis also showed strong agreement with experimental results, particularly for residues 320–330, which exhibited positive values relative to random coil reference values, suggesting the presence of helical structure (**Fig. S15B**).

Furthermore, the per-residue helix fraction from the simulation, calculated using DSSP (Kabsch and Sander, *Biopolymers*, 1983), showed reasonable agreement with experimental per-residue helix fraction computed based on chemical shifts using $\delta 2D$ (Camilloni et al., *Biochemistry*, 2012) (**Fig. S15C**).

Based on these observations, we selected AMBER03ws for our study, and we have now explained the rationale for this selection in the Methods section:

“We selected the AMBER03ws protein force field⁸⁶ for its proven ability to accurately capture both the global dimensions and local structural features of various intrinsically disordered proteins, including the TDP-43 CTD²⁷. Specifically, AMBER03ws has been shown to reliably reproduce NMR-derived helical propensities in our previous simulations of a subsegment of the TDP-43 CTD (amino acids 310–350) used in this study^{5,16}. Hence, we conducted 45 μ s-long unbiased simulations of TDP-43_{310–350} using this force field and found good agreement between the simulated and experimental NMR observables, including $^3J_{\text{HNH}\alpha}$ scalar coupling constants and chemical shifts, as well as the simulated and NMR-derived helix fractions¹⁶ (**Fig. S15**). Based on these results, we selected AMBER03ws to simulate the structural ensembles of the higher-order assemblies of TDP-43_{310–350} presented in this study.”

We have now included a table in the Methods section summarizing the details of the simulation setup.

Table S3. Details of the simulation setup.

System name	Box size (nm)	# of protein atoms	# of independent simulations	duration of each simulation (μs)	aggregate duration (μs)
Monomer	8.25 – 8.94	562	5	7.91 – 9.59	45.07
Dimer (AFv2.2)	8.00	1124	5	8.53 – 10.59	101.42
Dimer (AFv2.3)	9.77 – 10.66	1124	13	3.77 – 4.06	
Trimer	8.83 – 9.91	1686	8	4.37 – 5.34	40.97
Tetramer (AFv2.2)	8.98	2248	2	1.40	2.80
Tetramer (AFv2.3)	8.92 – 9.63	2248	5	3.97 – 5.00	23.40
Hexamer	8.80 – 9.42	3372	5	4.59 – 5.30	25.69
Octamer	9.23 – 9.82	4496	4	4.89- 5.36	20.28
CTD full-length tetramer	18.00	7692	5	0.30	1.50

2. In Figure 5B, it would be easier for readers to understand the plot of structural stability if different size of oligomers are labeled.

We agree with the reviewer’s suggestion that using distinct colors in Figure 5B for different multimer sizes would improve clarity. Accordingly, we have updated Figure 5B to reflect this change. Additionally, we have included the distance root-mean-square deviation (dRMS) analysis for each multimer size separately in Figure S11A.

We also want to highlight that we now use the dRMS of all atom-pair distances, rather than the intermolecular dRMS (inter-dRMS), to assess the structural stability of AlphaFold-predicted models in our MD simulations. The reason is that inter-dRMS focuses solely on changes in intermolecular contacts and may fail to capture structural destabilization within individual monomers. In contrast, dRMS considers both intra- and intermolecular changes, providing a more comprehensive assessment of overall structural stability.

Figure 5. B. AF2-Multimer predicted multimer structures are then fed to run unbiased AAMD simulations ($\sim 4\text{-}5\ \mu\text{s}$ each) and structural stability is analyzed by computing dRMS of structures from AAMD simulation trajectories. dRMS of all heavy atoms (excluding hydrogen atoms) for residues from 320 to 341 as a function of time with respect to initial conformations are shown. The dRMS values of the trimers are represented in dark green, tetramers in orange (with Tet-1 and Tet-2 specifically highlighted in red and blue, respectively), hexamers in magenta, and octamers in cyan. (See Fig. S11A for more details).

Our new dRMS analysis revealed that only one tetrameric model (Tet-1) was highly stable among all multimeric structures studied, and we have now incorporated this finding into the text as follows:

“Interestingly, all trimer (eight), hexamer (five), and octamer (four) models were found to be less stable, exhibiting large structural changes from their initial conformations (Fig. 5B, S11A). Notably, among the tetramer structures, one model (which we named Tet-1), with four-fold symmetry where adjacent helices run antiparallel, exhibited greater stability during the simulation (Fig. 5B). To further test Tet-1’s stability, we performed 50 independent AAMD simulations (each 100 ns) of Tet-1, along with Tet-2, the second most stable tetramer that has a distinct helical arrangement and, unlike Tet-1, quickly loses its symmetry (Fig. 5B). The dRMS distributions from these simulations further confirmed the higher stability of Tet-1 (Fig. 5C). Collectively, AAMD simulations of AF-predicted multimers reveal a stable tetramer structure, suggesting that the dynamic nature of CR self-interactions may adopt stable tetramer configurations, in agreement with AUC results.”

Supporting Figure S11. A. Distance root-mean-square (dRMS) analysis of multimeric models computed from the AAMD simulations ($\sim 4\text{--}5\ \mu\text{s}$ each) of AlphaFold-Multimer predictions. dRMS of all heavy atoms (excluding hydrogen atoms) for residues from 320 to 341 as a function of time with respect to initial conformations are shown.

3. The authors performed 5- μs AAMD simulation for each system. However, the duration of some trajectories are less than 5 μs in Figure S8A and S10C.

We apologize for the confusion regarding the duration of the AAMD simulations for higher-order multimers ranging from trimers to octamers. Our goal for these simulations was to assess structural stability by running them for approximately 4 to 5 μs . Simulations were run for at least 4 μs (Table S3), except for two simulations initiated using tetramer models from AF2-Multimer v2.2 that were found to be highly unstable within the first microsecond, with dRMS values exceeding 1 nm. As a result, we decided not to continue those two simulations.

To avoid any confusion, we have now revised the text as follows:

“Hence, we employed AAMD simulations ($\sim 4\text{--}5\ \mu\text{s}$ in duration, or until a significant structural deviation was observed, defined as $dRMS > 1\ \text{nm}$) to directly assess the structural stability of these multimers and to explore whether AAMD could result in refined interhelical contacts that stabilize the assembled states (Fig. 5B).”

4. Previous computational studies (10.1021/acs.jcim.3c00513; 10.1021/acs.jcim.4c00943) have investigated the dimerization of TDP-43 low-complexity domain and reported the conformational transition from helices to β -sheets, which may be helpful for the discussion.

We thank the reviewer for bringing to our attention two relevant computational studies that performed dimeric simulations of the full CTD and its subsegment (residues 311–360). Li et al. (*JCIM*, 2023) demonstrated that the ALS-associated mutation G335D may induce helix-to- β -sheet transitions. This supports our point in the Discussion that the CR must convert to β -sheet aggregates to form pathogenic fibrils, suggesting that pharmacologically stabilizing helical assemblies could be a therapeutic avenue. We have now cited this study accordingly:

“Given that the CR must convert to β -sheet aggregates to form pathogenic fibrils^{9,10,73,74}, an intriguing therapeutic avenue may be to pharmacologically stabilize the helical assembly observed here to prevent conversion of TDP-43 CTD into β -sheet aggregates.”

Similar to what we observe in the Tet-1 simulations, Tang et al. (*JCIM*, 2024) observed a propensity for the flanking IDRs to form transient β -sheets, though they did not report helix-to- β -sheet transitions. We have also added a citation to this paper in our Discussion in the following sentence:

“Consistent with our findings, previous studies on TDP-43 CTD dimerization also reveal transient intermolecular β -sheet formation in flanking disordered regions.⁷⁸”

Reviewer #2 (Remarks to the Author):

We appreciate the reviewer's thoughtful comments on the experimental design of CTD mutants used primarily in the analytical ultracentrifugation experiments. Before we address the individual questions below, we want to emphasize that we have thoroughly revised the analyses and text to address these concerns. We also apologize that, due to an error during manuscript preparation, the originally submitted version of the sedimentation velocity AUC methods section was inadvertently replaced with a shorter, draft/template text. We have now included the correct SV-AUC methods section, as shown below:

“Analytical ultracentrifugation

Sedimentation velocity experiments were conducted at 25°C and 50,000 rpm (201,600 x g at 7.20 cm) on a Beckman Coulter ProteomeLab XL-I or Beckman Optima XL-A analytical ultracentrifuge following standard protocols⁸³. Frozen protein stocks in 20 mM MES pH 6.1 and 8 M urea were thawed and diluted to 15 mL in a Falcon tube. Buffer was exchanged using a 10 kDa Amicon filter at 3,000 x g and 4°C. The material was washed three times until the urea concentration was below 20 mM and the protein concentration was approximately 100 µM. To avoid issues with sample aggregation, CTD S273C S-S was prepared such that the final urea concentration was ~100 mM. The stock of WT CTD was received in 20 mM MES pH 6.1. Before the sedimentation experiment, samples were centrifuged at 18,000 x g and 4°C to clear aggregated material. Proteins were diluted with the final Amicon filtrate and studied at various loading concentrations in 3 mm or 12 mm pathlength cells. Sedimentation velocity scans were collected using absorbance optical systems at 280 nm, and data were analyzed in SEDFIT⁸⁴ in terms of a continuous c(s) distribution of Lamm equations. When necessary, early scans were not used for model fitting to remove contributions from large aggregates. Solution densities ρ and solution viscosities η were measured at 20°C on an Anton Paar DMA 5000 density meter and Anton Paar AMVn rolling ball viscometer, respectively, and corrected to the experimental temperature. The partial specific volumes of the peptides were calculated in SEDNTERP⁸⁵ and corrected for isotopic substitution. Experimental sedimentation coefficients, s were corrected to standard conditions, $s_{20,w}$. To determine the affinities for self-association, the c(s) distributions were integrated to get the weighted-average sedimentation coefficients and sample concentration. The resulting isotherms were modeled in SEDPHAT⁸⁰. For WT CTD, sedimentation velocity data at two concentrations were modeled directly using Lamm equations to describe monomer-dimer-tetramer self-association and other alternative models. Similarly, in the case of S→E S317C, sedimentation velocity data at three concentrations were modeled directly using Lamm equations describing the dimer-tetramer self-association. Here, data were plotted in GUSS⁸¹. All concentrations are in dimer subunits for cross-linked dimers. For the non-self-associating species, linear plots describing the dependence of the sedimentation coefficient with concentration were generated in SigmaPlot (grafiti.com).”

Major comments

The authors developed a model for the oligomerization and initial stages of pathogenic aggregation of TAR DNA binding protein-43 (TDP-43) by structure/function analysis of the C-terminal domain (CTD) of TDP-43.

The study largely consists of the analysis of wild-type and mutant CTD proteins. The mutant proteins contained serine to glutamic acid mutations (S->E) to preserve solubility. CTD proteins were His-tagged for purification, followed by removal the tags and his-cleaving TEV protease. They were subjected to denaturing conditions in 500 mM GdnHCl during purification and exchanged into 8M urea for storage. It is not clear if, when or how urea was removed before subsequent analysis. What was the solvent used in the experiments? The native oligomeric state of these proteins isolated and stored under these harshly denaturing is open to question and seems to be a major flaw in the study.

We thank the reviewer for pointing out the lack of description. In phase separation and NMR experiments, urea is removed by Zeba spin desalting columns via buffer exchange into 20 mM MES pH 6.1 with minimal residual urea (we estimate < 20 mM, which is far below any known activity). We established this protocol in our studies of the TDP-43 CTD (Conicella et al., *Structure*, 2016) and continued using it in Conicella and Dignon et al., *PNAS*, 2020, as well as Mohanty et al., *PNAS*, 2023.

The SV-AUC methods section now also describes in detail how the urea was removed for these samples. Briefly, sample preparation involves buffer exchange with 20 mM MES pH 6.1 using Amicon centrifugal filters at 4 °C. An account of both the urea and protein concentrations was kept, as described, and the remaining urea is ≤ 20 mM except for the highly aggregation prone CTD 273C (in the WT background) which had ~ 100 mM residual urea. SV-AUC and NMR experiments were carried out in 20 mM MES pH 6.1 unless otherwise noted, in which case buffer exchange was carried out with the appropriate solvent.

While the reviewer raises a valid concern regarding the possible effects of the denaturing storage buffer, neither AUC nor NMR indicated anything amiss with the soluble material observed in the experiments. Of course, this is an aggregation-prone region and as noted in the AUC methods section, some aggregated material was always present following buffer exchange. While such aggregates would be invisible by NMR, they require removal by centrifugation at 18,000 x g for AUC studies.

Our extensive work with TDP-43 CTD, both as a monomer and as cross-linked dimers, has shown that this buffer retains the native partially helical state of this region. We stress that these are the storage conditions we have established and used for our work since 2016, and many others in the field have adopted these approaches. Indeed, although the native oligomeric state of the protein is disrupted in 8M urea, its helicity and helix-helix interactions return after removal of the denaturant, as demonstrated by our previous papers. Given the tight correspondence between the effect of mutations in the CR on the biophysical behavior of samples of the CTD prepared this way, on the behavior of samples of the full-length protein that were never denatured due to fusion with our C-terminal MBP tag (Wang and Conicella et al., *EMBO J.*, 2018; Hallegger et al., *Cell*, 2021), and on RNA processing of the TDP-43-binding transcriptome (Hallegger et al., *Cell*, 2021), there is significant evidence already existing to demonstrate that the native CR form is reconstituted after urea removal.

Fig. 3A shows SV AUC $c(s)$ distributions as a function of loading concentration of WT CTD. There is a major 1.5 S species and at the highest loading concentration, 107 μM , a minor 2.9 S species. From these results, the authors propose that there is a monomer-dimer equilibrium that possibly proceeds to higher order oligomerization. However, the small 2.9 $c(s)$ peak shown in Fig. 3A does not strongly support a monomer-dimer equilibrium of wild-type CTD, or at least one with a dissociation constant of less than $\sim 100 \mu\text{M}$.

We thank the reviewer for raising this important issue. We have now conducted an extensive analysis of WT CTD SV-AUC data, which provides strong evidence for a monomer–dimer–tetramer equilibrium of WT CTD, as summarized below.

In SV-AUC experiments, WT CTD at the lowest concentration of 17 μM shows a single species at 1.53 S with an estimated mass of 14.0 kDa, indicative of a monomer. As the concentration increases, the sedimentation coefficient of the monomer increases slightly, suggesting weak self-association and a negative second-order virial coefficient. At the highest concentration studied (107 μM , determined by integrating the absorbance $c(s)$ profile), faster-sedimenting species emerge. Notably, the s -values for these species differ between the absorbance and interference data (**Response Figure 1**).

Response Figure 1. The absorbance (left) and interference (right) $c(s)$ distributions for WT CTD obtained from SV-AUC experiments. Both absorbance and interference data were collected within the same SV experiment.

Since the sedimentation coefficient scales with the two-thirds power of molar mass, a dimer (assuming similar shape or frictional ratio as the monomer) would be expected to sediment approximately 1.58 times faster than a monomer. However, the observed sedimentation coefficient for the ‘complex’ is larger than expected for a dimer. Taken together, these findings suggest that CTD undergoes self-association, forming multiple species with weak affinities. Weak affinities correspond to complexes with fast k_{off} values, causing the complexes to dissociate and exchange

with free species during sedimentation. As a result, $c(s)$ profiles can be counterintuitive – the sedimentation coefficients do not correspond to discrete species but rather to mixtures of exchanging species and complexes, known as reaction boundaries. Additionally, the best-fit frictional ratios (used to estimate molar masses) incorporate contributions to boundary broadening due to chemical exchange. The extent of exchange depends on the absorption extinction coefficient or interference signal increment, which explains why different profiles may be observed for the same material using absorbance and interference detection systems.

Another notable observation is that at 58 μM , no evidence of faster sedimenting material is detected, whereas at 107 μM , a faster sedimenting oligomer/reaction boundary emerges, accounting for 15% of the absorbance signal. This rapid change suggests a cooperative event.

To best interpret the data, we reasoned that direct Lamm equation modeling of the SV-AUC scans would be the most appropriate approach. Therefore, we performed direct Lamm equation modeling on WT CTD SV-AUC experiments at 58 and 107 μM using absorbance data.

We tested “two-state” (monomer-dimer, monomer-trimer, monomer-tetramer) and “three-state” (monomer-dimer-trimer, monomer-dimer-tetramer) models. All two state models and the monomer-dimer-trimer model result in poor fits and/or physically unreasonable sedimentation coefficients. However, the monomer-dimer-tetramer model provided excellent fits with sedimentation coefficients of 1.55 S, 2.34 S, and 3.51 S, consistent with CTD monomer, dimer, and tetramer (**Figure 3B**).

The best fit yielded dissociation constants: $K_{d,\text{mon-dim}}$ of 2.0 ± 0.5 mM and a dimer–tetramer $K_{d,\text{dim-tet}}$ of 2.9 ± 0.4 μM , indicating a metastable dimer that cooperatively forms a tetramer (1000 \times tighter dimer-tetramer affinity compared to monomer-dimer). We note that this is quite reasonable, as there are systems exhibiting similar behavior where monomer-dimer equilibrium has weak affinity, but dimer-tetramer and tetramer-octamer oligomerization steps show progressively stronger affinities (Kotler et al., *PNAS*, 2019; Ahn et al., *Commun Biol.*, 2023).

Mass action predicts monomer, dimer, and tetramer concentrations (in monomer units) as follows:

17 μM : 16.7 μM monomer, 0.26 μM dimer, 0.02 μM tetramer – Experimental data are consistent with a WT CTD monomer. Trace amounts of oligomers shift the monomer’s s -value slightly.

58 μM : 52.8 μM monomer, 2.6 μM dimer, 2.4 μM tetramer – Discrete dimer and tetramer species are not observed in the $c(s)$ distribution because of the weak monomer-dimer affinity, fast k_{off} , and small proportions of oligomers. The s -value for the monomer is shifted slightly to faster values.

107 μM : 83.4 μM monomer, 6.9 μM dimer, 16.7 μM tetramer – A significant amount of tetramer is observed as a reaction boundary containing exchanging monomers, dimers, and tetramers.

Figure 3. TDP-43 CTD forms tetrameric multimers. **A.** Sedimentation velocity analytical ultracentrifugation (SV-AUC) of WT CTD as a function of protein concentration. SV-AUC data were analyzed as a continuous $c(s)$ distribution of sedimenting species suggesting the presence of species larger than dimer. Data were collected in 12 mm cells, except for WT CTD at 58 and 107 μ M, where 3 mm pathlength cells were used. **B.** Absorbance sedimentation data collected for WT CTD at (left) 58 μ M (3 mm pathlength cell) and (right) 107 μ M (3 mm pathlength cell) were analyzed globally in terms of a monomer-dimer-tetramer self-association using Lamm equation modeling, along with trace amounts of an aggregate. The analysis, carried out in SEDPHAT⁸⁰, returns $K_{d,mon-dim}$ of 2.0 ± 0.5 mM and a dimer-tetramer $K_{d,dim-tet}$ of 2.9 ± 0.4 μ M. Data were plotted in GUSSE⁸¹ and for clarity only every third scan and every third experimental data point are shown. Best-fits are represented by a solid line through the experimental points. A bitmap representation of the residuals, together with the combined residuals, are shown below each plot. **C.** Schematic of the monomer-dimer-tetramer equilibrium and dissociation constants.

We have updated the AUC section in the manuscript to address this issue as follows:

“At the lowest concentrations (17 μ M) where the protein remains monomeric, the best fit model shows a single species with sedimentation coefficient (corrected to standard conditions, water at 20°C), $s_{20,w}$, of 1.53 S and an estimated mass of 14.0 kDa, consistent with a monomer (Fig 3A). As the concentration is raised (33 μ M and 58 μ M), the apparent sedimentation coefficient increases slightly, consistent with weak self-association (Fig 3A). At the highest concentration studied, 107 μ M (a value obtained directly from the AUC data by integration of the absorbance $c(s)$ profile), the best fit model is notably different despite less than a two-fold increase in concentration, showing an

additional faster-sedimenting species (**Fig 3A**). Although a dimer would be expected to have an *s*-value approximately 1.58x that of the monomer, the *s*-value for this species is approximately 2x that of the monomer, suggestive of weak CTD self-association into multiple higher-order species, consistent with previous NMR data⁵. To properly account for the contribution of interchanging species to the sedimentation profiles, direct Lamm equation modeling was conducted. “Two state” monomer-dimer, monomer-trimer, or monomer-tetramer models and “three state” monomer-dimer-trimer models resulted in poor fits and/or physically unreasonable sedimentation coefficients. In contrast, excellent fits were found for a monomer-dimer-tetramer model, with sedimentation coefficients of 1.55 *S*, 2.34 *S*, and 3.51 *S* that scale as expected for CTD monomer, dimer, and tetramer (**Fig. 3B**). Using the data from the two highest concentrations simultaneously, the best fits yielded a monomer-dimer dissociation constant $K_{d,mon-dim}$ of 2.0 ± 0.5 mM and a dimer-tetramer $K_{d,dim-tet}$ of 2.9 ± 0.4 μ M. This indicates that the dimer is a weakly-binding intermediate, consistent with our simulated dimer ensemble, but that it cooperatively forms a tetramer, with 1000-fold tighter affinity for the dimer-tetramer transition than for monomer-dimer (**Fig. 3C**). At the 107 μ M condition, 6.6 μ M is predicted to be in the dimer form and 16.8 μ M in the tetramer form. Taken together, our SV-AUC data for WT CTD clearly demonstrate the presence of higher-order multimers (e.g., tetramers) larger than dimers.”

The authors proceed to state “to better characterize the assembled state, we experimentally trapped a dimeric state by forming a disulfide cross-link (S-S) between engineered cysteine residues at S273C”. This maneuver seems highly suspect – it may represent an artifice to create dimers and subsequent higher order oligomers with no physiologic or pathologic relevance. The rationale for making the mutations S273C, S305C and S317C is not presented. These mutations evidently are not associated disease states.

We agree with the reviewer that we did not sufficiently explain the rationale and the extensive prior support for the utility and relevance of creating cross-linked variants. Most importantly, we have extensively reorganized the manuscript to make clear the purpose of these variants: to help characterize the tetrameric states observed for the uncrosslinked wild-type form described above.

We have now added the rationale for the S273C, S305C, and S317C mutations in the manuscript and Supporting Information text. The CR states populated by the cross-linked dimers are explicitly shown to be higher-order multimers with direct physiological relevance. These variants are not related to disease, and we do not claim they have pathologic relevance or help define pathological states in these analyses. Finally, we clarify that we use these cross-linked forms to provide additional insight into the assembly stoichiometry and structural details, but the pivotal data are derived entirely from experiments performed on the uncrosslinked versions. Below is a detailed explanation:

As discussed above, our new analysis of SV-AUC data strongly supports a monomer-dimer-tetramer equilibrium for WT (uncrosslinked) CTD, where tetramers form cooperatively with a μ M K_d once weak dimer assemblies (with mM K_d) are formed. We have established the presence of higher-order multimers larger than dimers; however, structural characterization of these multimeric assemblies using the native construct remains challenging, as described in the manuscript as follows:

“Given the transient, dynamic nature of CR interactions and their small population, structural characterization of the CR helix-helix interaction is not amenable to x-ray crystallography or electron cryo-microscopy (cryoEM). Hence, we next sought to determine the structure of the assembled CR region of TDP-43 CTD using nuclear Overhauser enhancement (NOE) NMR. However, several features make CR multimerization an extremely challenging target. First, NOE experiments are often conducted at 1 mM but TDP-43 CTD samples at higher concentrations ($\geq 100 \mu\text{M}$) are not stable and convert to aggregates, preventing long NMR experiments. Second, the multimer is not fully populated at achievable conditions, reducing NOEs from intermolecular contacts. Third, exchange between the monomeric and multimeric forms broadens the resonances from the CR¹⁶. Fourth, the CR itself is aggregation prone, as a shorter construct that isolates the CR (aa: 310-360) resulted in essentially complete insolubility (data not shown), suggesting the flanking IDR regions contribute to the solubility. Hence, we explored mutants of TDP-43 CTD IDRs to further enhance solubility while keeping the CR and adjacent regions (310-350) unaltered to preserve CR multimerization.”

To characterize the structural details of CR multimers with NMR, reducing the effects of the above-mentioned challenges, we designed multiple mutants of TDP-43 CTD IDRs with the goal of enhancing IDR solubility while keeping the CR unaltered to preserve its helical multimerization. One strategy we employed to design CTD mutants for NMR characterization was to create disulfide cross-linked dimers. We reasoned that this approach would help to overcome the barrier posed by weak dimer assembly in WT CTD and shift the equilibrium toward higher-order multimerization, as observed through SV-AUC experiments of WT CTD.

We started with covalently cross-linking two CTDs using single cysteines engineered at position 273, which over 40 residues away from the CR, as we did in our previous studies (Conicella and Dignon et al., *PNAS*, 2020). Previously, we demonstrated that this variant (10 μM , in monomer units) exhibits chemical shift perturbations across all CR residue positions that mirror those observed for WT CTD at various concentrations (20 μM , 45 μM , and 90 μM), as shown in Figure 1 of Conicella and Dignon et al., *PNAS*, 2020. This suggests that WT CTD and the cross-linked dimer sample similar interaction modes and chemical environments. Furthermore, cross-linking at the C-terminal end of the CR (at S387C) produced a similar chemical shift pattern in the CR region (RMSD ~ 0.07 ppm), indicating that the chemical environment and thus the multimeric states sampled by the TDP-43 CR upon cross-linking is independent of the cross-linking position (Conicella and Dignon et al., *PNAS*, 2020).

[editorial note: figure redacted]

Figure 1 from Conicella and Dignon et al., PNAS, 2020. E) Schematic of cross-linked CTD variants. (**G**, Left) Select regions from ^1H - ^{15}N TROSY spectra of disulfide-linked homodimeric TDP-43 CTD S273C (red) show line broadening and large upfield ^{15}N chemical shift differences in 321–343 region compared to monomeric reduced S273C (blue), consistent with formation of structure. Chemical shifts for residues in the dimer state were assigned by extrapolation (dashed black line) from spectra of 20 μM (cyan), 45 μM (green), and 90 μM (orange) WT. (**G**, Right) Overlay of ^1H - ^{15}N spectra regions for homodimeric S273C and S387C are similar in the central 320–340 region.

AUC experiments for the S273C cross-linked variant showed sedimentation coefficients consistent with tetramers or higher-order multimers at increasing protein concentrations, supporting the cooperative assembly seen in WT CTD. However, the low solubility of S273C samples limits structural characterization of higher-order oligomers by NMR. To address this, we designed more soluble CTD variants by introducing charged glutamic acid residues at serine positions (S→E) in the disordered flanking regions of CTD, aiming to preserve the helical structure of the CR, as confirmed by the NMR spectral fingerprint of the monomeric S→E variant (**Fig. S6**). Yet, cross-linking at residue 273 in this soluble variant (S→E S273C) resulted in minimal self-assembly, with only small chemical shift perturbations (**Fig. S7**), likely due to electrostatic repulsion. Therefore, we designed two obligate dimers with the cross-link positioned closer to the CR, at residues S305 and S317 (S→E S305C and S→E S317C), reasoning that cooperative CR-CR interaction could be enhanced by doing so, thereby driving the equilibrium toward CR-mediated higher-order assembly. We confirmed that cross-linked dimers closer to CR preserve the CR's fundamental structural features via unbiased all-atom MD simulations ($\sim 45 \mu\text{s}$) of a dimeric CTD subregion (aa 310–350) cross-linked at 317C, which showed that both the WT and cross-linked dimers exhibit similar helix-helix contacts and helix propensity (**Fig. S8**).

We again emphasize that these variants are not associated with disease mutations but are used solely for technical purposes. This method is justified and provides crucial information on CTD multimerization, specifically revealing new structural details on helix-helix interactions within the conserved region.

Another important point is that we also characterized the structural details of CR multimers using the TDP-43 CTD 6F→A variant (without cross-linking), which replaces phenylalanines in the IDRs with alanine. We recently demonstrated that this mutation reduces CTD aggregation (Pantoja-Uceda et al., *PLOS Biol.*, 2021) while retaining phase separation and CR helix-helix interactions (Mohanty et al. *PNAS*, 2023). This variant also showed similar NOE patterns to those observed in the cross-linked CTD forms, suggesting that CR assembly occurs similarly in both contexts.

We have now added these explanations in the manuscript as follows:

“Hence, we explored mutants of TDP-43 CTD IDRs to further enhance solubility while keeping the CR and adjacent regions (310-350) unaltered to preserve CR multimerization. We recently demonstrated that an aggregation-reducing variant of TDP-43 CTD that replaces phenylalanines in the IDRs with alanine⁶² (TDP-43 CTD 6F→A) retains phase separation and CR helix-helix interaction at higher protein concentrations²⁰.”

*“To complement the structural data from TDP-43 CTD 6F→A, we also extensively characterized the assembly and NOE constraints of a series of cross-linked variants of TDP-43 (See Supporting Text, **Figs. S6-9**). Phosphorylation at the very C-terminal part of TDP-43 CTD (S403, S404, S409, S410) is a major feature of TDP-43 inclusions and placing aspartic acid residue substitutions mimicking phosphorylation at these sites can drive aggregation⁶⁴. In contrast, placing aspartic acid residues at 12 sites across the entire CTD appears to counteract pathological aggregation⁶⁵, enhance the liquidity and dynamic nature of TDP-43 condensates, while preserving nuclear import or RNA regulatory functions, similar to what we have observed for poly-phosphomimetic FUS⁶⁶. Similarly, here we made serine to glutamate (S→E) substitutions across the disordered flanking regions of CTD (See Table S1), which significantly enhanced TDP-43 solubility while preserving the helical structure in the CR structure as indicated by NMR (**Fig. S6**). When cross-linked to enhance interaction as we previously demonstrated for the wild-type protein⁵, these variants populate multimeric assemblies that depend on the helical region, as seen by AUC (**Fig. S7,9**). Using cross-linked forms created from 1:1 mixture of unlabeled and ¹³C/¹⁵N proteins, we again see a similar pattern of methyl NOEs as observed for TDP-43 CTD 6F→A (**Fig. S10**), suggesting that CR assembly is similar in both contexts where the TDP-43 CTD solubility is enhanced without altering the CR.”*

Supporting Figure S10. Intermolecular NOEs for cross-linked variants show key contacts for TDP-43 CR assembly. ^{13}C filtered-edited NOE-HSQC strips of S \rightarrow E S273C (A) and S305C (B) cross-linked dimers. Cross-linked dimers were formed by 1:1 mixture of $^{13}\text{C}/^{15}\text{N}$ labeled and unlabeled monomers in the presence of 1 mM copper(II) phenanthroline catalyst. Spectra were recorded at 47 °C using a dimer concentration of 70 μM in 20 mM MES pH 6.1 with 10% D_2O .

We have now added these explanations in the Supporting Information, as follows:

“We formed disulfide cross-links (S-S) between engineered cysteine residues at S273C (see Methods), as we previously used¹. This cross-link is more than 40 residues away from the CR and previously we showed it results in chemical shift perturbations at all CR residue positions that mirror those shown for WT CTD with increasing concentrations, just with higher magnitude, suggesting similar interaction modes sampled by the TDP-43 CR¹. At low concentrations (~5 μM), AUC experiments for the cross-linked variant show a sedimentation coefficient consistent with a dimer form. (Fig. S7A). At ~11 μM and above a species with sedimentation coefficients consistent with a tetramer or higher is also observed, consistent with cooperative assembly (Fig. S7A). However, at these concentrations, the samples show evidence of aggregation by AUC experiments, as we previously observed for NMR sample¹ and the low solubility of S273C cross-linked samples hinders further characterization of the multimeric state using these samples.

To overcome low solubility, we engineered a new CTD variant by introducing charged, glutamic acid residues at serine positions (S \rightarrow E) in the disordered flanking regions of CTD, mimicking how phosphorylation may counteract pathological aggregation, enhance the liquidity and dynamic nature of TDP-43 condensates, while preserving nuclear import or RNA regulatory functions.^{2,3} (See Table S1). This modification significantly enhanced solubility while preserving the helical structure in the CR structure, as indicated by NMR spectral fingerprint of the monomeric S \rightarrow E variant (Fig. S6), suggesting that this variant should not disrupt helical multimerization. However, placing the cross-link position again at residue 273 resulted in no apparent self-assembly and small chemical shift

perturbations (Fig. S7B,E), suggesting that electrostatic repulsion discourages self-interactions. Reasoning that cooperative CR-CR interaction could be enhanced by placing the cross-link position closer to the CR, we designed two obligate dimers with cross-linking site closer to CR at position S305 and S317 (S→E S305C and S→E S317C). To demonstrate that cross-linking closer to the CR does not alter the CR structure, we simulated the cross-linked dimer, TDP-43₃₁₀₋₃₅₀ S317C S-S, by mutating S317 to cysteine in each chain of AF2-Multimer dimer models using Chimera⁴ and adding a disulfide bond (S-S) between cysteine residues using the pdb2gmx -ss command. Simulations of the S317C cross-linked dimer (total time ~ 45 μs) show that per-residue helix fractions and pairwise intermolecular contacts for the cross-linked dimer ensemble remained similar to WT dimer (Fig. S8), suggesting that the fundamental features of CR remain intact.”

SV AUC of the S305C and S317C CTD mutants revealed the formation of higher order oligomers beyond the dimer. Modeling with to dimer-tetramer and dimer-tetramer-octamer equilibria was done by sw isotherm analysis and Lamm equation modeling. Although good fits to the models was obtained, it is not clear if equally good fits to alternative models would be obtained. This does not detract substantially from the results, which clearly reveal some sort of oligomerization or aggregation beyond the dimer. However, as noted above, for proteins that are engineered to be soluble with S → E mutations, stored in chaotropic buffers and artificially disulfide-linked to form dimers, the relevance of these higher order structures is questionable.

As noted above, we have extensively rewritten the manuscript to explain the rationale behind, as well as the significance of, each construct and experiment throughout – admittedly, it is complex. Yet to make progress on this incredibly important and challenging problem, we have also added text to explain the **necessity** of these complex approaches due to the incredible experimental challenges this protein presents.

Regarding concerns about chaotropic buffers, we have now improved the text to clearly describe that urea is removed, and we have extensively validated our procedures for creating “native-like” structural equilibria after removal of urea. Regarding the relevance of disulfide-linked dimers, as explained above, we provided the rationale for engineering soluble, cross-linked CTD variants to further study the tetramerization observed for the WT CTD in the absence of crosslinks and the relevance of the crosslinked species to the higher-order assemblies formed by the uncrosslinked CTD. The cross-linked species were confirmed to form multimers larger than dimers through extensive AUC analysis and to form similar patterns of intermolecular NOEs via NMR. In short, due to the weak monomer-dimer self-association of WT CTD and the solubility challenges of the native construct, we engineered disulfide-linked S→E mutant dimers to gain crucial structural insights into CR-mediated multimerization via NMR, which is not feasible for WT CTD.

Based on sedimentation velocity data, the S→E S305C and S→E S317C CTD mutants self-associate, and the S→E S273C and A326P S→E S305C CTD mutants do not, highlighting the advantages of SV-AUC when studying self-association. The simplest model is reversible dimerization, and such a model (referred to as dimer–tetramer) fits the data collected for the S317C CTD mutant well. The model is supported by the c(s) distribution (Fig. S7), which only shows dimers and tetramers, and the rigorous Lamm equation modeling (Fig. S9B). It is unlikely that higher-order species are present in the concentration range studied. Even though the s_w isotherm does not

populate higher concentrations, it returns the same dissociation constant as that obtained using the Lamm equation modeling, further justifying the dimer–tetramer self-association model used.

In the case of the S305C CTD mutant the $c(s)$ distributions show clear evidence for further self-association beyond a tetramer. We chose the dimer–tetramer–octamer self-association model as it is the simplest model that describes the data. The dimer–tetramer affinity is similar (within the error of the method) to that observed for the S317C CTD mutant, and the tetramer–octamer affinity highlights the cooperative nature of self-association. Based on the $c(s)$ distribution, it is unlikely that larger species exist within the studied concentration range. However, this does not rule out other models, such as dimer–tetramer–hexamer–octamer that would fit the data equally well but would be challenging to analyze due to overparameterization.

As noted by the reviewer, this does not detract substantially from the results, which clearly reveal some sort of multimerization beyond the dimer.

With explanations provided above, outlining the rationale behind the design of S→E mutations and cross-linked dimers, and clarifying that the storage buffer used in this and previous studies does not impact the multimeric state of TDP-43 CTD in the final samples, we hope we have addressed these concerns effectively.

The peak areas of $c(s)$ distributions seem inconsistent with the loading concentrations. In Fig. 3A there is a ~6-fold increase in loading concentrations. Are the associated peak areas consistent with this? In Fig. 3B, are the $c(s)$ peak areas consistent with loading concentrations of 11 μM and 21 μM ?

We thank the reviewer for bringing this apparent discrepancy to our attention. Sedimentation experiments were carried in 12 mm or 3 mm pathlength cells depending on the sample concentration, in order to keep the measured absorbance in the linear range, and also based on availability. In Figure 3A, low concentration samples at 17 and 33 μM were loaded in 12 mm pathlength cells, whereas high concentration samples at 58 and 107 μM were studied in 3 mm pathlength cells. Similarly, in Figure S7A, the low concentration samples at 5.8 and 11 μM were loaded in 12 mm pathlength cells, whereas the highest concentration sample at 21 μM was studied in a 3 mm pathlength cell. We also note that the concentrations reported for all the sedimentation velocity experiments are those obtained by integration of the $c(s)$ distributions, and therefore self-consistent. To address this issue, we have modified the sedimentation velocity AUC figure legends to include the cell pathlengths used for sample analysis.

For nearly monodisperse systems, continuous $c(s)$ distribution analysis often produces an accurate estimate of the molar mass from combined estimates of the diffusion coefficient (expressed in terms of the frictional ratio, f/f_0) and sedimentation coefficient. Were the molar mass estimates of the 1.5 S species and the 2.9 S species consistent of monomeric and dimeric CTD? Fig. 3A legend states a molar mass of 31.5 kDa. Where did this number come from?

In Figure 3A, we show the $c(s)$ distributions for samples of WT CTD at different loading concentrations. At the lowest concentration of 17 μM , we observe a species at 1.53 S. Based on the

best-fit frictional ratio f/f_0 , the analysis returns a molar mass of 14 kDa, consistent with a TDP-43 WT CTD monomer (expected mass is 14.89 kDa). The $c(s)$ distribution provides accurate molar mass estimates for non-interacting systems or for interacting systems with slow exchange. In the case of interacting systems with fast exchange, boundary broadening will have contributions from diffusion and the chemical equilibrium, resulting in smaller than expected frictional ratios and smaller mass estimates than expected. We have removed the mass estimate for the 2.9 S species that represents a reaction boundary for self-association. Instead, we have performed direct Lamm equation modeling on the WT CTD SV-AUC experiments at 58 and 107 μM . Using the absorbance data, we obtain excellent fits for monomer-dimer-tetramer model with sedimentation coefficients of 1.55 S, 2.34 S, and 3.51 S, consistent with CTD monomer, dimer, and tetramer.

Minor comments

Rotor speeds evidently were 50000 rpm, not 5000 as stated in the manuscript.

We apologize for the confusion regarding the rotor speeds used in the SV AUC experiments (see our comment above that we used a template/incorrect segment of text due to an editing error). The corrected SV AUC methods section shows that a rotor speed of 50,000 rpm was used. The SV AUC rotor speed and temperature are no longer mentioned in the Figure legends.

The authors state the SV-AUC interference optical system was used, but only absorbance data are shown.

As noted in the corrected SV AUC methods section, both a Beckman Coulter ProteomeLab XL-I and Beckman Optima XL-A analytical ultracentrifuge were used, with the latter only providing absorbance data. The majority of experiments were carried out on the XL-A, and for this reason, we only have absorbance data presented. In the case analyzed on the XL-I, namely WT CTD in Figure 3A, the interference data return similar $c(s)$ profiles, but the s -values differ for faster-sedimenting species at 107 μM between the absorbance and interference data (**Rebuttal Figure 1**), as discussed above.

References are not given for SEDPHAT, Lamm equations modeling or GUSI.

We have now updated the methods section for AUC experiments and caption for Supporting Figure S9, including references for SEDPHAT, Lamm equations, and GUSI.

ALS is not defined in the abstract.

We have now defined ALS as amyotrophic lateral sclerosis in the abstract.

Regarding the statement "... the dynamic association of TDP-43 CTD ...", it is not clear what TDP-43 CTD is associating with.

We have now changed the statement to "... the dynamic self-association of TDP-43 CTD ...", to reflect we are dealing with the dynamic self-association of TDP-43 CTD.

The authors state in that SV AUC experiments “were conducted at the specified temperature”. Fig. 3 and Fig. S6 state SV AUC was done at 25 C. The temperature is not specified in Fig S7. If all experiments were at 25 C, please revise Methods to state that SV AUC experiments were conducted at 25 C.

We apologize for any confusion regarding the temperature used in the SV AUC experiments. To clarify, all experiments were conducted at 25°C, as stated in the revised (and original) SV-AUC methods section. The SV AUC rotor speed and temperature are no longer mentioned in the Figure legends.

Reviewer #3 (Remarks to the Author):

The manuscript by Rizuan et al. focuses on the role of the conserved region (aa 310-350) in TDP-43's low complexity region (LCD), in particular its structural details and role for TDP-43 assembly. For this, they integrate several biophysical and biochemical approaches as well as molecular dynamic simulations to investigate the structural rearrangement and interactions of the conserved region during higher order assembly. The study not only confirms the presence of an alpha-helix under various buffer conditions, but also pinpoints the importance of several residues for promoting self-assembly, in particular L330, W334 and methionines, while most polar residues appear to dampen phase separation. The significance of this study lies in the characterization of the conformational states of the alpha helix of TDP-43, highlighting its role in promoting oligomer formation, which might affect biophysical and functional activity of TDP-43.

Overall, the authors provide extensive biochemical and biophysical data supporting their findings. The characterization of the role of different residues in TDP-43 LCD can potentially provide important insight into the mechanism of TDP-43 self-assembly. Structural details of TDP-43's CR can so far only be obtained using the LCD, and biochemical experiments have been performed introducing disulfide bonds to promote oligomer formation, which is also a major limitation of the study and should be discussed accordingly by the authors (and to some extent addressed as mentioned below).

In summary, we recommend publication of this work in Nature Communications, provided that the following points are being addressed by the authors:

We thank the reviewer for their positive and encouraging feedback on our work. We appreciate their recognition of the extensive biochemical and biophysical data we have provided and the acknowledgment of the insights our study offers into the mechanisms of TDP-43 self-assembly.

Major points:

- I) the condensates presented in Figure 2B are very small and it is nearly impossible to see the described morphological differences described by the authors, such as reduced roundness. Please additionally provide zoomed-in images in this panel. Also, please provide scale bars to allow judgement of the condensate size.

We thank the reviewer for their valuable suggestion. In response to their comment, we have fully repeated the microscopy experiment to capture higher quality images that better show the characteristics of the droplets formed (Fig. 2B). We revised the sentence introducing the microscopy to emphasize that this is a qualitative metric complementing the phase separation quantification assays:

“Microscopy was used to qualitatively evaluate phase separation and whether droplets remained dynamic and liquid-like or transitioned to static aggregates.”

We also use these images to support the conclusions that W334A does not phase separate at these conditions:

“Highlighting the essential role of a single tryptophan in promoting CTD condensation, W334A shows the most profound effect on phase separation, resulting in no droplets at the studied conditions (**Fig. 2B, Fig. S1**).”

With superior microscopy, the potential for evaluating subtle changes in morphology emerges. So, we have revised the text for precision:

Glutamine to alanine mutations (Q327A, Q331A, Q343A) display the greatest enhancement of phase separation with c_{sat} approximately three-fold lower than WT c_{sat} . Microscopy of these variants after phase separation confirm phase separation into droplets. For Q327A and S333A, the droplets appear slightly less round, suggesting these variants may form more viscous or viscoelastic assemblies in addition to enhancing phase separation, while N319A robustly forms morphologically irregularly shaped assemblies (**Fig. 2B**).

Figure 2. B. DIC micrographs of CTD variants at 80 μ M in 150 mM NaCl. Scale bar, 10 μ m. All variants form round liquid-like droplets except N319A that forms morphologically irregular assemblies, Q327A and S333A that form slightly less round droplets, and W334A that does not form droplets at these conditions, highlighted with a blue box.

II) Due to the strong focus on structural analysis of the LCD, the study is limited with respect to the full length protein. To support their model for a role of the CR in TDP-43 self-assembly, the authors should support their findings by more cellular data in addition to the ones already provided in the SI. Here they should use key mutants identified in their structural analysis that either disrupt or enhance the alpha-helix (w.g. by comparing WT and A326P with at least L330A and W334A). Are those mutants also in context of the full-length protein in cells less/more prone to undergo higher order

assembly/phase separation? This could for example be done by studying their subcellular localization in response to stress or by applying size exclusion chromatography as previously shown by Dos Passos et al (PMID: 38422113) using their stable cell lines. Ideally, this should be performed in absence of endogenous TDP-43 (if possible) to avoid complex formation of mutant with wt TDP-43.

We appreciate the reviewer's concerns and we highlight here how our additional work here and our past work addresses these concerns. A core feature of our response is that we have added more cellular data.

The studies in Fig. 2H and S4 were conducted using established cellular models, stable and isogenic HEK293 cells expressing a single copy of HA-tagged full-length TDP-43. This group of cell lines include WT and mutant full-length TDP-43, including the substitutions that either disrupt or enhance the α -helix, listed by the reviewer. We also included a previously characterized A326P variant, which disrupts TDP-43 nuclear retention by decreasing higher-order macromolecular assembly, as suggested by our previous SEC data (Dos Passos et al., *PLoS Biol.*, 2024). We have added both nuclear localization microscopy and analysis of cellular growth restriction data of important variants (see below) to add additional characterization of the full-length protein in cells.

There is evidence from our (Ayala lab) prior work that the presence of endogenous TDP-43 does not significantly affect nuclear retention or influence nuclear-cytoplasmic localization (Dos Passos et al., *PLoS Biol.*, 2024 and Koehler et al., *Front Neurosci.*, 2022). This conclusion is based on our studies showing that the presence of endogenous TDP-43, which is predominantly nuclear under physiological conditions, does not alter the formation of large macromolecular complexes by exogenous HA-TDP-43. For example, the levels of endogenous TDP-43 in cells induced to express WT HA-TDP-43 are substantially lower (>60%) compared to non-induced cells (Koehler et al., *Front Neurosci.*, 2022). In contrast, mutations that decrease nuclear retention and increase cytoplasmic localization (e.g., A326P) form smaller complexes in cells (Dos Passos et al., *PLoS Biol.*, 2024), and these cells express higher levels of endogenous TDP-43 due to defects in autoregulation (**Rebuttal Fig. 2**). Therefore, the presence of endogenous TDP-43 does not significantly alter the behavior of TDP-43 variants in the assays used here because we previously observed a significant loss of TDP-43 nuclear retention when endogenous TDP-43 levels were greater - compared to cell lines expressing mutants that do not significantly alter macromolecular complex formation and nuclear retention (Dos Passos et al., *PLoS Biol.*, 2024 and Koehler et al., *Front Neurosci.*, 2022).

Rebuttal Figure 2. Disruption of autoregulatory function by A326P compared to WT TDP-43 in HEK293 cells. This corresponds to greater endogenous TDP-43 expression in A326P cells compared to WT (not shown).

To make clear these aspects, we have made changes to the text:

"Our previous studies of macromolecular complexes including exogenous HA-TDP-43 in HEK293^{HA-TDP-43} cells suggest that endogenous TDP-43 does not significantly contribute to increased nuclear retention. WT HA-TDP-43 expression in these cells decreases endogenous TDP-43 levels by approximately 60% due to autoregulation, which serves as a negative feedback loop controlling protein synthesis⁶. WT HA-TDP-43 exhibits similar nuclear retention and macromolecular size distribution as endogenous TDP-43⁶⁰. In contrast, expression of A326P does not significantly reduce endogenous TDP-43 levels due to defects in autoregulation³. Despite the presence of endogenous TDP-43, A326P shows lower nuclear retention as well as decreased formation of large macromolecular complexes, compared to WT HA-TDP-43⁶⁰. Overall, these findings indicate that the structures populated upon phase separation in vitro as detected by NMR are important for cellular TDP-43 assembly."

III) Regarding Figure S3, please indicate the difference between upper and lower blot panel in A and B, respectively, as the signal distribution in the different fractions for TDP-43-HA appears to be quite variable. Also, why is endogenous TDP-43 not visible in the upper blot panel? Which blots were used for the quantification? Also, are those mutants in intact cells localized to the nucleus or already cytoplasmically mislocalised (i.e. more than WT as this is an overexpression system)? The provided assay mainly addresses the extractability/ nuclear retention of TDP-43 upon cell lysis – this should also be clear from the experimental description. The authors should therefore only use “nuclear retention” and not refer to “nuclear localization” or “nuclear depletion” (unless specific mutants show a significant cytoplasmic mislocalization in intact cells).

We apologize for the lack of description in the legend of Figure S3 (current Figure S4). This is now clarified in the updated figure caption for Figure S4 (see below). The differences between the top and bottom panels in Figs. S4A, B are the antibodies used to probe the different blots. These top panels show representative images of membranes probed with antibodies to detect Lamin A/C (as

a control for the nuclear fraction) and hemagglutinin (HA)-tagged TDP-43. Thus, endogenous TDP-43 was not detected. The bottom blots were probed with antibodies detecting tubulin (as a cytoplasmic control) and total TDP-43, which detected both HA-tagged and endogenous TDP-43. Quantification of TDP-43 was conducted by measuring HA-TDP-43 levels. At least five independent replicates were performed.

The reviewer also raises an important point: although our established (Dos Passos et al., *PLOS Biol.*, 2024) biochemical fractionation assay shows that the CR is essential to nuclear retention, cellular localization may not directly correspond to nuclear retention (i.e., TDP-43 variants may leave the nucleus during the biochemical fractionation process used to measure retention). This would still be a robust assay and an important finding, but it may not represent localization in live cells.

To address whether the nuclear/cytoplasmic ratio of HA-TDP-43 corresponds to the observations with biochemical fractionation assays, we performed immunofluorescence-based quantification. Stable and isogenic HEK293 cells expressing a single copy of HA-tagged full-length WT or mutant TDP-43 were fixed and probed using HA antibody and stained with DAPI and wheat germ agglutinin to define the cytoplasmic area. Representative cell images, including heatmap of fluorescence intensity, are shown in the new Fig. S4C. We quantified nuclear and cytoplasmic intensity using standard protocols in ImageJ. The resulting nuclear/cytoplasmic ratios, summarized in Fig. S4D, reflect data from six independent replicates with more than 300 cells counted for each TDP-43 variant. We found that S333A, which enhances phase separation and nuclear retention, showed a trend toward increased nuclear retention. Conversely, A326P, the mutation that most strongly disrupts phase separation, significantly decreases the N/C ratio, enhancing cytoplasmic distribution. W334A and L330A, which reduce phase separation more modestly, show a trend toward lower N/C ratios, but these differences were not statistically significant.

We note that distinguishing nuclear to cytoplasmic ratio by image quantification resulted in large uncertainties, as indicated by the broad distributions observed for each variant. This suggests that only very large changes in localization can be robustly detected. Biochemical fractionation, therefore, offers greater sensitivity for quantifying protein levels compared to immunocytochemistry. One reason may be the incomplete accessibility of epitopes due to protein conformation or complex formation. However, we agree with the reviewer that biochemical fractionation may accentuate disruptions in nuclear retention if proteins exit the nucleus during fractionation. Accordingly, we now refer to these differences as changes in nuclear retention rather than nuclear localization.

Supporting Figure S4. TDP-43 CR helix assembly affects TDP-43 nuclear retention. A-B. Representative immunoblot of total, cytoplasmic and nuclear fractions of HEK293HA-TDP-43 cells from the individual experiments that were performed together. Lamin A/C and tubulin were used as nuclear and cytoplasmic controls, respectively, ensuring proper separation and integrity of each subcellular compartment. Total TDP-43 protein levels were assessed by probing the membrane with an antibody against TDP-43 (bottom panel), while (HA)-tagged TDP-43 was specifically detected using an anti-HA antibody. **C.** Quantification of TDP-43 nuclear to cytoplasmic ratio using immunofluorescence analysis. Stable and isogenic HEK293 cells expressing a single copy of HA-tagged WT or mutant full-length in HEK293^{HA-TDP-43} probed with anti-HA antibody and stained with DAPI and wheat germ agglutinin (WGA) to define the nucleus and the cytoplasmic area, respectively (top: immunofluorescence, bottom: pseudo-color look-up table). The intensity histogram for each image was independently maximized across the full range. Scale bar, 10 μ m. **D.** Nuclear and cytoplasmic HA-TDP-43 intensity was calculated using standard protocols in ImageJ. Graph shows the mean and SD of HA-TDP-43 nuclear/cytoplasmic ratio from \geq six individual replicates, where >300 individual cells

were quantified in total. Statistical analysis was assessed using the non-parametric Mann-Whitney U test, ***p = 0.0003.

Changes made to the main text:

“We also performed immunofluorescence microscopy to compare the subcellular localization of HA-tagged TDP-43 variants in intact cells (Fig. S4C,D). Consistent with our fractionation assays and previous report⁶⁰, A326P significantly decreases the N/C ratio. W334A and L330A show a trend toward lower N/C ratios, while S333A shows a moderate increase in TDP-43 nuclear localization compared to WT; however, these differences do not reach statistical significance (Fig. S4D). Differences in the extent of nuclear localization loss caused by mutations disrupting phase separation, more pronounced in nuclear-cytoplasmic fractionation than in immunofluorescence, may reflect leakage of HA-TDP-43 during biochemical nuclear isolation.”

Minor points:

1. The authors start their work by showing the conservation of the CR in vertebrates – yet they only show 2 species (zebrafish and homo sapiens). A comparison with additional species would further strengthen this point.

We thank the reviewer for their suggestion. In our recent paper (Mohanty et al., *PNAS*, 2023), we performed sequence alignment analysis of the TDP-43 CTD across its 93 homologs, showing that the CR residues (aa: 319-341) are 100% conserved among these vertebrates. We have now included this reference to further support our point in the revised first sentences of the second paragraph in the Introduction as follows:

“The CR in CTD is uniquely well-conserved (showing 100% identity) throughout vertebrates, in contrast to the flanking IDRs^{16,21}.”

2. The authors start their result section with referring to “recent work using hydrogen-bond disrupting variants” that suggested a β -sheet conformation for the CR of TDP-43 – yet they fail to provide any references.

We apologize for the oversight and have now added the missing reference for the referred paper.

3. There appears to be a typo in the sentence “Expanding from our previous results for the 5M->A variant within CR, which collectively impaired the CTD phase separation, each single methionine substitution within CR (M322A, M333A, M336A, M337A, M339A) also reduce phase separation with approximately two-fold higher csat than WT (Fig. 2A)” -probably the author refers to M323A.

We have corrected the typo to accurately reflect M323A.

4. Sedimentation assay shown in Figure S1: Even though standard deviations of three replicates are mentioned in the figure legend, they appear to be lacking from the figure itself.

We have added standard deviation error bars to the sedimentation assay in Figure S1, as mentioned in the figure legend. However, the error bars were initially not visible because they were quite small

and hidden by the markers. This small magnitude of variability can also be observed in Figure 2A. In the updated version of Figure S1, we have now placed the error bars on top of the markers to improve their visibility.

Supporting Figure S1. Phase separation saturation concentrations of TDP-43 CTD and designed variants from alanine-scanning mutagenesis. A-C. Quantification of phase separation assay showing the protein remaining in the supernatant after phase separation for WT CTD and its single alanine substitution variants at non-alanine positions within CR and its adjacent residues measured from 0 to 300 mM NaCl from the individual experiments that were performed together. Standard deviations of three replicates are represented as error bars.

Did the author compare these results with other methods employed to characterize protein multimers/oligomers (e.g. DLS or are strongly assembly-promoting mutants measurable by SV-AUC)??

We thank the reviewer for their suggestion, which is similar to that of Reviewer 4. However, the complementary methods suggested by both reviewers do not work for our system, as explained below.

Dynamic light scattering (DLS) is not suitable in this case, as WT CTD samples inevitably contain trace amounts of aggregated material, which would dominate the signal. Techniques such as mass photometry are also not ideal due to the need to use high sample concentrations to form the assembly, well above the nanomolar range recommended for mass photometry. We considered static light scattering, but for composition gradient MALS (CG-MALS), which could in theory be used to fit a binding model (see our previous work, Wang et al., *EMBO J*, 2018, where we used CG-MALS for TDP-43's folded NTD), we cannot achieve the high stock concentrations of WT CTD required, and trace aggregates complicate MALS in a manner similar to DLS. We also considered SEC-MALS - even if trace aggregates are removed by the SEC separation, the weak dissociation constant and transient assembly kinetics for assembly mean that we would always have an ensemble of monomer and multimer states at any elution time. Using AUC, we have quantitatively confirmed the presence of higher-order tetramers for WT CTD. Therefore, we have performed all the accessible experiments that in our view provide the most reliable and informative characterization of the system.

We have clarified this point in the manuscript with the following addition:

“Given the transient, dynamic nature of CR interactions and their small population, structural characterization of the CR helix-helix interaction is not amenable to x-ray crystallography or electron cryo-microscopy (cryoEM).”

5. TDP-43 cell fractionation. How many replicates were performed for this experiment? Please indicate this also in the respective figure legend. Also, to allow the reader a better understanding of the reproducibility/variability of this assay, it would be better to present also the individual values for each experiment in Figure 2H. Additionally, the ANOVA statistical test assumes that the data are normal distributed, did the author check this assumption? Alternatively, I would recommend the use of non-parametric test like Kruskal-Wallis.

We thank the reviewer for their important suggestions. The nuclear/cytoplasmic distribution of TDP-43 mutants versus WT was performed in at least five independent experiments, and we have now indicated this in the figure legend for clarity. Additionally, we have included the individual values for each experiment in Figure 2H (shown below).

Regarding the statistical analysis, we understand the reviewer’s concern about the use of ANOVA. In our figure displaying the individual data points (see below), the data do not appear dramatically skewed, and hence assuming a Gaussian distribution seems justified. Although it is difficult to reject the null hypothesis of normality, the number of replicates per group was insufficient to conclusively determine whether the data follow a Gaussian distribution. Therefore, we chose to use the non-parametric Mann-Whitney test, which did not change the conclusion from the previous analysis.

Figure 2H. TDP-43 nuclear retention was quantified by measuring the ratio of HA-TDP-43 in nuclear and cytoplasmic fractionation experiments (n ≥ five independent experiments). Quantification of TDP-43 was conducted by measuring HA-TDP-43 levels in the independent replicates. Nuclear retention ratio of WT TDP-43 and its single alanine substitution variants within CR in HEK293^{HA-TDP-43} were quantified from immunoblotting. Nuclear retention ratio of helix-disrupting mutant, A326P is included as a control of reduced nuclear retention. Statistical analysis was assessed using the non-parametric Mann-Whitney U test. ****p < 0.0001, ***p < 0.0005, **p = 0.001, *p = 0.01, ns = non-significant. Mean ± SD is shown for ≥ 5 independent replicates.

6. It is very interesting that a mutation of two closed amino acid very like S333A and W334A leads to two opposite effects on the stability of the alpha helix (Figure 2A, 2B). The author may use this example to emphasize the specific contribution of W on alpha helix stabilization.

We thank the reviewer for mentioning this important point. Neighboring residues S333A and W334A indeed lead to opposite effects on phase separation, with S333A enhancing and W334A disrupting it. Although we did not focus on this in the original manuscript, as suggested by the reviewer, the spectrum of W334A does show partial helical destabilization (**Fig. S2**), suggesting that the disruption of intermolecular interactions in W334A underscores the critical role of tryptophan in mediating interactions that stabilize the α -helix within the CR. We also show data suggesting that S333A does not significantly increase helicity across the CR at the monomeric level (**Fig. S2**), despite alanine's potentially higher helical propensity, further highlighting the potential importance of intermolecular interactions in stabilizing the helix, as observed by NMR (Conicella and Dignon et al., *PNAS*, 2020). We have added this discussion to the manuscript as follows:

*“However, the opposing effects on CTD phase separation of alanine variants at two neighboring residues S333 and W334 present an interesting case. NMR spectra of monomeric W334A reveal features suggesting the variant reduces helicity, highlighting W334’s unique role (**Fig. S2**) and suggesting that this variant may suppress phase separation by decreasing helicity and/or weakening inter-helix interactions. Although alanine can moderately enhance local helicity compared to serine⁵, S333A does not markedly alter the helical signatures across the monomeric CR, implying that its enhancement of phase separation primarily driven by strengthened inter-helical interactions or by local helical stabilization (**Fig. S2**).”*

Supporting Figure S2. Comparison of effects of example alanine variants on NMR spectra of monomeric TDP-43 CTD. S333A (left) shows spectral changes near the mutation site as expected for a covalent modification but shows little change to the CR – for example helical M322, A324, A325, A326 that are distant from the site of mutation show little change in helicity. W334A (right), however, shows changes at these positions far from the mutation site, consistent with reduction in helical population.

7. In case there is a public-repository for NMR data, the author should upload the raw data to make them accessible for the scientific community (FAIR principles to optimize the reuse of data).

We appreciate this suggestions and we will work with the BMRB to deposit raw data for this study.

8. The authors indicate that Q327, S332A and S333A were not measurable by NMR due to broadening of the CR signal and refer to the strong self-assembly they have observed before. Yet, Q331 showed a similar strong self-assembly in Figure 2A, but could be measured by NMR? Do the authors have an explanation for this?

We thank the reviewer for pointing out this important point. The variants Q327A, S332A, and S333A were examined by NMR, but at high concentration point (90 μM) used for the entire series of variants, the NMR signals arising from the CR region were broadened beyond detection due not only to enhanced population of higher molecular weight multimeric forms, but also due to the details of the exchange kinetics linking monomer and multimer. Additionally, the ensemble of assembled states may be perturbed such that exchange of conformations within the multimer also contributes to exchange broadening. The precise on- and off- rates differ in ways that are hard to predict, making their effects on NMR signals difficult to anticipate. We have addressed in the manuscript as follows:

“Q327A, S332A, and S333A were excluded due to broadening of CR signals beyond detection in HSQC experiments at the higher concentration (90 μM), unlike Q331A, which remained detectable despite also leading to enhanced self-assembly (Fig. 2A). This difference may result from variations in monomer-multimer exchange kinetics, binding and dissociation rates, and conformational changes within the assemble state.”

9. Please include page numbers for revisions.

We have now included page numbers in the revised document.

Reviewer #4 (Remarks to the Author):

TDP-43 has been identified as a major component of inclusions found in patients with Amyotrophic Lateral Sclerosis (ALS) and certain forms of dementia. This RNA-binding protein contains an intrinsically disordered region (IDR) approximately 160 amino acids long within its C-terminal domain (residues 267–414). Numerous studies, including analyses of postmortem patient biopsies, have demonstrated that these aggregates primarily consist of fragments from the IDR. Additionally, most disease-associated familial mutations are located within this region. Therefore, understanding the aggregation mechanisms of the IDR is crucial for elucidating the underlying causes of these diseases.

Current knowledge from various biophysical characterizations indicates that the IDR of TDP-43 forms an amyloid-dominant structure, as evidenced by cryo-electron microscopy (cryo-EM) studies on inclusions from postmortem patients. Although not entirely disordered, the IDR contains a transient alpha-helical region (residues 319–341; the CR in this manuscript) that has a propensity to self-assemble. This domain facilitates the protein's ability to undergo phase separation, which is related to its role in forming stress granules or RNA granules.

In the manuscript by Rizuan et al., the authors aim to understand the detailed structure of the assembly of this alpha-helical region in its dynamic state. While their experiments and simulations are extensive and methodologically sound, the article falls short of achieving its main goal—to provide a detailed understanding of the structural aspects of the C-terminal region of TDP-43. The reliance on molecular dynamics (MD) simulation interpretations is heavy, yet the experimental support is weak. The rationale behind the design of specific mutants and their subsequent interpretations is unclear. Furthermore, some interrelated factors, such as the interplay between alpha-helical propensity and self-association tendency—which could significantly affect the results—have been overlooked.

We thank the reviewer for their thoughtful comments and for recognizing that our experiments and simulations are extensive and methodologically sound.

We note that our new analyses now provide direct evidence of a tetrameric structure by AUC, which is transiently populated at conditions where phase separation is suppressed (e.g., low salt, whereas at physiological salt conditions, the structural experiments could not be conducted at all). Importantly, our data here and prior data we extensively reference suggest this tetrameric form is highly likely to be a critical functional species formed in cells.

We have utilized state-of-the-art experimental methods (NMR and AUC) to characterize the dynamic multimerization of the WT CTD, which has so far resisted high-resolution structural characterization. However, this remains a highly challenging problem that inherently requires an integrative experimental and computational approach and cannot be addressed using experimental tools alone. The experiments we conducted represent the cutting edge of what is currently possible.

As a result, our use of AI-based structural predictors and MD simulations is indeed significant, as the reviewer noted. It is important to highlight the growing role of computational methods in structural biology (Turoňová et al., *Science*, 2020, Casalino et al., *ACS Cent Sci.*, 2020, Zhao et al., *Sci Adv.*,

2023, Mondal et al., *JCIM*, 2023, Lazaridi et al., *Sci. Rep.*, 2024, Berkeley et al., *JACS*, 2025), as discussed in a recent review (Schwalbe et al., *Structure*, 2024). For example, development of AlphaFold, recently recognized with a Nobel Prize, has revolutionized structural biology by predicting protein structures with remarkable accuracy.

However, AlphaFold and similar methods, primarily trained on ground-state structural data for folded domains, may fall short when applied to dynamic assemblies like the TDP-43 CTD oligomers, where transient and heterogeneous interactions dominate (Shor and Schneidman-Duhovny. *Curr Opin Struct Biol.*, 2024). In such cases, the use of MD simulations, combined with AlphaFold predictions and experimental insights, becomes essential for providing atomic-level insights into these dynamic assemblies; insight that is currently inaccessible to either experimental or AI-based tools alone. Utilizing AlphaFold and MD simulations allows us to generate and refine plausible structural models of CTD multimers, providing atomic detail on potential structural configurations that can guide future experimental validation efforts.

We have now revised our Introduction to emphasize both the complexity of the problem and the critical need for integrative modeling:

“However, the molecular structure of these self-assembled species remains unresolved due to the challenges of rapid aggregation of CTD at higher concentrations needed for structure determination and dynamic interconversion of weak affinity states. Determining structural models of TDP-43 CR assembly therefore necessitates an integrative approach that combines artificial intelligence (AI)-based modeling, computer simulations, and constraints from multiple complementary experimental methods.”

We have also expanded the discussion of the increasing implementation of computational tools for structural characterization, particularly for dynamic assemblies, in the revised manuscript:

“The emergence of AlphaFold2 (AF2) has revolutionized the field of structural biology, offering accurate predictions of protein 3D structures²⁵. Though best known for highly accurate determination of protein structures from primary sequences, a recent extension, AlphaFold2-Multimer (AF2-Multimer), marks a significant step forward in the structure prediction for both homomeric and heteromeric protein complexes²⁶. However, the structural training set and co-evolutionary sequence data for predicting oligomerization are much more limited, especially for transient, weakly-binding, dynamic assemblies that are underrepresented in the Protein Data Bank (PDB). Hence, it is not clear if AF2-Multimer can generate reliable predictions for such assemblies or discriminate between potential structures. Instead, we propose that structural outputs from AF2-Multimer can serve as starting configurations for unrestrained molecular dynamics (MD) simulations, providing insights into the conformational dynamics of oligomeric states. Recent advancements in atomistic force fields^{27,28}, combined with GPU-accelerated MD engines^{29,30}, have significantly improved the accuracy and efficiency of simulations exploring conformational landscapes of intrinsically disordered proteins (IDPs) and their assemblies^{23,31-35}. MD simulations can offer feedback on the stability of multimeric assemblies generated by AF2-Multimer, serving as a valuable complement to biophysical experiments and offering atomic-level insight not currently achievable through experimental or AI-

based tools alone. These approaches provide workable structural hypotheses that can guide future experimental validation³⁶. Together, AF2 and MD simulations have become indispensable tools in structural characterization, as demonstrated by their increasing use in recent studies³⁷⁻⁴², significantly accelerating progress in structural biology³⁶.”

Thank you again for your constructive feedback, which has helped us clarify and strengthen the scientific motivation and broader significance of our study.

The major concerns of this work are listed below.

(1) It has been reported that the IDR of TDP-43 exhibits a stronger tendency to aggregate at higher pH levels and increased NaCl concentrations. This is because the net positive charge of the disordered region leads to enhanced self-association as the pH rises, reducing protonation and thereby diminishing electrostatic repulsion (PMID: 30814253). Similarly, higher NaCl concentrations screen electrostatic repulsive forces, further promoting aggregation (PMID: 30489059). Although the alpha-helical propensity remains and the chemical shift perturbations between pH 7 with 150 mM NaCl and pH 6.1 are minimal, the effects on self-association and alpha-helical tendency—which are key for the protein's phase separation or aggregation—are also small, as demonstrated in the authors' seminal paper (Figures 5 and 6; PMID: 27545621). Given that even small differences in alpha-helical structural propensity under these two conditions can significantly impact the helical component population and self-assembly—as shown in their previous work—such variations should not be dismissed as negligible or overlooked in subsequent studies.

We appreciate the opportunity to revise the text to clarify these important points. The main purpose of the data presented in Figure 1 is to demonstrate that the native state of TDP-43 CR retains alpha-helical character under physiological salt conditions, countering recent claims that the CR may adopt a β -sheet conformation at neutral pH and physiological salt (Zhou et al., *Science*, 2022), despite the lack of direct structural evidence in that study. We support this point by showing that the differences in α -helical propensity between pH 6.1 (low salt) and pH 7.0 (150 mM NaCl) are minimal, as also pointed out by the Reviewer.

But, as the reviewer points out, small changes in helicity could lead to differences in phase separation and helical population that may contribute to changes in behavior. We have revised the manuscript to clarify this point:

“Positive $\Delta\delta C_{\alpha}$ values within the CR reveal its helical character at neutral pH and physiological salt conditions. In fact, helicity is slightly enhanced at these conditions. Enhancement in helicity could also lead to enhanced helical interaction and cooperate with reduced electrostatic repulsion at higher pH and higher ionic concentration that drive TDP-43 phase separation as well as aggregation as previously observed^{A7,48}.”

(2) In the alanine-scanning studies on the CR region, the alpha-helical propensity would differ among these mutations, and this structural tendency modulates self-association, as also noted in the authors' previous work (PMID: 32132204). However, the dynamic nature of the helical structure in self-association seems to have been overlooked. Although the manuscript discusses helical

propensity changes in one paragraph, these are attributed solely to results from "dimerization"—bearing in mind that the dimer was the authors' input for AlphaFold-multimer, not an outcome of the simulation. In the chemical shift analysis ($\delta^{15}\text{N}$ CSP), this parameter is affected by both the equilibrium between different structural populations (random coil versus alpha-helix) and interactions from self-association. Thus, the concluding sentence in this section, "This support(s) the idea that residues stabilizing CR helix-helix contacts in the dimer state contribute to the phase separation of CTD," may not be fully justified.

We appreciate the reviewer's comments and we note that we have substantially restructured this section of the manuscript to improve clarity and explanation. We agree with the reviewer's main point that the alanine mutations may alter not only helix-helix contacts but may also alter helical stability. This is an important point.

We have updated the section title to read:

TDP-43 CR helicity and helix-mediated interactions are linked to CTD phase separation

And the text to say:

*"We note that these CR variants may contribute to disruption in helix-helix interactions by destabilizing the helicity of the monomeric form (which is significantly populated from 320-330) as previously seen for some ALS-associated, engineered, and phosphomimetic variants in the 321-333 region^{16,52}, disrupting the helix-helix interface without changing the helicity of the monomer as suggested for other ALS-associated and engineered variants at M337, or a combination of these factors. Previously, we found that simultaneous substitution of all five CR methionines to alanine does not suppress helicity but does disrupt phase separation²⁰. However, the opposing effects on CTD phase separation of alanine variants at two neighboring residues S333 and W334 present an interesting case. NMR spectra of monomeric W334A reveal features suggesting the variant reduces helicity, highlighting W334's unique role (**Fig. S2**) and suggesting that this variant may suppress phase separation by decreasing helicity and/or weakening inter-helix interactions. Although alanine can moderately enhance local helicity compared to serine⁵, S333A does not markedly alter the helical signatures across the monomeric CR, implying that its enhancement of phase separation primarily driven by strengthened inter-helical interactions or by local helical stabilization (**Fig. S2**)."*

(3) In their attempt to verify the helix-mediated assembly and function, the authors used a cell model to measure the nuclear-to-cytoplasmic ratio. They found that mutants enhancing helical assembly also increased nuclear localization. However, the underlying mechanism by which helix-promoted self-assembly enhances nuclear import is not apparent; one might expect that larger assemblies would impede nuclear entry. Although the authors note this discrepancy by stating that it "highlights the need for further investigation into this mechanism," this point should be elucidated. Furthermore, since the most extensively studied characteristics of TDP-43 assembly involve its cellular toxicity or amyloid-fibril formation, the authors should also investigate how these helix-mediated assembly mutants affect these well-characterized phenomena.

We thank the reviewer for raising this important point, which prompted us to further clarify our interpretation of the nuclear-to-cytoplasmic ratio data.

Our findings suggest that mutants that enhance phase separation increase nuclear retention not due to elevated nuclear import, but because the formation of TDP-43 condensates or multimeric assemblies prevents passive exit of TDP-43 into the cytoplasm. TDP-43 import into the nucleus is importin-dependent and requires active nuclear localization signals (Winton et al., *J Biol Chem.*, 2008, Nishimura et al., *Brain.*, 2010). Importantly, binding to importins actively disassembles both TDP-43 phase-separated assemblies and even aggregates (e.g., Guo et al., *Cell*, 2018, describing importin-mediated disaggregation). Thus, it is widely believed that TDP-43 is imported as a monomer. By contrast, nuclear exit of TDP-43 is not driven by active export pathways. Accordingly, macromolecular size and multimerization play an important role in regulating nuclear retention, as supported by several studies from the Ayala lab and others (Ederle et al., *Sci Rep.*, 2018, Pinarbasi et al., *Sci Rep.*, 2018, Archbold et al., *Sci Rep.*, 2018, Duan et al., *Cell Rep.*, 2022, Dos Passos et al., *PLOS Biol.*, 2024). Once bound to RNA, WT and condensate-competent TDP-43 mutants form large macromolecular assemblies that help retain the protein in the nucleus. Thus, mutations that enhance phase separation may shift the equilibrium toward these higher-order assemblies and increase nuclear retention.

We also thank the reviewer for the insightful suggestion to investigate how CR mutants that alter helix-mediated assembly affect cellular toxicity and amyloid fibril formation. While these questions are highly relevant, they extend beyond the scope of this study, which focuses primarily on the structural features of helix-mediated CR multimerization. Moreover, collecting new cellular toxicity or aggregation data is not feasible within a reasonable timeframe. Nonetheless, we have initiated new collaboration with Dr. Randal Halfmann (Stowers Institute) and continuing our collaboration with Prof. Yuna Ayala (Saint Louis University) to pursue these directions in future work.

In the meantime, we have incorporated additional analyses based on published cellular growth restriction / toxicity data from the Lehner lab (Bolognesi et al., *Nat Commun.*, 2019), which includes deep mutagenesis of over 1,200 single-residue variants across the TDP-43 CTD. In this study, full-length TDP-43 variants were overexpressed in yeast, and their impact on growth served as a measure of relative toxicity compared to WT TDP-43. Analysis of these data reveals that mutations within the CR region have a significantly stronger effect on cellular toxicity than those in the flanking IDRs, underscoring the CR's central role in TDP-43 function. Notably, mutations at W334, shown by us to be critical for CTD self-assembly in this paper, generally increase toxicity, with the exception of substitutions like L and M.

Next, we examined whether the *in vitro* phase separation propensity of alanine mutants within the CR correlates with yeast toxicity. While some alanine mutants were missing from the Lehner lab, those available revealed a strong relationship (**Figure S5**): alanine substitutions that reduce yeast toxicity tend to enhance *in vitro* phase separation, while those that increase toxicity tend to reduce it. This inverse correlation (Pearson $r = 0.79$; Spearman $r = 0.86$) further supports the functional relevance of helix-mediated CR assembly.

Importantly, Lehner and coworkers also observed altered localization (into assemblies at the nuclear periphery) for variants that they show are more toxic (less cell growth) and that we show decrease

nuclear retention. Based on this, we propose that helix-mediated TDP-43 self-assembly enhances functional TDP-43 assembly.

We have now incorporated these points into the manuscript with a new Supporting Figure S4, as follows:

"We next explored the impact of helix-mediated assembly of full-length human TDP-43 expressed in yeast on growth restriction using deep mutagenesis data from the Lehner lab, which includes over 1,200 single mutations in TDP-43 CTD⁶¹. Mutations within the CR showed a stronger influence on yeast growth restriction compared to mutations in the flanking IDRs, underscoring the critical role of the CR in TDP-43 function and toxicity⁶¹. Notably, mutations at W334 increased toxicity, except for L and M substitutions, correlating with W334's essential role in CTD self-assembly and nuclear retention. A strong correlation (Pearson $r = 0.7$; Spearman $r = 0.86$) was observed between in vitro phase separation propensity and yeast toxicity (Fig. S5). Alanine mutations that reduce toxicity in yeast cells enhance phase separation, whereas ones that increase toxicity do the opposite. This further supports the cellular significance of helical self-assembly, as mutants enhancing cellular toxicity also enhanced accumulation in the cytoplasm at the nuclear periphery. We propose that helix-mediated TDP-43 self-assembly aids nuclear retention, mitigating improper localization and toxicity."

Figure S5. The relative toxicity of single alanine substitution variants at non-alanine positions within the CR and adjacent residues in yeast cells¹⁴ shows a strong positive correlation (Pearson correlation coefficient, $r = 0.79$) and Spearman's rank correlation coefficient, $(r = 0.86)$) with the C_{sat} , normalized by the C_{sat} of the WT CTD, from single alanine substitution variants at the same positions.

(4) The AUC data indicated the presence of dimers and possibly larger oligomeric species in equilibrium. However, additional complementary methods could be employed to more conclusively determine the size and distribution of these oligomers. Techniques such as dynamic light scattering, nanoparticle tracking analysis, or even cryo-electron microscopy imaging could provide more insights into the oligomerization states. Applying these methods would strengthen the rationale for the subsequent molecular dynamics simulations that use different oligomeric starting models.

We thank the reviewer for their suggestion, which is similar to that of Reviewer 3. However, the complementary methods suggested by both reviewers are not well-suited for our system, as explained below.

Dynamic light scattering (DLS) is not suitable in this case, as WT CTD samples inevitably contain trace amounts of aggregated material, which would dominate the signal. Techniques such as mass photometry are also not ideal due to the need to use high sample concentrations to form the assembly, well above the nanomolar range recommended for mass photometry. We considered static light scattering, but for composition gradient MALS (CG-MALS), which could in theory be used to fit a binding model (see our previous work, Wang et al., *EMBO J.*, 2018, where we used CG-MALS for TDP-43's folded NTD), we cannot achieve the high stock concentrations of WT CTD required, and trace aggregates complicate MALS in a manner similar to DLS. We also considered SEC-MALS - even if trace aggregates are removed by the SEC separation, the weak dissociation constant and transient assembly kinetics for assembly mean that we would always have an ensemble of monomer and multimer states at any elution time. Using AUC, we have quantitatively confirmed the presence of higher-order tetramers for WT CTD. Therefore, we have performed all the accessible experiments that in our view provide the most reliable and informative characterization of the system.

Regarding methods to characterize structural details of helical CR multimers seen by AUC, techniques such as X-ray crystallography or electron cryo-microscopy (cryoEM) are not suitable due to the transient and dynamic nature of CR interactions. Additionally, CTD multimers are not fully populated under the conditions achievable for these techniques. We have clarified this point in the manuscript with the following addition:

“Given the transient, dynamic nature of CR interactions and their small population, structural characterization of the CR helix-helix interaction is not amenable to X-ray crystallography or electron cryo-microscopy (cryoEM).”

(5) The rationale for mutating 13 serines to glutamates (S to E) in the non-CR regions to enhance solubility is unclear. Many serine phosphorylation sites are associated with this protein's pathological aggregation (S379, S403, S404, Ser409, Ser410), and substituting serine with glutamate is often used to mimic phosphorylation effects, which leads to amyloid-fibril. Introducing an additional cysteine to force dimerization is also questionable. While their previous work investigating alpha-helical propensity using disulfide linkages was reasonable, applying this method here may not be suitable because the aim of this manuscript is to characterize the structure of the dynamic alpha-helical assembly. The antiparallel conformation in the AlphaFold-multimer-generated structures for

MD simulations is not considered in these cases. Furthermore, the NMR experiments in Figures 3D–F are expected results because cross-linking at the N-terminus connects two monomers, inducing helix-helix interactions (evident from chemical shifts) and increasing R_2 relaxation rates due to the larger molecule reducing overall tumbling. As mentioned above (point 2), chemical shifts are influenced by both structural contacts and the equilibrium between coil and helix conformations. Here, the measured R_2 is also related to the chemical exchange between coil and helical conformations, adding to the line broadening caused by interactions between monomeric and dimeric states. These factors are ignored and cannot be simply attributed to the formation of higher-order (beyond dimer) structures.

We appreciate the Reviewer's thoughtful concern regarding the design of CTD mutants used in our experiments and interpretations of the resulting data.

Because this comment addresses several distinct issues, we are provide a point-by-point response below:

(i) Regarding the role of serine phosphorylation in pathological aggregation:

We have updated the text to make important changes brought up by the reviewer, highlighting how phosphorylation and phosphomimetic substitution can enhance aggregation but, if at enough positions, disrupt aggregation:

Phosphorylation at the very C-terminal part of TDP-43 CTD (S403, S404, S409, S410) is a major feature of TDP-43 inclusions and placing aspartic acid residue substitutions mimicking phosphorylation at these sites can drive aggregation⁶⁴. In contrast, placing aspartic acid residues at 12 sites across the entire CTD appears to counteract pathological aggregation⁶⁵, enhance the liquidity and dynamic nature of TDP-43 condensates, while preserving nuclear import or RNA regulatory functions, similar to what we have observed for poly-phosphomimetic FUS⁶⁶. Similarly, here we made serine to glutamate (S→E) substitutions across the disordered flanking regions of CTD (See Table S1), which significantly enhanced TDP-43 solubility while preserving the helical structure in the CR structure as indicated by NMR (Fig. S6).

(ii) Rationale for cysteine-based cross-linking and S→E mutations:

This point is similar to a concern raised by Reviewer 2. As explained there, our goal was to design soluble CTD variants that permit NMR-based structural characterization of CR multimers, which are otherwise difficult to study due to their aggregation-prone nature. We also discussed the reasoning for using cross-linking and mutations to enhance IDR solubility while preserving the CR structure.

Additionally, we emphasize that our qualitative validation of helical CR tetramer structural models is not solely dependent on cross-linked CTD forms. We also employed NOE NMR data from 6F→A CTD variant (without cross-linking), which replaces phenylalanines in the IDRs with alanine (with no changes in or near the CR), to enhance TDP-43 CTD solubility without altering the CR structure. We recently demonstrated that this mutation reduces aggregation of CTD while retaining phase

separation and CR helix-helix interactions (Mohanty et al., *PNAS*, 2023). The similarity in NOE patterns between the 6F→A (non-cross-linked) and cross-linked CTD forms suggests that CR assembly occurs similarly in both contexts.

We have revised the manuscript and Supporting Information accordingly to clarify this rationale and ensure that the structural and functional integrity of the CR was preserved in all engineered variants.

(iii) Regarding the possibility of antiparallel orientation of helices in multimers with cross-linked CTD forms:

As we discussed above, our prior studies demonstrated that two obligate CTD dimers, cross-linked near each terminus of the CTD at S273C and S387C, exhibit similar chemical shift perturbations in direction and pattern to non-cross-linked CTD. This suggests that cross-linking does not impose a strong bias on helix-helix orientation due to the symmetric positioning of the cross-link sites (Conicella and Dignon et al., *PNAS*, 2020). Positioning the cross-link site near the N-terminal part of the helix (S305C, S317C) may favor parallel dimer association. However, even in these cases, parallel dimers can further associate into antiparallel tetramers, adopting an up-down-up-down topology as seen in Tet-1. In this arrangement, the two helices in the "up" position remain parallel to each other, as do the two in the "down" position.

(iv) Interpretations of NMR experiment results in current Figure S6:

We very much agree with the comments of the reviewer regarding the NMR. We emphasize that observing the helical assembled state is extremely challenging because trapping it (driving the system to 100% population of this structure) has been elusive to everyone in the field – here we have enabled many ways to probe the assembly. We have extensively revised the parts of the paper with interpretation of these data. The main text now says:

When cross-linked to enhance interaction as we previously demonstrated for the wild-type protein⁵, these variants populate multimeric assemblies that depend on the helical region, as seen by AUC and show enhanced helicity compared to the uncross-linked forms (Fig. S7,9).

(6) The NOE data provided information only on the types of amino acid contacts but did not reveal detailed structural insights. While it is understandable that these may be the best results obtainable using NOE to characterize such a heterogeneous interacting system, the NOE experiments themselves are also limited because they were conducted on a construct where six phenylalanine residues in the non-CR region were replaced with alanine to increase solubility. This modification could overlook potential assembly contacts, such as the interaction involving tryptophan 334 in the CR region through pi-pi stacking (PMID: 33909608, PMID: 29511089). While the authors aim to focus on alpha-helix interactions, it has been reported that other parts of the IDR contribute to phase separation—for example, aromatic residues acting as "stickers" in the organization and formation of higher-order oligomers. These interactions may be critical in assembly studies and experimental characterizations but are overlooked in the MD modeling work. Furthermore, the "structure" of the helical assembly used for MD simulations was derived from AlphaFold-Multimer models. Lacking direct evidence of tetramer formation—and considering that many experimental investigations have used induced dimerization through disulfide linkages—it is difficult to be convinced by the simulated

results. Collectively, these data are not convincing that this manuscript significantly advances our current knowledge or offers novel insights into the structural mechanisms of TDP-43 aggregation.

Because this comment includes multiple concerns, we address them point by point below. To summarize, we appreciate the reviewer's concerns. Our revised manuscript has been altered to address these concerns to demonstrate that focusing on the helical region via experiments and simulations of both the wild-type protein and a series of variants each useful and validated against the behavior of the wild-type. The most important is that we have added direct evidence of tetramer formation of the wild-type protein without cross links. Together, these data now make a strong case that the structural insights achieved here provide a key to understand the structural mechanisms of TDP-43 functional assembly (and, we note, not disease-associated aggregation).

(i) Lack of direct evidence for tetramer formation:

This is the a very important issue that we have now directly resolved. We now provide new SV-AUC data supporting a monomer–dimer–tetramer equilibrium for WT CTD. These results show that once weakly bound dimers are formed ($K_d \sim \text{mM}$), cooperative formation of tetramers occurs with μM affinity, consistent with the existence of a metastable dimeric intermediate. This experimentally observed equilibrium supports the structural models explored in our MD simulations. These results directly address the reviewer's concern about the lack of experimental support for tetramer formation.

(ii) Rationale for CTD variants (e.g., 6F→A) used in NOE experiments:

As discussed above, WT CTD poses substantial challenges for structural characterization of higher-order multimers by NMR. To overcome these limitations, we designed multiple CTD variants that improve solubility while preserving CR multimerization via helix-mediated interactions. One such variant is the non-cross-linked 6F→A CTD as discussed above. Another solubility-enhancing approach involved introducing negatively charged glutamic acid residues at serine positions (S→E) in the flanking IDRs, coupled with disulfide cross-linking at sites near the CR (e.g., S305C, S317C) to stabilize multimeric assemblies for structural analysis (see responses to Reviewer #2 for more details).

We emphasize that similar NOE patterns were observed in both the 6F→A and S→E cross-linked variants, suggesting that CR assembly is preserved across these experimental contexts. We have clarified these points in the manuscript as follows:

“Given the transient, dynamic nature of CR interactions and their small population, structural characterization of the CR helix-helix interaction is not amenable to x-ray crystallography or electron cryo-microscopy (cryoEM). Hence, we next sought structural information for the assembled CR region of TDP-43 CTD using nuclear Overhauser enhancement (NOE) NMR. However, several features make CR multimerization an extremely challenging target. First, NOE experiments are often conducted at 1 mM but TDP-43 CTD samples at higher concentrations ($\geq 100 \mu\text{M}$) are not stable and convert to aggregates, preventing long NMR experiments. Second, the multimer is not fully populated at achievable conditions, reducing NOEs from intermolecular contacts. Third, exchange between the monomeric and multimeric forms broadens the resonances from the CR¹⁶. Fourth, the

CR itself is aggregation prone, as a shorter construct that isolates the CR (aa: 310-360) resulted in essentially complete insolubility (data not shown), suggesting the flanking IDR regions contribute to the solubility. Hence, we explored mutants of TDP-43 CTD IDRs to further enhance solubility while keeping the CR and adjacent regions (310-350) unaltered to preserve CR multimerization. We recently demonstrated that an aggregation-reducing variant of TDP-43 CTD that replaces phenylalanines in the IDRs with alanine⁶² (TDP-43 CTD 6F→A) retains phase separation and CR helix-helix interaction at higher protein concentrations²⁰. ”

*“To complement the structural data from TDP-43 CTD 6F→A, we also extensively characterized the assembly and NOE constraints of a series of cross-linked variants of TDP-43 (See Supporting text, **Figs. S6-9**). Phosphorylation at the very C-terminal part of TDP-43 CTD (S403, S404, S409, S410) is a major feature of TDP-43 inclusions and placing aspartic acid residue substitutions mimicking phosphorylation at these sites can drive aggregation⁶⁴. In contrast, placing aspartic acid residues at 12 sites across the entire CTD appears to counteract pathological aggregation⁶⁵, enhance the liquidity and dynamic nature of TDP-43 condensates, while preserving nuclear import or RNA regulatory functions, similar to what we have observed for poly-phosphomimetic FUS⁶⁶. Similarly, here we made serine to glutamate (S→E) substitutions across the disordered flanking regions of CTD (See Table S1), which significantly enhanced TDP-43 solubility while preserving the helical structure in the CR structure as indicated by NMR (**Fig. S6**). When cross-linked to enhance interaction as we previously demonstrated for the wild-type protein⁵, these variants populate multimeric assemblies that depend on the helical region, as seen by AUC and show enhanced helicity compared to the uncross-linked forms (**Fig. S7,9**). Using cross-linked forms created from 1:1 mixture of unlabeled and ¹³C/¹⁵N proteins, we again see a similar pattern of methyl NOEs as observed for TDP-43 CTD 6F→A (**Fig. S10**), suggesting that CR assembly is similar in both contexts where the TDP-43 CTD solubility is enhanced without altering the CR. ”*

(iii) Role of MD simulations in structures derived from AF2-Multimer:

We would like to clarify that AlphaFold-Multimer predicted many multimer structures that were not found to be stable for TDP-43 CR region in MD simulations. These structures merely provided a starting point and were carefully evaluated through extensive MD simulations to arrive at the tetramer model presented in this manuscript as the potential functional state. AlphaFold alone is not designed to rank structural stability, especially for weakly associated or transient assemblies. Therefore, we used all-atom MD simulations – 50 independent 100-ns runs per model – with the state-of-the-art Amber03ws force field to identify the most stable tetrameric structure. This model (Tet-1) exhibited strong structural stability and was consistent with biophysical and NOE experimental data. We recently used a similar strategy to probe conformational dynamics of HSPB1 chaperone oligomers, another difficult target for experimental structural determination (Berkeley et al., JACS, 2025).

We have added this explanation to the manuscript:

“Though best known for highly accurate determination of protein structures solely from their primary sequences, a recent extension of AF2, AlphaFold2-Multimer (AF2-Multimer), has marked a significant step forward in the structure prediction for both homomeric and heteromeric protein

complexes²⁶. However, the structural training set and co-evolutionary sequence data for predicting multimerization data is much more limited, especially for transient, weakly-binding, dynamic assemblies that are not extensively found in the Protein Data Bank (PDB). Hence, it is not clear if AF2-Multimer can generate reliable predictions of dynamic multimeric assemblies or discriminate between potential structures. Instead, we propose that the structural outputs from AF2-Multimer can serve as starting configurations for unrestrained molecular dynamics (MD) simulations, providing insights into the conformation and dynamics of multimeric states. Recent advancements in atomistic force fields^{27,28}, combined with GPU-accelerated MD engines^{29,30}, have significantly improved the accuracy and efficiency of exploring the conformational landscapes and interactions of intrinsically disordered proteins (IDPs) and their assemblies^{23,31-35}. MD simulations can offer feedback on the stability of multimeric assemblies generated by AF2-Multimer, serving as a valuable complement to biophysical experiments and offering atomic-level insights that are not currently achievable through experimental or AI-based tools alone. These approaches provide workable structural hypotheses that can guide experimental validation³⁶. Together, AF2 and MD simulations have become indispensable tools in structural characterization, as demonstrated by their increasing application in recent studies³⁷⁻⁴², significantly accelerating progress in structural biology³⁶.”

(iv) IDR interactions being overlooked in computational modeling:

We agree that interactions involving aromatic and aliphatic residues in the IDRs (e.g., W334, F313, F397) play an important role in TDP-43 CTD assembly. In our earlier modeling work, we focused on residues 310–350 to isolate the CR while maintaining flanking sequence context. This design was chosen to balance biological relevance and computational tractability, and because short sequence inputs have been shown to better capture bound conformations in AlphaFold-Multimer predictions (PMID: 38238291, 36413069). We have now added the following to the revised manuscript:

“This length of protein fragment, from our previous work on TDP-43 CTD^{5,16}, was selected as it is more computationally efficient for MD simulations and better suited for determination of the correct bound conformation using AlphaFold-Multimer^{53,54}.”

To directly investigate the role of flanking regions in the tetrameric state, we have now constructed full-length CTD tetramers using our most stable helical model (Tet-1) as a core. IDR regions were modeled with MODELLER (Eswar et al., *Curr Protoc Bioinformatics*, 2006), pre-equilibrated via coarse-grained simulations using the HPS-Urry model (Regy et al., *Protein Sci.*, 2021), and back-mapped to atomistic detail. We then conducted five 300-ns AA simulations (5 replicates) at physiological salt concentrations. These simulations demonstrate that CR helicity and interface contacts remain stable within the full-length CTD. In particular: (a) W334 remains surface-exposed in Tet-1 and forms additional contacts with both CR residues and flanking IDR sequences such as 276–283, 307–313, and 383–387. (b) Aromatic and aliphatic residues in the flanking IDRs—including F283, F313, F367, M307, and W385—form intermolecular contacts consistent with previous findings.

Supporting Figure S12. Analysis of CTD full-length tetramer simulations. **A.** Full-length CTD (aa: 267-414) tetramer model generated using MODELLER¹⁰, using Tet-1 structure for CR residues (aa: 319-341). **B.** Five representative CTD tetramer structures used as starting conformations for all-atom MD simulations. These structures were obtained from CG simulations of the model shown in (A) and subsequently back-mapped to atomistic resolution using MODELLER¹⁰. **C.** Per-residue α -helical fraction computed from CTD full-length tetramer AAMD simulations, averaged across all trajectories, highlighting the stability of helicity within the CR region (residues 320–341). **D.** dRMS analysis from AAMD simulations of Tet-1 model (CTD₃₁₀₋₃₅₀) (50 runs with 100 ns each) compared to CTD full-length (CTD₂₆₇₋₄₁₄) tetramer (5 runs with 300 ns each). dRMS distribution of all heavy atoms (excluding hydrogen atoms) for residues from 320 to 341 as a function of time with respect to initial conformations are shown. dRMS for CR tetramer of CTD₃₁₀₋₃₅₀ and CTD₂₆₇₋₄₁₄ shown in red and blue colors, respectively. The lighter colors represent the dRMS distributions for independent runs, while the darker colors represent the average dRMS. **E.** (top) Pairwise intermolecular contact maps of CTD full-length tetramer simulations (bottom) Total number of contacts per residue position (N_{contact}) derived through summation of all pairwise contacts along y-axis based on two dimensional pairwise

intermolecular contact maps. **F.** Per-residue intermolecular contacts within CR (aa: 320-341) calculated from CTD₃₁₀₋₃₅₀ and CTD₂₆₇₋₄₁₄ tetramer simulations show a strong correlation (Pearson correlation coefficient, $r = 0.95$, Spearman correlation coefficient, $r = 0.92$).

We have added a new Supplementary Figure S12 and corresponding supporting text to present these results as follows:

“Structural model of the tetrameric helical assembly of TDP-43 CTD

We and others have demonstrated the critical role of flanking region residues in CTD self-assembly⁷⁻⁹. Our recent work showed that CTD self-assembly is driven by helix-mediated CR:CR interactions, strengthened by transient interactions involving the flanking disordered regions, with aromatic and methionine residues playing key roles⁷. To investigate the role of flanking IDR residues in the tetrameric state, the full-length CTD tetramer structure was prepared by using the Tet-1 model for CR residues and modeling the remaining CTD residues with MODELLER¹⁰ (**Fig. S12A**). A 300 ns coarse-grained (CG) simulation was then performed using the MODELLER-generated structure, keeping the CR residues fixed as a rigid body in the Tet-1 conformation while allowing the IDR regions to relax, following our previous protocol¹¹. This pre-equilibration step was essential for refining the IDR regions and ensuring that subsequent AA simulations could be conducted within a reasonably sized simulation box. For this, we used the HPS-Urry model¹², which we previously demonstrated to be effective in capturing both the monomeric ensemble and the phase separation propensity of the CTD⁷. Following the CG simulation, the TtClust clustering method¹³ was applied to identify the 5 most representative structures of the full-length CTD tetramer for subsequent AA simulations (**Fig. S12B**). These CG structures were then back-mapped to atomistic representations using MODELLER¹⁰, and 5 independent AA simulations were set up using each of these back-mapped structures. In full-length CTD tetramer simulations, the CR helicity and CR:CR tetrameric interfacial contacts remained stable (**Fig. S12C,D**), confirming the structural stability of the helical CR tetramer structure within full-length CTD. Residues within the CR, including M322, A326, A329, L330, S333, M336/337, and L340, maintained high intermolecular contacts, consistent with shorter fragment simulations of Tet-1 (**Fig. S12E, F**).

”

Changes to the main text:

*“The most stable Tet-1 model also remained stable within the full-length CTD (see Supporting Text, **Fig. S12**). W334, surface-exposed in Tet-1, showed enhanced interactions in the full-length CTD, interacting with hydrophobic segments of the flanking regions (²⁷⁶FGGNPGGF²⁸³, ³⁰⁷MGGGMNF³¹³, ³⁸³IGWGS³⁸⁷), consistent with its essential role in CTD self-assembly (**Fig. 2A,B**). Additionally, aromatic (F283, F289, F313, F316, F367, Y374, W385, F397, F401, W412) and aliphatic (M307, M311, I383) residues in the flanking IDRs formed significant intermolecular contacts in agreement with previous findings²⁰. These results reinforce the essential contributions of flanking regions and W334 in CTD self-assembly and further support the structural relevance of the Tet-1 model.”*

In summary, our study integrates experimental (AUC, NOE NMR), computational (MD simulations), and AI-based (AF2-Multimer) approaches to reveal structural features of the TDP-43 CR assembly.

The dynamic nature of this system limits the applicability of single-method strategies. We believe the combined evidence presented in the revised manuscript now more robustly supports our conclusions and advances structural understanding of TDP-43 CTD multimerization.

Reviewer #1 (Remarks to the Author):

The revisions are well executed, and I have no further suggestions at this stage.

Response to Reviewer #1

We thank the reviewer for their thorough reading of the manuscript.

Reviewer #2 (Remarks to the Author):

In the revised manuscript, the authors provide more detail of SV AUC of TDP-43 CTD in an attempt to establish the stoichiometry of a putative self-association. They report that Lamm equation modeling SV scans of TDP-43 loaded at 58 μM and 107 μM using SEDPHAT is consistent with a monomer – dimer – tetramer model with weak self-association to the dimer and stronger association of the dimer to the tetramer. The fitted dissociation constants for the monomer – dimer and dimer – tetramer equilibria were 2000 μM and 2.9 μM , respectively. They found that alternative monomer-dimer, monomer-trimer, monomer-tetramer and monomer-dimer-trimer models “resulted in poor fits and/or physically unreasonable sedimentation coefficients”, but that “excellent fits” were found for the monomer-dimer-tetramer model. The analysis represents state of the art use of SV AUC to model self-association equilibria, although they could also have presented F-statistics, which are used in SEDPHAT to show statistical significance of differences in fitting alternative models. The acceptance of the model must be viewed with skepticism. The highest loading concentration of 107 μM is ~20-fold lower than the K_d for the weaker, obligate dimerization step. As a result, as they note, there is very little population of the tetramer. To accept a binding model with confidence, a system should approach saturation. It is surprising that reported errors are relatively small. Overall, the available concentrations limit the investigation of an apparent ultraweak self-association of the protein.

Response to Reviewer 2:

We appreciate the reviewer’s salient comments and the opportunity to revisit our Lamm equation modeling for the SV-AUC data on TDP-43 CTD. During the process of recalculating the confidence interval limits for the interaction affinities, we discovered that some of the best-fitting models were converging to a local minimum in the parameter error space. The problem was corrected, and the affinities and sedimentation coefficients reported for the monomer–dimer–tetramer model have changed slightly. The proper values are now reported in the revised manuscript, noting that the fitted dissociation constants for the monomer–dimer, and dimer–tetramer equilibria are now 2.4 mM and 2.7 μM , respectively (slightly changed but within the previous confidence interval). The confidence interval representing one standard deviation for the monomer – dimer equilibrium are 1.1 – 3.4 mM (it is not technically normally distributed because the dissociation constant cannot be 0 and, when the uncertainty is sizable, the skew in this parameter is more evident) and for the dimer – tetramer equilibrium \pm 0.6 μM .

As requested, we now report the global reduced chi-square values for the best fits obtained from the various Lamm equation models. The values indicate the goodness of fit and serve as a criterion for identifying the best model. According to the new Supplementary Table S1, the monomer–dimer–tetramer model provides the best fit with a global reduced chi-square value of 0.1657. Additionally, for comparison, we report the estimated values for the critical reduced chi-

square value at a 68.3% confidence level (one standard deviation) obtained from SEDPHAT using F-statistics.

While the monomer–dimer–tetramer model returns the best fit based on the globally reduced chi-square, the monomer–tetramer model is now statistically indistinguishable. The monomer–tetramer model yields an affinity $K_a = 3.96 \times 10^{10} \text{ M}^{-3}$ with an equivalent concentration of $2.94 \times 10^{-4} \text{ M}$ (= 294 μM), at which the monomer concentration equals the tetramer concentration (because the monomer–tetramer model has an association constant with M^{-3} units, this concentration where $[\text{monomer}] = [\text{tetramer}]$ can be useful, similar to how K_d is often presented). Though the monomer–dimer–tetramer model is depicted as a two-step process, the equivalent monomer–tetramer affinity for that model is $K_a = 6.36 \times 10^{10} \text{ M}^{-3}$ with an equivalent $[\text{monomer}] = [\text{tetramer}]$ concentration of $2.51 \times 10^{-4} \text{ M}$ (= 251 μM). In other words, these models are very similar and both agree on the tetramer effective association constant. Additionally, we note that although the reviewer is rightfully concerned that “The highest loading concentration of 107 μM is ~20-fold lower than the K_d for the weaker, obligate dimerization step”, the tetramer is significantly populated due to the much tighter dimer–tetramer affinity. Indeed, the 107 μM condition is less than 3x below the equivalent $[\text{monomer}] = [\text{tetramer}]$ concentration as represented in both of the models that explain the data. Although it would be ideal to perform the experiments at higher concentrations, this is not tractable as described below.

The observation above illustrates some of the issues that the reviewer correctly points out; however, we stress that the system is challenging in many ways. First, we are limited by the highest concentrations that can be studied because of solubility issues. It simply is not possible based on our experience to generate samples of the wild-type protein at higher concentrations without aggregation. Despite the low concentrations, the sedimentation velocity data show clear evidence for the formation of higher-order multimers. Furthermore, in the analysis of a self-associating system using Lamm equation modeling, Brautigam (<https://pubmed.ncbi.nlm.nih.gov/21187153/>) demonstrates that one does not require saturating concentrations to reliably measure an affinity due to the large number of data points fitted in the analysis (in this case, more than 71,000) and the extensive exploration of the parameter space.

Secondly, we are limited to sedimentation velocity AUC (SV-AUC). We would have preferred also to use sedimentation equilibrium AUC (SE-AUC) to distinguish between the self-association models, in part to avoid having to model kinetic parameters (k_{off}) for the interaction. Again, these experiments were impractical because of stability and solubility issues as SE requires longer sample life than these samples demonstrate, especially at the higher concentrations.

Other issues that could influence higher concentration studies include hydrodynamic (SV-AUC) and thermodynamic (SV and SE-AUC) non-ideality. As these processes are not an issue and the data are well modeled, it may be a reason why the reported errors returned from SEDPHAT based on F-statistics are relatively small.

As noted, we have corrected the sedimentation coefficients and affinities for the best-fit monomer–dimer–tetramer model following a reanalysis. Consequently, we have updated Figures 3B and 3C, along with their corresponding legend. In addition, we have added Supplementary Table S1, which describes the global reduced chi-square and other parameters for the various Lamm equation models that best fit. The **most critical finding** is that these models show that solution that do not involve a tetramer, such as a monomer–dimer model, result in poor fits, physically unreasonable parameters, or both.

The updated text is reproduced below, but the most important part is that “taken together our SV-AUC data for WT CTD strongly support the presence of tetrameric multimers.”

“Two state” monomer-dimer or monomer-trimer models and “three state” monomer-dimer-trimer models resulted in poorer fits and/or physically unreasonable sedimentation coefficients (**Table S1**). In contrast, excellent fits were found for a monomer-dimer-tetramer model with sedimentation coefficients of 1.54 S, 2.43 S, and 3.94 S that scale as expected for a CTD monomer, dimer, and tetramer (**Fig. 3B**). Using the data from the two highest concentrations simultaneously, best fits yielded a monomer-dimer dissociation constant $K_{d,mon-dim}$ of 2.4 mM (confidence interval corresponding to one standard deviation: 1.1 – 3.4 mM) and a dimer-tetramer $K_{d,dim-tet}$ of $2.7 \pm 0.6 \mu\text{M}$, suggesting that the dimer is only a weakly-binding intermediate, consistent with our simulated dimer ensemble, but that it cooperatively forms a tetramer, with 1000-fold tighter affinity for the dimer-tetramer transition than for monomer-dimer (**Fig. 3C**). At the 107 μM condition, 6.2 μM is predicted to be in the dimer form and 14.2 μM in the tetramer form. Additionally, a two-state monomer-tetramer model results in similar fit quality and shows comparable effective binding equilibria, as seen in the monomer-dimer-tetramer model (i.e. c_{eq} , the concentration for equal population of monomer and tetramer is in the range of 250 – 300 μM for both models). Both models are also consistent with previous analyses at slightly different conditions using NMR relaxation dispersion of TDP-43 CTD exchange between monomeric and multimeric structures¹⁶. Although it remains difficult to conclusively distinguish between models where the dimer is a very weakly populated intermediate (monomer-dimer-tetramer model) or a transiently populated transition state (monomer-tetramer model) in part due to the inability to make samples at higher concentrations due to aggregation, taken together our SV-AUC data for WT CTD strongly support the presence of tetrameric multimers.

Reviewer #4 (Remarks to the Author):

I thank the authors for addressing my previous concerns with their revised manuscript. Especially, the updated analysis of the AUC data demonstrates the monomer-dimer-tetramer model as the best fit, which effectively resolves many of my concerns regarding the computational parts. However, I suggest that the alternative models evaluated (i.e., monomer-dimer, monomer-trimer, monomer-tetramer, and monomer-dimer-trimer) and their respective fitting parameters (such as R_square or reduced chi_square values) be included in the Supporting Information for completeness and clarity. Additionally, based on the relaxation dispersion NMR data previously presented in Figure 7A of the 2016 Structure paper, I did a quick estimation of the apparent dissociation constant (K_d). Using the reported values of $k_{off} = 1100 \text{ s}^{-1}$ and $k_{on} = 100 \text{ s}^{-1}$, and considering the protein concentration ($[\text{protein}] = 107 \mu\text{M}$), the estimated K_d is approximately $(1100/100) \times 107 = 1177 \mu\text{M}$. This estimate aligns well with the magnitude derived from the current AUC fitting and may provide valuable complementary support. If I am correct on this estimation, I recommend that the authors consider including this comparative estimation to strengthen the coherence of their analyses.

Response to Reviewer #4

We thank the reviewer for their comments and now present the global reduced chi-square values for the various Lamm equation models in Supplementary Table S1. As noted in our response to Reviewer #2, a reanalysis of the sedimentation velocity data yields better fits and slightly different sedimentation coefficients and affinities.

We also appreciate the reviewer's suggestion to compare to our previous work more directly. We had not considered trying to extract an apparent K_d directly from that work. We note the challenges are higher given the complex equilibria and the potential confusion as the actual dissociation constant for a monomer-tetramer or monomer-dimer-tetramer is not easily expressed as a traditional dissociation constant. We have decided to add a description of the effective concentration where the monomer and the tetramer have equal populations in each model and we added the following comment regarding our previous work in the text:

Both models are also consistent with previous analyses at slightly different conditions using NMR relaxation dispersion of TDP-43 CTD exchange between monomeric and multimeric structures¹⁶.

Reviewer #5 (Remarks to the Author):

By incorporating the revisions and including microscopy analyses highlighting the distinct localization of the mutants (new Fig. S4C), the authors have adequately addressed my comments. Therefore, I recommend the manuscript for publication.

Response to Reviewer #5

We thank the reviewer for their thorough reading of the manuscript.

We appreciate the reviewer's comments and the opportunity to address the critiques regarding the conclusions. We have carefully addressed the comments and adjusted the text to be sure we do not assert conclusions beyond the resolution or reliability of the available data.

Reviewer #2 (Remarks to the Author):

The authors now present global reduced chi-square values for the fits with various Lamm equation models (Supplementary Table S1). They report a significantly better fit with the monomer–dimer–tetramer model, which has improved the manuscript.

The authors state solubility issues limit the highest concentration allowed in SV AUC experiments. While this is unfortunate, it is unsound to assert conclusions that extend beyond the resolution or reliability of the available data.

We fundamentally agree with the reviewer and appreciate the need for caution. We have carefully updated the manuscript. Our previous manuscript text could be seen as relying too heavily on the AUC of TDP-43 CTD WT to say that the multimers are tetrameric. Now, we have changed the manuscript to highlight that integrative lines of evidence support the model. In this case, we have 1) evidence from AUC for multimers larger than dimer that are most likely tetramer, but cannot say conclusively, and 2) additional evidence pointing to tetramers from AlphaFold2-Multimer combined with molecular simulation showing that the tetramer are the only stable structures and the only structures that explain all the prior and current mutagenesis and NMR data. This model is further supported by additional NMR and AUC on cross-linked forms. We stress again in the revised abstract and main text that the conclusions we draw are integrative and this is the best that we can do given the challenges and put forward the tetrameric models as the best models given the data.

We have updated the main text to add the following caution:

We caution that because samples at higher concentrations lead to aggregation, we cannot approach saturation of the multimeric forms. Hence, the SV-AUC data for WT CTD support the presence of multimers that are likely tetrameric, but we cannot conclusively establish the stoichiometry by AUC of WT CTD alone. Therefore, below we employ additional AUC on samples designed to boost self-assembly and complementary experimental and computational tools to further test this picture.

In the Conclusions, we stress that integrative analysis suggesting the tetrameric model and cast this as providing complementary evidence for, not conclusively showing, a specific multimeric state:

In this study, we provide evidence from complementary computational and experimental techniques suggesting that α -helical tetramers are the most likely native form of the CR that is important for its splicing and nuclear retention functions.

And to the main text Figure 3 caption we have added:

We note that the aggregation-prone nature of TDP-43 CTD prevented saturation of the multimeric form and hence the data suggest but cannot conclusively demonstrate that the most stable multimeric form is the tetramer.

The authors cite Brautigam (<https://pubmed.ncbi.nlm.nih.gov/21187153/>) and state that the paper “demonstrates that one does not require saturating concentrations to reliably measure an affinity due to the large number of data points fitted in the analysis (in this case, more than 71,000) and the extensive exploration of the parameter space.” They are referring to Lamm equation modeling by Brautigam in which an estimate of 100 uM was obtained for the K_d a monomer – dimer equilibrium in which the highest SV AUC loading concentration 87 uM. In a static system, this predicts monomer and dimer concentrations at equilibrium of 47 and 20 uM. In contrast, for the monomer - dimer - tetramer equilibria proposed by the authors with K_d's for the monomer -dimer and dimer - tetramer equilibria of 2.4 mM and 2.6 uM, respectively, the equilibrium concentrations of monomer, dimer and tetramer at a loading concentration of 107 uM would 91, 3 and 5 uM respectively. Thus, the comparison to Brautigam is not particularly convincing.

We agree with the reviewer that the comparison is not exactly the same situation. We did present our similar calculations of the populations and we agree that the assembled forms are not the majority population. While This does not prevent precise modeling, but, as we state above, does limit the confidence in the model from these data alone. Hence our changes described above.

The large data set argument is somewhat specious. For example, there are a lot more SV data points with interference optics compared to absorbance optics because of the data density and smaller minimum scanning interval. However, interference optics generally do not provide higher sensitivity or better estimates of fitting parameters than absorbance optics.

We agree with the reviewer that this is not a critical point and note that our discussion of the data set size does not appear in the text.

The integration of the manuscript with the Supplement lacks coherence. In the Supplement, the authors state “to overcome low solubility, we engineered a new CTD variant by introducing charged, glutamic acid residues at serine positions (S->E) in the disordered flanking regions of CTD ...” and that they “designed two obligate dimers with cross-linking site closer to CR at position S305 and S317 (S->E S305C and S->E S317C)”, a “the helix-breaking mutant, S->E A326P S305C dimer” and they “formed disulfide cross-links (S-S) between engineered cysteine residues at S273C”. The S273C mutant evidently is S->E S273C described in Supporting Fig. S9. SV-AUC experiments are described for these variants in the Supplement, but not in the manuscript.

We agree with the reviewer that the linking between the supporting text and the main text was not clear. We have now thoroughly updated and clarified the text and Supplemental Text to specify that we have AUC experiments both on S273C cross-linked obligate dimers as well as the S→E sequences with cross-links at a series of positions. We have updated the main text:

We previously showed that adding disulfide cross-links to WT TDP-43 CTD enhances CR interaction⁵ and we find here that it enhances multimerization as seen by AUC (Fig. S7A), but it also enhances aggregation. However, cross-linked S→E variants remain sufficiently soluble for AUC analysis across a range of concentrations to approach saturation of the assembly (Fig. S7,S9). These cross-linked variants populate multimeric assemblies requiring the helical region, as seen by AUC, and show enhanced helicity compared to the uncross-linked forms (Fig. S7,S9).

and the supplementary text:

Probing the assembly of TDP-43 CTD CR by biophysical techniques is complicated by aggregation at concentrations below those needed to saturate the CR multimerization. To drive TDP-43 CTD toward multimer formation, we formed disulfide cross-links (S-S) between engineered cysteine residues at S273C in TDP-43 CTD (with a sequence that is otherwise the wild-type sequence) (see Methods), as we previously used⁵.

Minor comments

The authors state SV experiments were conducted at 7.20 cm, which is not a meaningful statement in AUC. The figures show that SV data are analyzed at radial positions ranging from ~6.2 to 7.1 cm.

We thank the reviewer for their comment. We want to clarify that we do not state that the experiments were conducted at 7.20 cm, as is evident from the Lamm equation plots. Data is collected across a radial range of 6.0 to 7.2 cm, of which a subset contains the important information. There is a reason we stated: "Sedimentation velocity experiments were conducted at 25°C and 50,000 rpm (201,600 x g at 7.20 cm) ..." in the methods section. Our understanding is that Nature Communications and other Nature family journals require on reporting rotor speeds as relative centrifugal forces (g). Because this force depends on the radius, and because data modeling accounts for this fact, we report the g-force at a specific radius. To avoid confusion, we have modified the sentence in the manuscript as follows: "Sedimentation velocity experiments were conducted at 25°C and 50,000 rpm (corresponding to 201,600 x g at 7.20 cm) ..."

The authors state in the manuscript "for the non-self-associating species, linear plots describing the dependence of the sedimentation coefficient with concentration were generated in SigmaPlot (grafiti.com)". They are referring to a figure in the Supplement, which is not stated in the manuscript. Additionally, the plots are not linear, possibly because they are semi-log plots or because fitting polynomials of degree greater than 1 is linear regression.

We agree the previous description could be confusing. We have now extensively clarified this section of the Methods to remove "linear" and also reference the Figure S9 as well as make sure the methodological details for all samples are clearly stated. We also state in the caption of Figure S9 that we have used a log scale for the dimer concentration axis as well as explain that in the case of the non-self-associating species, there is a linear decrease in the sedimentation coefficient with increasing concentration due to non-ideality.